# TPV: Parameter Perturbations Through the Lens of Test Prediction Variance

**Devansh Arpit** [1]

## Abstract

We introduce *test prediction variance* (TPV)—the first-order sensitivity of a trained model's outputs to parameter perturbations—as a unifying framework for analyzing post-training robustness. TPV's trace form $\mathrm{Tr}(H_{\mathrm{eff}}C)$ separates the geometry of the trained model $H_{\mathrm{eff}}$ from the perturbation covariance $C$, placing SGD noise, label noise, quantization, and pruning under a single lens. The resulting expressions recover the wide-minima hypothesis for SGD and quantization noise, and yield a distinct Jacobian-spectral characterization for label noise connecting label-noise TPV with benign overfitting in nonlinear networks.

Theoretically, we prove that training-set TPV converges to its test-set counterpart in the overparameterized limit, irrespective of generalization performance, providing the first result that prediction variance under local parameter perturbations can be inferred from training inputs alone. Empirically, this stability holds far more broadly, including at very low widths. Further, TPV correlates well with test loss, enabling practical applications: JBR, a label-free pruning criterion derived from TPV geometry matching state-of-the-art baselines; and training-set based model selection signal for in-distribution and transfer learning scenarios. Code Available Here

## 1. Introduction

A central challenge in deep learning is understanding the robustness of a *specific trained model* to the perturbations it faces in practice: stochastic gradient noise near convergence, finite-precision arithmetic, label noise during fine-tuning, or post-training modifications such as pruning. Existing theoretical perspectives—wide minima (Hochreiter & Schmidhuber, 1997; Keskar & et al., 2017), implicit optimization

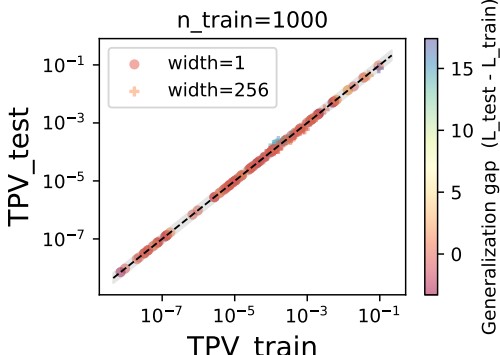

*Figure 1.* **TPV stability on synthetic data:** Each point corresponds to one synthetic configuration (dataset type, input dimension, network width, depth) and one perturbation source (label noise; SGD noise). Axes show empirical TPV on the training and test sets; $y = x$ is the ideal reference line along with a gray colored 50% error band; colormap indicates generalization gap $L_{\mathrm{test}} - L_{\mathrm{train}}$. We ran 324 configurations from label-noise and SGD-noise, spanning more than five orders of magnitude in TPV and different levels of generalization gaps. Despite this heterogeneity, all points concentrate tightly around the diagonal $\mathrm{TPV}_{\mathrm{train}} = \mathrm{TPV}_{\mathrm{test}}$; most surprisingly, even for width$= 1$. This demonstrates that: i) TPV stability holds true even at extremely low widths; ii) TPV stability is decoupled from generalization.

bias (Mandt et al., 2017; Smith & Le, 2018; Chaudhari & Soatto, 2018; Soudry et al., 2018; Zhang et al., 2017), benign overfitting (Belkin et al., 2019; Bartlett et al., 2020; Belkin et al., 2020), and NTK theory (Jacot et al., 2018; Adlam & Pennington, 2020a)—each illuminate part of the puzzle, but they operate through different analytical lenses and are rarely tied to a single quantity that directly governs the test-set behavior of a fixed, trained model under realistic post-training noise.

These perspectives share a common limitation: they ask *which* $w^\star$ the optimizer finds or prefers — through implicit bias, risk minimization, flat-minima selection, or infinite-width training dynamics — rather than characterizing the local robustness of a *given* $w^\star$ to the specific perturbations it faces *after* training. In practice, all of the perturbation sources above act *locally* around a fixed trained solution, not by retraining from scratch. This motivates a shift from global notions of prediction variability to a *local* quantity that measures the test sensitivity of $f(x; w^\star)$.

We formalize this sensitivity through the *test prediction*

[1]Modelable AI. Correspondence to: Devansh Arpit <devansharpit@gmail.com, devansh@modelable.ai>.

*Proceedings of the 43rd International Conference on Machine Learning*, Seoul, South Korea. PMLR 306, 2026. Copyright 2026 by the author(s).

*variance* (TPV): the local variance of a trained model's predictions under parameter perturbations. Crucially, note that this is different from global prediction variance (see Section 3). Under a first-order approximation, TPV reduces to a compact trace form

$$\text{TPV}(w) \approx \text{Tr}\big(H_{\text{eff}}\, C\big), \tag{1}$$

where $H_{\text{eff}}$ is second moment of the output-parameter Jacobian and $C = \mathbb{E}[\delta w \delta w^\top]$ is the perturbation covariance. This decomposition separates a *label-free geometric factor* $H_{\text{eff}}$ from the *noise mechanism* encoded in $C$ and reveals that diverse perturbations—including SGD noise, label noise, quantization noise, and pruning masks—all influence test predictions through the same curvature–covariance interaction. Variants of $\text{Tr}(H_{\text{eff}}C)$ appear in analyses of SGD dynamics (Zhu et al., 2018; Thomas et al., 2020; Bar et al., 2024). However, prior work does not interpret this object as a *predictive variance functional* nor use it as a unifying lens for heterogeneous real-world perturbations.

The key modeling insight is that *all* of these noise mechanisms can be well-approximated, near a minimum, as inducing small zero-mean parameter perturbations. Once this step is accepted, the first-order TPV expression $\text{Tr}(H_{\text{eff}}C)$ applies uniformly: every perturbation type acts through the same geometric factor $H_{\text{eff}}$ and is distinguished only by its covariance $C$. Our contributions are as follows:

1. **Test Prediction Variance (TPV) as a unified perturbation lens:** We formalize TPV as a local prediction-variance functional and show that SGD noise, label noise, quantization, and pruning all influence test robustness through the same trace form $\text{Tr}(H_{\text{eff}}C)$.

2. **TPV Trace Stability:** We prove that, in overparameterized networks, training-set TPV converges to test-set TPV, providing the first theoretical result showing that logit prediction variance under parameter perturbations, when evaluated on training-set inputs, is a reliable test-time estimator, irrespective of the model's generalization performance. Empirically, we find that TPV stability holds true even at very low widths and only breaks if either the number of samples are low or the induced perturbations are too large.

3. **Correlation with Test Loss:** We find that empirical TPV estimates correlate well with test loss. Together with TPV stability result, TPV unlocks training set based model-selection capabilities based on robustness and generalization behavior under targeted post-training noise source (e.g. label noise during fine-tuning, quantization, etc.) without access to test labels.

4. **Noise Sources:** We analytically derive TPV under label noise, SGD noise at convergence, and quantization.

For label noise, Theorem 4.2 recovers benign overfitting in linear models (Bartlett et al., 2020) as a special case and extends it to nonlinear networks via Jacobian geometry, explaining why overparameterization reduces label-noise sensitivity. For SGD and quantization noise, TPV recovers the wide-minima hypothesis.

5. **Applications:** We introduce JBR—a practical pruning criterion derived from TPV geometry—that matches or exceeds state-of-the-art baselines on CIFAR-10/100 and ImageNet. We further show four frequently used deployment scenarios where training-set based empirical TPV estimate can be used as a model-selection criteria without a test-set.

## 2. Relation to Prior Work

Work on double descent and benign overfitting explains how heavily overparameterized models can interpolate noisy data yet still generalize, typically by analyzing the test risk of minimum-norm or ridgeless interpolators as a function of model capacity or feature spectrum (Belkin et al., 2019; Bartlett et al., 2020). This line of work focuses on global risk curves and asymptotic regimes; in particular, the benign overfitting result for linear models is recovered as a special case of the TPV framework (Remark 4.1), which Theorem 4.2 extends to nonlinear networks by replacing the data matrix with the output-parameter Jacobian and characterizing label-noise sensitivity through its spectral geometry.

Work on implicit optimization bias studies how SGD dynamics select particular solutions among many interpolators — either by modeling SGD as sampling from a local Gibbs distribution (Mandt et al., 2017; Smith & Le, 2018; Chaudhari & Soatto, 2018), or by showing convergence to structured solutions such as max-margin classifiers (Soudry et al., 2018). Similarly, classical work on wide/flat minima (Hochreiter & Schmidhuber, 1997; Keskar & et al., 2017) and sharpness-aware training methods such as SAM (Foret et al., 2020) are concerned with how loss-landscape geometry correlates with generalization. These works characterize *which* $w^\star$ the optimizer finds. TPV takes a complementary perspective: given a fixed $w^\star$, it asks how test predictions vary under realistic parameter perturbations around it, and is agnostic to how $w^\star$ was reached.

Classical bias–variance analyses decompose prediction error into contributions from the mean predictor and its variability under retraining (Geman et al., 1992; Friedman, 1997). Neal et al. (2018) and Yang et al. (2020) show that in wide networks both global bias and variance can decrease together with width, and Adlam & Pennington (2020b) provide a fine-grained decomposition identifying regimes where both are small. Most relevantly, Bordelon & Pehlevan (2023) analyze prediction fluctuations in finite-width networks via dynamical mean-field theory, characterizing variance aris-

ing from kernel fluctuations *during training*. All of these are global notions, averaging over the distribution of solutions produced by the learning algorithm. In contrast, TPV is a *local*, post-training quantity measuring how predictions at a *fixed $w^\star$* vary under realistic parameter perturbations; it exhibits a systematic positive correlation with test error in the low training loss regime and typically an inverse correlation in the high training loss regime. A potentially similar two-phase structure has been observed in information bottleneck analyses of training dynamics (Tishby & Zaslavsky, 2015; Shwartz-Ziv & Tishby, 2017), where post-fit training acts as a compression phase.

Our TPV trace-stability theorem (Theorem 3.1) uses NTK stability (Jacot et al., 2018; Allen-Zhu et al., 2019) to show that $\mathrm{Tr}(H_{\mathrm{eff}}\, C)$ estimated on the training set converges to its test-set counterpart in the overparameterized limit — providing theoretical grounding for training-set-based TPV estimation in finite networks.

## 3. Test Prediction Variance

Classical bias–variance analyses (Geman et al., 1992; Friedman, 1997) study the *global* prediction variance $\mathrm{Var}_w[f_w(x)]$ obtained by retraining the model under different sources of randomness (e.g., parameter initialization, stochastic gradient noise, stochastic regularization such as dropout). For instance, Neal et al. (2018) revisit this global notion for deep networks, analyzing the variability induced by initialization, data sampling, and optimization. Accordingly, global variance provides insight into properties of the *learning algorithm*—such as algorithmic stability, the double descent phenomenon, and the benefits of ensembling.

In contrast, test prediction variance (TPV) is a *localized* quantity that measures the effect of infinitesimal weight-space perturbations $\delta w$ around a *fixed* solution $w^\star$:

$$\mathrm{TPV} := \mathbb{E}_{x,\,\delta w}\Big[\big\| f_{w^\star + \delta w}(x) - f_{w^\star}(x)\big\|^2\Big]. \quad (2)$$

This local perspective is more directly appropriate for understanding robustness to realistic noise sources near or after convergence—label noise during fine-tuning, SGD stationary noise, quantization noise, dropout during inference, post-training pruning masks—all of which act as *perturbations around a trained model* rather than as full retraining procedures. In contrast, global variance averages over the entire distribution of solutions obtainable by the learning algorithm. The two quantities therefore capture fundamentally different notions of variability.

We now formalize test prediction variance and derive a compact spectral expression that will be reused in all settings.

### 3.1. Our starting point: expected test error

We posit that the relevant quantity is the *expected test error* of the trained model $w^\star$, decomposed into a model-bias term and a *test prediction variance* term that captures the effect of small parameter perturbations around $w^\star$.

Let $\delta w$ denote a zero-mean parameter perturbation drawn from a distribution $R$ that models the sources of noise acting near or after convergence (label noise at convergence, SGD stationary noise, finite precision, pruning masks, etc.), denote its covariance matrix by $C := \mathbb{E}_R[\delta w\, \delta w^\top] \in \mathbb{R}^{p\times p}$. The perturbed predictor is $f_{w^\star + \delta w}$. For a test pair $(x, y)$ with $y = f^\star(x)$ (noiseless labels), the expected test error is

$$\mathcal{E}_{\mathrm{test}} = \mathbb{E}_{x,y,R}\big[(f_{w^\star+\delta w}(x) - y)^2\big].$$

Expanding this quantity and using $\mathbb{E}_R[\delta w] = 0$ gives the decomposition

$$\mathcal{E}_{\mathrm{test}} = \mathrm{TPV} + \mathbb{E}_x\big[\underbrace{\big(f_{w^\star}(x) - f^\star(x)\big)^2}_{\mathrm{bias}^2}\big]. \quad (3)$$

where for a fixed test point $x$, we assume $f_w(x)$ is smooth in $w$ and linearize around $w^\star$:

$$f_{w^*+\delta w}(x) \approx f_{w^*}(x) + J(x)\delta w, \quad (4)$$

where

$$J(x) := \nabla_w f_{w^*}(x) \in \mathbb{R}^{1\times p} \quad (5)$$

is the Jacobian of the model output with respect to parameters at $w^* \in \mathbb{R}^p$. Under this approximation,

$$\begin{aligned}
\mathrm{TPV} &\approx \mathbb{E}_{x,R}[(J(x)\delta w)^2] \\
&= \mathbb{E}_{x,R}[\mathrm{Tr}(J(x)^\top J(x)\delta w\delta w^\top)] \\
&= \mathrm{Tr}(\mathbb{E}_x[J(x)^\top J(x)]\mathbb{E}_R[\delta w\delta w^\top]) \\
&= \mathrm{Tr}(H_{\mathrm{eff}}C),
\end{aligned} \quad (6)$$

where

$$H_{\mathrm{eff}} := \mathbb{E}_x[J(x)^\top J(x)] \in \mathbb{R}^{p\times p} \quad (7)$$

is the second moment of the Jacobian. Note that $H_{\mathrm{eff}}$ is not the Hessian in general, though it does become equivalent to the Hessian under special circumstances. Further, note that $H_{\mathrm{eff}}$ is taking expectation on the test distribution, while the second moment of the parameter perturbations $C$ either depends on the training dataset (e.g. label noise, SGD stationary noise) or is data-agnostic (e.g. quantization).

Eq. (6) is the central object in this paper: TPV, and in turn the expected test error (due to Eq. (3)), is controlled by the interaction between the spectrum of $H_{\mathrm{eff}}$ and the perturbation covariance $C$. See Appendix B for TPV equation under scalar vs vector output models.

### 3.2. TPV Trace Stability

In this section, we prove that for trained overparameterized networks of width $m \to \infty$, TPV estimated entirely using the training dataset acts as a good estimator of the test set TPV object, irrespective of a model's generalization performance. The analysis relies on simplifying assumptions (e.g., isotropic parameter perturbations) that enable theoretical tractability and serve to motivate the empirical TPV stability observed in more general settings.

Denote by $X_{\text{tr}} = \{x_i\}_{i=1}^n$ and $X_{\text{te}} = \{x_i\}_{i=1}^{n_{\text{te}}}$ training and test datasets sampled i.i.d. from the same underlying data distribution $\mathcal{D}$. Let $w^\star$ be the parameter vector obtained after training a deep network of width $m$ on $X_{\text{tr}}$. We denote the estimator of $H_{\text{eff}}$ on the dataset $X \in \{X_{\text{tr}}, X_{\text{te}}\}$ as:

$$H_{\text{eff}}(w^\star; X) := \frac{1}{|X|} \sum_{x_i \in X} J(x_i; w^\star)^\top J(x_i; w^\star).$$

**Theorem 3.1** (TPV Trace Stability)**.** *The following upper-bound holds for over-parameterized networks:*

$$\big| \text{TPV}(w^\star; X_{\text{tr}}) - \text{TPV}(w^\star; X_{\text{te}}) \big| \leq c_1 \text{Tr}(C). \quad (8)$$

*where $c_1 := \frac{(n_{\text{tr}} + n_{\text{te}})}{p} \varepsilon_{\text{NTK}} + o_{n_{\text{tr}}, n_{\text{te}}}(1)$ and $C$ is the parameter perturbation covariance matrix (assumed to be isotropic), such that $\varepsilon_{\text{NTK}} \to 0$ as $m \to \infty$ and $o_{n, n_{\text{te}}}(1) \to 0$ as $n, n_{\text{te}} \to \infty$.*

See Appendix A for proof. This result is important because it shows that, in sufficiently overparameterized networks, the TPV estimate $\text{Tr}(H_{\text{eff}}(w^\star; X_{\text{tr}}) C)$ and equivalently Eq. 2 (under the first order approximation) estimated using the training set already carries enough information to approximate the *test-set* TPV that governs test prediction variance.

The proof uses two observations–1. Jacot et al. (2018); Allen-Zhu et al. (2019) show that the NTK kernel remains stable during training; 2. due to random initialization and i.i.d. sampling of datasets, $H_{\text{eff}}(w_0; X)$ concentrates around its population counterpart in operator norm for both $X_{\text{tr}}$ and $X_{\text{te}}$ due to the Law of Large Numbers. Combining these two observations allows us to upper-bound the distance between the training set and test set TPV objects at $w^\star$.

## 4. Parameter Perturbation Sources

In this section, we analytically derive the specific form of the TPV formula (Eq. 6) under different parameter perturbation sources. Specifically, we revisit label noise, SGD mini-batch noise, and quantization noise, and show that the robustness benefits of over-parameterization and wide minima arise from the common mechanism of suppressing TPV.

### 4.1. Label Noise

In this section, we do not assume TPV stability and study the TPV object in its general form.

**Linear Case**

*Remark* 4.1 (TPV in Linear Case)**.** Let the training labels be generated as $y_i = \theta^{\star\top} x_i + \varepsilon_i$, where $\varepsilon_i \sim \mathcal{N}(0, \sigma_\varepsilon^2)$ are i.i.d. label-noise variables, and denote by $X \in \mathbb{R}^{n \times d}$ the training dataset matrix containing $n$ samples of dimension $d$. Assume: (i) $d \geq n$ and $X$ has full row rank, (ii) the data distribution is whitened so that $\mathbb{E}_x[xx^\top] = I_d$, and (iii) $\theta^\star$ lies in the row span of $X$. Let $w^\star$ be a minimizer of the empirical linear regression loss using SGD. Then the TPV under linear regression is given by,

$$\boxed{\text{TPV}_{\text{label}} = \sigma_\varepsilon^2 \text{Tr}\big((XX^\top)^{-1}\big).} \quad (9)$$

The proof is shown in Appendix C for completeness. Note that in the linear regression setting above, the "global" prediction variance $\mathbb{E}_x \text{Var}_\varepsilon(f_{w^\star}(x))$ obtained by retraining on different noisy label realizations coincides exactly with the *local* TPV quantity $\text{Tr}(H_{\text{eff}} C)$, and is a well known result (Bartlett et al., 2020) in literature characterizing benign overfitting in linear regression models. This is because the conditioning of the matrix $XX^\top$ improves as $d$ grows beyond $n$, resulting in lower $\text{TPV}_{\text{label}}$, making the model less sensitive to label noise. Even more specifically, if training and test distributions are identical, under isotropic/whitened assumptions on the rows of $X$, random matrix theory (Wishart) gives $\mathbb{E}[(XX^\top)^{-1}] \approx \frac{1}{d} I_n$ (when $n \ll d$). Therefore, for large enough $n$, $\text{TPV}_{\text{label}} \approx \sigma_\varepsilon^2 n/d$.

**Non-Linear Case**

**Theorem 4.2** (TPV in Non-Linear Case)**.** *Let the training labels be generated as $y_i = f_{\theta^\star}(x_i) + \varepsilon_i$, where $f_{\theta^\star}$ is any fixed target function and $\varepsilon_i$ are i.i.d. zero-mean noise variables with variance $\sigma_\varepsilon^2$. Let $w^\star$ be a parameter vector of the network satisfying $f_{w^\star}(x_i) = f_{\theta^\star}(x_i)$ for all training inputs. Under the first-order approximation of the network around $w^\star$, let $w^\star + \delta w$ be a stationary point of the MSE loss w.r.t. $\delta w$ when training on the noisy labels $y_i = f_{\theta^\star}(x_i) + \varepsilon_i$ in this linearized model. If $\delta w$ is chosen to be the **minimum-norm stationary point**, then the expected test prediction variance due to label noise is,*

$$\boxed{\text{TPV}_{\text{label}} \approx \sigma_\varepsilon^2 \sum_{i=1}^r \frac{B_{ii}}{s_i^2},} \quad (10)$$

*where:*

- *$s_i$'s ($i \in [r]$) are the nonzero singular values of the output-parameter Jacobian $J$ evaluated on the training set,*

- $B_{ii}$ denotes the $i$-th diagonal entry of $B := V^\top H_{\text{eff}} V$, where $V$ contains the right singular vectors of $J$, and $H_{\text{eff}}$ depends on the test distribution

The proof is provided in Appendix D. The linear result (Eq. 9) is a special case of Eq. 10 when the nonlinear Jacobian is replaced by the data matrix. To our knowledge, Eq. 10 is the first expression of label-noise–induced prediction variance for finite-width deep networks in terms of a test-Jacobian operator and a general parameter-noise covariance $C$, that also accommodates other perturbation sources (e.g., SGD noise, finite precision, pruning masks) within the same $\text{Tr}(H_{\text{eff}} C)$ template.

Equation 10 highlights that label-noise sensitivity is dominated by directions where $B_{ii}$ is large while $s_i$ is small—i.e., where the test-distribution Jacobian aligns with poorly conditioned training directions. This makes explicit how the *local geometry* around $w^\star$—particularly the conditioning and alignment of Jacobian modes—governs TPV.

**Connection to Over-Parameterization and Benign Overfitting.** Over-parameterization induces solutions whose Jacobians are well-conditioned. This phenomenon has been rigorously studied through NTK theory. Let $J(w) \in \mathbb{R}^{n \times p}$ denote the Jacobian of network outputs on the $n$ training points and let $G(w) = J(w)J(w)^\top$ be the empirical NTK matrix. When $p \gg n$, the non-zero singular values of $J(w)$ correspond to the eigenvalues of $G(w)$. For two-layer ReLU networks, Du et al. (2018) show that with sufficient width, the smallest eigenvalue $\lambda_{\min}(G(w_0))$ is bounded away from zero at initialization and remains so during training. Bombari et al. (2022) extend this to deep networks with $\Omega(n)$ parameters, establishing well-conditioned NTK structure at initialization. Moreover, Allen-Zhu et al. (2019) prove that SGD remains in a neighborhood of initialization where the NTK matrix stays close to its well-conditioned initial value.

Consequently, in the overparameterized regime, the non-zero singular values of $J(w)$ remain bounded away from zero, suppressing the $\frac{B_{ii}}{s_i^2}$ terms in Eq. 10 and thereby yielding lower label-noise TPV. In this sense, **overparameterization and benign overfitting in the nonlinear case can be interpreted through the TPV lens**: overparameterized networks have stable Jacobian geometry, with small TPV, hence better robustness to label noise.

**Pathological Cases** Two pathological regimes are worth noting w.r.t. Eq. 10. (i) When the linearized system around $w^\star$ interpolates the noisy labels (e.g., $J\delta w = \varepsilon$ with full row rank), both the training TPV and the label-noise TPV in Eq. 10 collapse to $\sigma_\varepsilon^2$ (or to 0 as $\sigma_\varepsilon^2 \to 0$) if the training and test distributions match. TPV stability then holds trivially, but TPV becomes uninformative across architectures. (ii) When $\sigma_\varepsilon^2$ is sufficiently large that the noisy solution either

leaves the linearization regime (breaking the assumptions of Theorem 4.2) or induces a large $\text{Tr}(C)$ making the upper bound in Theorem 3.1 weak, it can both cause TPV stability to break. The non-degenerate, small-but-finite noise regime—where **interpolation does not occur and sufficiently small perturbations are used, and the min-norm solution is achieved**—is the regime in which label noise TPV exhibits meaningful geometric dependence and is useful. However, it is not feasible to find this exact solution for modern deep networks. See Appendix D.2 and D.3 for a detailed discussion on the label noise TPV object, and D.3.4 for a practical label noise TPV estimation algorithm.

### 4.2. SGD Stationary Noise Near Convergence

**Theorem 4.3** (TPV under SGD Noise at Convergence). *Consider a scalar-output non-linear model $f(x; w)$ trained using SGD with squared loss $L$ on a fixed dataset $\{(x_i, y_i)\}_{i=1}^n$ sampled i.i.d. from an underlying distribution $\mathcal{D}$ without any label noise, and assume that there exists a minimizer $w^\star$ with small but nonzero residuals. Then,*

$$\boxed{\text{TPV}_{\text{SGD}} \approx \frac{\eta \sigma_\varepsilon^2}{2b} \text{Tr}(\nabla_w^2 L(w^\star))} \quad (11)$$

*where, $\eta$ and $b$ are the SGD learning rate and batch size, and $\sigma_\varepsilon^2$ denotes the variance of the residual error over the training samples at convergence.*

The proof is shown in Appendix E and assumes TPV stability because we use $H_{\text{eff}}$ computed on the training set.

**Connection to Wide-minima Hypothesis**: There has been much debate on whether or not the flatness of minima affects the generalization performance of the final model achieved at the end of the training process (Keskar & et al., 2017; Wu et al., 2022; 2018). The classic counter-argument against the wide-minima hypothesis by Dinh et al. (2017) is that by reparameterizing the weights of a network, the spectrum of the Hessian can be changed without changing the input-output function map.

However, in reality, noise sources like SGD mini-batch or finite precision prevent the weights from reaching $w^\star$ precisely. Theorem 4.3 shows that small perturbations in parameters due to a fixed SGD noise ($\eta/b$) in a sharp basin make the TPV (and in turn test error) large; thus, SGD effectively samples from a high-variance region of the loss surface, leading to unstable test behavior and degraded robustness despite low training error.

We also note that a common practice in the flatness literature is to evaluate flatness on the training set and correlate it with test-set generalization, implicitly assuming training-set flatness approximates its test-distribution counterpart. The SGD-noise TPV theorem along with TPV stability (Theorem 3.1) justify this substitution under i.i.d. sampling:

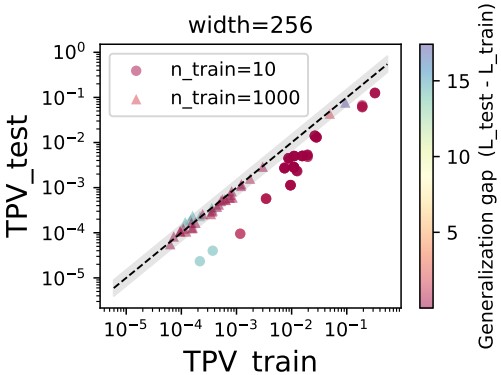

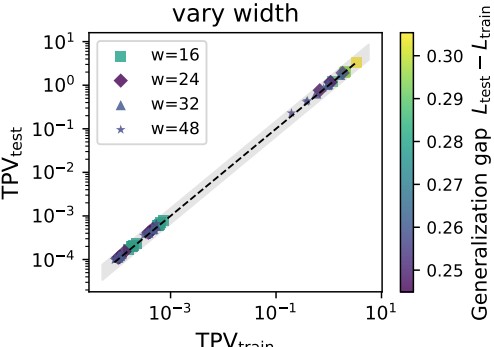

*Figure 2.* **TPV stability on synthetic data:** Analogous to Figure 1, Each point corresponds to one synthetic configuration and one perturbation source. We ran 324 configurations. In Figure 1, we fixed the number of training samples $n_{\text{train}} = 1000$ and varied network width. In this experiment, we fixed width to 256, and vary $n_{\text{train}}$. We find that **TPV stability breaks when $n_{\text{train}}$ is too low** (10; circles)– points are far outside the gray 50% error band, but holds for sufficiently large $n_{\text{train}}$.

training-set $\text{Tr}(H_{\text{eff}})$ approximates its test-distribution counterpart, and at squared-loss minima with small residuals (Appendix E.2) $\text{Tr}(H_{\text{eff}}) \approx \text{Tr}(\nabla_w^2 L(w^\star))$, completing the justification for using training-set sharpness as a proxy for test-distribution flatness.

### 4.3. Finite-precision noise

*Remark* 4.4 (TPV under Quantization Noise). Consider a scalar-output non-linear model $f(x; w)$ trained using GD (not SGD) with squared loss $L$ on a fixed dataset $\{(x_i, y_i)\}_{i=1}^n$ sampled i.i.d. from an underlying distribution $\mathcal{D}$ without any label noise, and assume that there exists a minimizer $w^\star$ with small but nonzero residuals. Then,

$$\boxed{\text{TPV}_{\text{quant}} \approx \frac{\delta^2}{12} \text{Tr}(\nabla_w^2 L(w^\star))} \tag{12}$$

where, $\text{Var}(\delta w_j) = \delta^2/12$ denotes each parameter's variance under quantization under a simple independent per-coordinate quantization model, $\delta w_j \sim \text{Unif}(-\delta/2, \delta/2), \forall j \in [p]$.

The proof is shown in Appendix F. Thus, similar to section 4.2, the above claim shows that sharper minima hurts robustness under quantization/finite precision.

## 5. Empirical Verification

We run experiments around TPV stability between training and test sets and empirically identify the conditions when it holds and when it breaks. We then study TPV under label noise and how width plays an important role in modulating it, and how it correlates with generalization. We do not include experiments on the relation between the flatness of minima (TPV under SGD noise) and generalization

*Figure 3.* **TPV stability on CIFAR-10:** Analogous to Figure 1, this scatter plot shows that **TPV stability holds for architectures with different widths** on CIFAR-10.

because several prior research works have confirmed a similar relationship (Keskar & et al., 2017; Smith & Le, 2018; Jastrzebski et al., 2017).

### 5.1. TPV Stability

Theorem 3.1 guarantees that training-set TPV approximates test-set TPV in the overparameterized limit with large $n$, relying on NTK stability and the Law of Large Numbers at initialization. In the experiments here we use the original form (Eq. 2) rather than the trace form, and find that TPV stability holds far more broadly than the theorem requires.

**Synthetic data:** We construct a large benchmark spanning 324 distinct configurations of dataset type, input dimension, network width, depth, and training-set size, with two perturbation sources (label noise and SGD stationary noise), for a total of $\sim$20 independent runs per configuration. Full details are in Appendix G.1.1. Figure 1 plots training-set vs. test-set TPV on a log-log scale for $n_{\text{train}} = 1000$, varying width. TPV values span more than five orders of magnitude, yet all points cluster tightly along the diagonal — most strikingly, stability holds even at width= 1. Points with large generalization gaps lie just as close to the diagonal as well-generalizing models, confirming that TPV stability is decoupled from generalization.

Figure 2 fixes width at 256 and varies $n_{\text{train}} \in \{10, 1000\}$: stability breaks severely at $n_{\text{train}} = 10$ but holds tightly at $n_{\text{train}} = 1000$.

**CIFAR-10/100:** Figure 3 shows that stability holds across MobileNetV2 width multipliers on CIFAR-10. Figure 12 shows that stability breaks at $n_{\text{train}} = 1$, holds mostly for $n_{\text{train}} = 10$, and is tight for $n_{\text{train}} = 10000$. Analogous CIFAR-100 results appear in Appendix G.1.2. Full experimental details are in Appendix G.1.2.

**Summary:** Three conclusions follow: (i) TPV stability holds regardless of the model's generalization gap; (ii) it

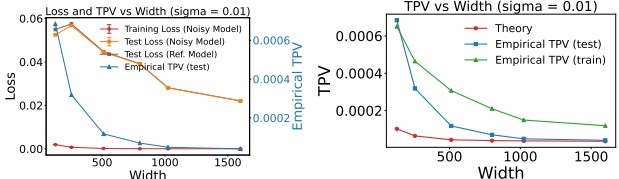

Figure 4. Empirical and theoretical TPV estimates under label noise on synthetic data for noise standard deviation $\sigma = 0.01$. As width increases, both TPV estimates reduce. Further, TPV correlates with test loss.

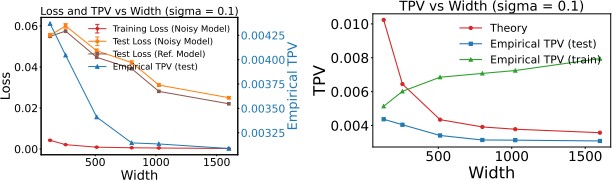

Figure 5. Empirical and theoretical TPV estimates under label noise on synthetic data for noise $\sigma = 0.1$. As width increases, theoretical TPV and empirical test set TPV reduce, but training set TPV increases, breaking TPV stability when $\sigma$ is large.

holds even at very small widths; (iii) it breaks only when $n_{\text{train}}$ is very small.[1]

## 5.2. Model Width and Label Noise TPV

We empirically study how network width modulates label-noise TPV (Theorem 4.2). Recall that for additive zero-mean label noise with variance $\sigma_\varepsilon^2$, sensitivity is governed by $T_{\text{base}} := \sum_{i=1}^{r} B_{ii}/s_i^2$, where $s_i$ are training-set Jacobian singular values and $B_{ii}$ reflects test-distribution Jacobian alignment. Over-parameterization keeps Jacobian singular values bounded away from zero, suppressing $T_{\text{base}}$ and yielding lower label-noise TPV. Our empirical goals are to verify that: (i) empirical TPV tracks the theoretical $T_{\text{base}}$ and decreases with width; (ii) TPV stability holds; (iii) lower TPV correlates with lower clean test loss.

**Synthetic data.** We train MLPs of varying widths on a synthetic Gaussian linear teacher task ($n_{\text{train}} = 1000$, $n_{\text{test}} = 5000$, input dim 20). Full details are in Appendix G.2.1. As width increases, models achieve lower test loss (benign overfitting), and both training-set and test-set empirical TPV decrease together with the theoretical prediction $\sigma^2 T_{\text{base}}$ (Figs. 4, 16).

At large noise ($\sigma = 0.1$), TPV stability breaks: training-set TPV saturates near $\sigma^2$ once the model fits the noisy targets, while test-set TPV continues to decrease with width and track the theoretical prediction (Fig. 5). We attribute this to the large perturbation covariance weakening the upper bound in Theorem 3.1.

---

[1]TPV stability (Eq. 2) is distinct from TPV trace stability (Eq. 6), which uses the trace form.

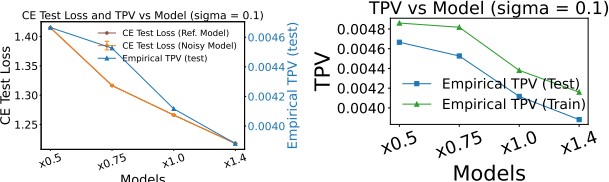

Figure 6. **Empirical TPV under logit noise on CIFAR-100 ($\sigma = 0.1$).** Both TPV estimates decrease with width and track the reference model's clean test cross-entropy loss.

**CIFAR-10/100.** We perform an analogous experiment on CIFAR-100 using pretrained MobileNetV2 models with varying width multipliers. For each reference model, clean logits are perturbed with i.i.d. Gaussian noise ($\sigma = 0.1$) and the model is fine-tuned on the noisy regression targets. Figure 6 shows that wider models achieve lower test cross-entropy and lower empirical TPV, with training-set and test-set TPV tracking each other closely. CIFAR-10 results and full experimental details appear in Appendix G.2.2.

## 5.3. Label Noise TPV and Generalization

The experiments in the previous section showed that empirical TPV correlates with test loss. We now examine the scope of this correlation more carefully. Fixing an MLP architecture on CIFAR-10 and sweeping a single regularizer (weight decay, dropout, or label smoothing) reveals a *U-shaped* relationship between TPV (local variance) and test loss (bias) in Eq. 3: both test-set TPV and test loss decrease together in the low training loss regime, until regularization becomes strong enough to induce underfitting, after which, TPV continues to decrease, while test loss rises in the high training loss regime. The latter part happens because TPV estimates the variance component of expected test error and is small when model underfits. Note that the relation between TPV and test loss is an empirical one. Figure 7 illustrates this setup for label smoothing regularization; the same pattern holds for dropout and weight decay (see Appendix G.3 for details). Across architectures, TPV[2] still correlates with test loss in the low training loss regime, though the correlation is noisier within architecture-regularization groups with similar loss values.

We further investigate the U-shaped relationship claim above by tracking training-set TPV across epochs along a single training trajectory. Specifically, we train ResNet-18 on CIFAR-100 with $30\%$ label noise with label smoothing regularization (Szegedy et al., 2016), and estimate empirical training-set TPV along the way. Fig. 8 plots training accuracy, validation accuracy, and training-set TPV against epoch. The vertical dashed line marks the epoch at which TPV achieves its maximum; this empirical landmark cleanly

---

[2]Note that the above experiment uses test-set TPV. In §6 we leverage TPV stability and use training set TPV instead to identify the best model/training recipe without any test labels.

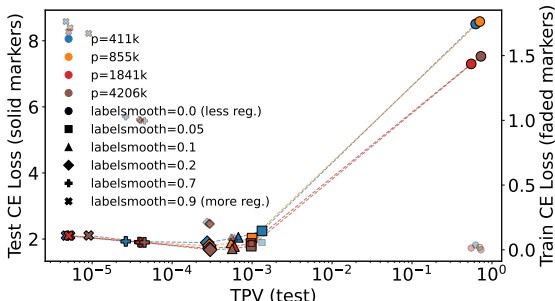

*Figure 7.* **TPV vs. test loss across architecture size and label smoothing strength on CIFAR-10.** Each color is one architecture; each marker shape is one label smoothing value. U-shape emerges: in the high training loss regime (left) lower TPV corresponds to higher train and test loss due to underfitting; in the low training loss regime (right) higher TPV corresponds to lower train loss and higher test loss due to overfitting.

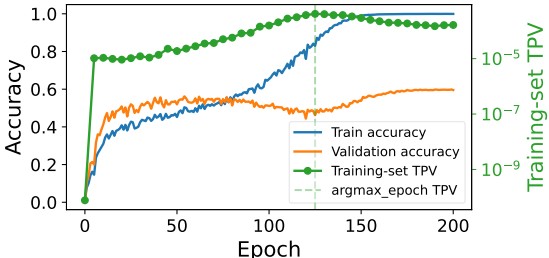

*Figure 8.* **TPV trajectory (ResNet-18 on CIFAR-100 with** $30\%$ **label noise).** TPV and validation accuracy are negative correlated in the low training loss regime (right of vertical green line).

separates two regimes that mirror the two branches of the U-shape in Figure 7. *Left of the line* (high-training-loss regime), training accuracy and TPV initially rise together (underfitted phase), and then validation accuracy dips after initial rise due to noisy label overfitting. *Right of the line* (low-training-loss regime), training accuracy saturates and validation accuracy exhibits a second rise—the epoch-wise double-descent pattern characteristic of training under label noise (Nakkiran et al., 2019). In this regime, TPV and validation accuracy are negatively correlated. See details in Appendix G.3.3 and additional NLU experiments using BERT in Appendix G.3.4.

# 6. Applications

TPV provides a training-set-only proxy for a model's sensitivity to specific post-training perturbations, without requiring test labels. We demonstrate two practical uses of this property: structured pruning (§6.1) and training-set-based model selection (§6.2).

## 6.1. Pruning

We adopt the viewpoint that pruning should preserve the model's predicted class on correctly-classified training samples. Modeling pruning as a structured parameter pertur-

bation within the TPV framework (Appendix H) yields *Jacobian-Based Rebalancing* (JBR), a label-free pruning criterion that assigns each parameter group $g$ the importance score

$$\text{score}_{\text{JBR}}(w_g) = \mathbb{E}_x\left[w_g^\top J_g(x)^\top \delta_u(x)\,\delta_u(x)^\top J_g(x)\,w_g\right]$$

where $J_g(x) = \partial f(x;w)/\partial w_g$ and $\delta_u(x) = \partial u(x;w)/\partial f(x;w)$ with $u(x;w) = -\log p_{c(x)}(x;w)$ being the negative log-probability of the predicted class. Groups with small scores contribute little to test prediction variance and are removed first. JBR is closely related to the Jacobian Criterion (JC) (Chen et al., 2025); their connection and differences are discussed in Appendix H.1.

**Results.** We evaluate JBR against seven baselines (Jacobian, L1, BN Scale, FPGM, WHC, Taylor, Random) following the OBC global channel-pruning protocol (Chen et al., 2025) with no fine-tuning between iterations. Results on CIFAR-10/100 and ImageNet are shown in Figures 21–24 in Appendix H; JBR matches or exceeds all baselines across all four settings.

## 6.2. Model Selection

A recurring practical challenge is selecting, from a pool of candidate models or training recipes, the one that will (i) generalize best within the same domain, (ii) generalize best in a new domain, or (iii) remain most robust when subsequently fine-tuned under label noise—all without access to test labels. We show that training-set TPV under label noise addresses all three scenarios. Notice we leverage TPV stability in these experiments.

**In-distribution training recipe selection:** Consider a practitioner who needs to select a training recipe among several recipes involving multiple hyperparameters. We sample five *joint* configurations (weight decay, dropout, label smoothing), each drawn independently from a candidate grid per regularizer, and train ResNet architectures of different sizes under each configuration. Figure 9 shows that training-set TPV ranks configurations consistently with test CE within each architecture (low training loss regime), aggregating the joint effect of multiple regularizers into a single scalar ranking that no individual regularizer value can recover. Experimental details appear in Appendix I.1.

**In-distribution cross-architecture model selection:** Consider a practitioner who has several trained architectures on a particular dataset and needs to select one with best generalization performance using a small training subset and does not have access to a validation set. We compute empirical TPV for several pretrained ImageNet architectures and compare against validation accuracy. As shown in Fig. 10, lower TPV tracks higher accuracy, and train/test

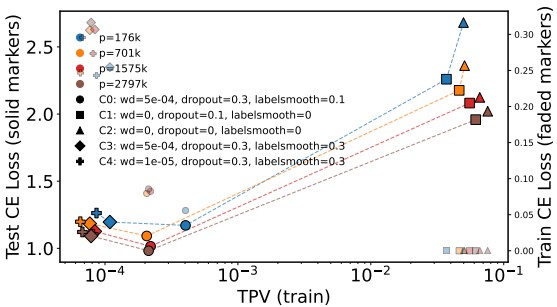

*Figure 9.* **TPV vs. test CE under multi-regularization on CIFAR-10.** Each color is one architecture; each marker is one of five joint configurations (weight decay, dropout, label-smoothing). TPV is proportional to test loss in the low training loss regime and inversely proportional to test loss in the high training loss regime.

TPV estimates remain closely aligned, enabling training-set-only model selection. See Appendix I.2 for details.

**Transfer learning with label noise:** Consider a practitioner who finetunes an architecture on a downstream dataset (with label noise) that is different from the original dataset the model was trained on, and needs to select the best training recipe without access to a clean validation set. We finetune four ImageNet-pretrained backbones on Oxford-IIIT Pets (Parkhi et al., 2012) with $10\%$ label corruption, under five joint regularization configurations each. As shown in Fig. 25, within each backbone lower training-set TPV consistently tracks higher validation accuracy; across architectures the negative trend holds but is weaker. See Appendix I.3 for details.

**Sensitivity to label noise—training recipe selection:** Consider a practitioner who has trained the same architecture under different hyperparameter choices and wants to select the recipe most robust to label noise during subsequent fine-tuning — without running any test evaluation. We train the same MLP architecture under seven weight-decay values and compare two training-data-only diagnostics for identifying the recipe most robust to subsequent label-noise fine-tuning: (a) *sharpness* $= \mathrm{Tr}(H_{\mathrm{eff}})$ estimated via Hutchinson's estimator; (b) training-set label-noise TPV. As shown in Figure 11, sharpness is not positively correlated with label-noise sensitivity, while label noise training-set TPV correctly identifies the most robust recipe. See Appendix I.4 for details.

## 7. Conclusion and future work

We introduced TPV as a local, post-training measure of how a trained model's predictions vary under parameter perturbations. Its trace form $\mathrm{Tr}(H_{\mathrm{eff}}C)$ separates model geometry from noise mechanism, allowing SGD noise, label noise, quantization, and pruning to be analyzed under a single framework: while SGD-noise and quantization TPV

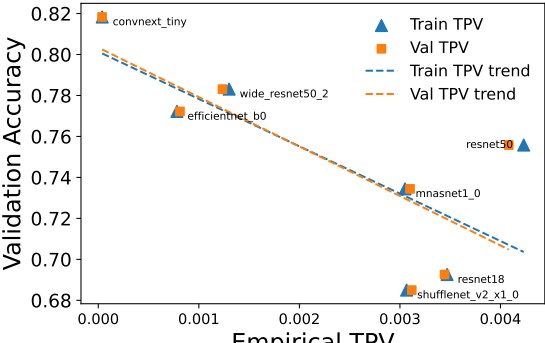

*Figure 10.* TPV estimates under label noise on Imagenet. TPV stability holds and models that generalize better typically have lower TPV estimates.

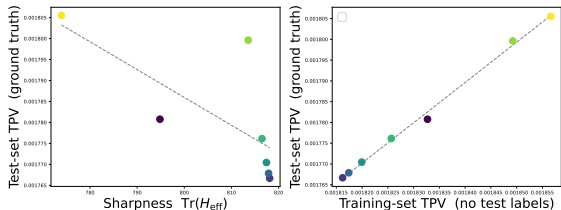

*Figure 11.* **Training recipe selection:** Each point is a model trained with a different weight decay value. *Left:* sharpness vs. test-set label noise TPV. *Right:* training-set vs. test-set label noise TPV. Training set label noise TPV is predictive of test set label noise sensitivity while sharpness (SGD noise TPV) is not.

recover the wide-minima hypothesis, label-noise TPV is governed by Jacobian spectral geometry explaining why overparameterization reduces label-noise sensitivity.

TPV stability (Theorem 3.1) — the result that training-set TPV reliably estimates its test-set counterpart — underpins the paper's practical contributions. It justifies using training-set TPV as a label-free proxy for test-time robustness, enabling training-set-based model selection across four deployment scenarios (§6.2). Empirically, stability holds far more broadly than the theorem requires, including at very low widths and under structured perturbations.

We also identify an empirical two-phase relationship between TPV and test loss: in the low training loss regime, TPV and test loss decrease together; in the high training loss regime, lower TPV corresponds to underfitting. This U-shaped pattern holds consistently across regularizers, architectures, and datasets, and is recoverable along a single training trajectory via the argmax-TPV landmark.

We believe the TPV framework opens several directions for future work: (i) extending TPV stability theory beyond NTK-based and isotropic assumptions, to close the gap with the broader empirical stability observed; (ii) formalizing TPV for input distribution shift; (iii) improving empirical estimation of label-noise TPV (Appendix D.3); (iv) extending the framework to losses beyond MSE and architectures with discrete or structured outputs.

## Impact Statement

This work advances understanding of how trained neural networks respond to the parameter perturbations they face in practice — stochastic gradient noise, finite-precision arithmetic, label noise during fine-tuning, and post-training pruning — by placing them under a single analytical lens. The practical benefits are training-set-only diagnostics for model selection and a label-free pruning criterion, which can reduce reliance on held-out labeled data and lower the compute cost of deployment. Our theoretical guarantees rely on overparameterized and isotropic-perturbation assumptions, and empirical TPV stability, while broader than the theory predicts, can break under very small training sets or large perturbations. We see no immediate societal risks specific to this work beyond those general to deep learning research.

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

# A. TPV Trace Stability via NTK Stability and LLN

We provide a finite-sample version of the TPV trace stability result used in the main text. Let

$$X_{\text{tr}} = \{x_i\}_{i=1}^{n_{\text{tr}}}, \qquad X_{\text{te}} = \{x_j'\}_{j=1}^{n_{\text{te}}}$$

denote the training and test sets, respectively, with both drawn i.i.d. from the same data distribution $\mathcal{D}$ independently of the network initialization.

For any dataset $X$ of size $n_X$, let $J_X(w) \in \mathbb{R}^{n_X \times p}$ be the Jacobian of network outputs with respect to parameters $w$. Following the main text, we define

$$H_{\text{eff}}(w; X) = \frac{1}{n_X} J_X(w)^\top J_X(w), \tag{13}$$

and the corresponding empirical NTK Gram matrix

$$G_X(w) = \frac{1}{n_X} J_X(w) J_X(w)^\top. \tag{14}$$

Note that the nonzero eigenvalues of $H_{\text{eff}}(w; X)$ and $G_X(w)$ are identical for every $w$.

The TPV trace on $X$ is

$$\text{TPV}(w; X) = \text{Tr}(H_{\text{eff}}(w; X) C), \tag{15}$$

where $C \succeq 0$ is the parameter-perturbation covariance.

## A.1. Assumptions

We make the following standard assumptions.

**Assumption A.1** (i.i.d. train/test). $X_{\text{tr}}$ and $X_{\text{te}}$ are drawn i.i.d. from the same distribution $\mathcal{D}$, independently of $w_0$.

**Assumption A.2** (Finite-set NTK stability). Let $w_t$ be the parameter vector after $t$ steps of gradient descent on $X_{\text{tr}}$, and let $w^\star := w_T$ be the trained network. In the infinite-width NTK regime ($m \to \infty$), individually for the fixed finite datasets $X \in \{X_{\text{tr}}, X_{\text{te}}\}$,

$$\| G_X(w^\star) - G_X(w_0) \|_{\text{op}} \leq \varepsilon_{\text{NTK}}, \tag{16}$$

with probability $\to 1$ as $m \to \infty$.

**Assumption A.3** (Isotropic Covariance). The perturbation covariance matrix is isotropic, i.e., $C = \sigma^2 \mathbf{I}_p$, where $p$ is the number of parameters and $\sigma$ is a scalar.

## A.2. Initialization Consistency via LLN

Define the per-example $H_{\text{eff}}$ contribution

$$h(x; w_0) := J(x; w_0)^\top J(x; w_0),$$

so that

$$H_{\text{eff}}(w_0; X) = \frac{1}{n_X} \sum_{x \in X} h(x; w_0).$$

**Lemma A.4** (LLN at initialization). *Let $\bar{H}_{\text{eff}}(w_0) := \mathbb{E}_{x \sim \mathcal{D}}[h(x; w_0)]$. If $\mathbb{E}\|J(x; w_0)\|_F^2 < \infty$ (i.e., well behaved output-parameter Jacobian), then under Assumption A.1,*

$$\|H_{\text{eff}}(w_0; X_{\text{tr}}) - H_{\text{eff}}(w_0; X_{\text{te}})\|_{\text{op}} = o_{n_{\text{tr}}, n_{\text{te}}}(1), \tag{17}$$

*where $o_{n_{\text{tr}}, n_{\text{te}}}(1)$ converges to 0 almost surely as $n_{\text{tr}}, n_{\text{te}} \to \infty$.*

*Proof.* Since $h(x; w_0) := J(x; w_0)^\top J(x; w_0)$ is integrable in Frobenius norm ($\mathbb{E}\|h(x; w_0)\|_F < \infty$) and takes values in the finite-dimensional normed space ($\mathbb{R}^{p \times p}, \| \cdot \|_F$), applying the strong law of large numbers (LLN) individually to $X \in \{X_{\text{tr}}, X_{\text{te}}\}$, we obtain:

$$\frac{1}{n_X} \sum_{i=1}^{n_X} h(x_i; w_0) \xrightarrow[n_X \to \infty]{\text{a.s.}} \mathbb{E}_{x \sim \mathcal{D}}[h(x; w_0)] = \bar{H}_{\text{eff}}(w_0),$$

in Frobenius norm. Using $\| \cdot \|_{\text{op}} \le \| \cdot \|_F$, we also have

$$\|H_{\text{eff}}(w_0; X) - \bar{H}_{\text{eff}}(w_0)\|_{\text{op}} \xrightarrow[n_X \to \infty]{\text{a.s.}} 0,$$

To obtain the difference bound (17), we use the triangle inequality in operator norm:

$$\|H_{\text{eff}}(w_0; X_{\text{tr}}) - H_{\text{eff}}(w_0; X_{\text{te}})\|_{\text{op}} \le \|H_{\text{eff}}(w_0; X_{\text{tr}}) - \bar{H}_{\text{eff}}(w_0)\|_{\text{op}}$$
$$+ \|\bar{H}_{\text{eff}}(w_0) - H_{\text{eff}}(w_0; X_{\text{te}})\|_{\text{op}}.$$

Each term on the right-hand side converges almost surely to 0 by the LLN argument above, so the sum converges to 0 almost surely as $n_{\text{tr}}, n_{\text{te}} \to \infty$. $\qquad\square$

## A.3. Main Result

We now state and prove a finite-sample TPV trace stability result.

**Theorem A.5** (TPV trace stability under isotropic perturbations). *Under Assumptions A.1, A.2, and A.3, we have*

$$\left| \text{TPV}(w^\star; X_{\text{tr}}) - \text{TPV}(w^\star; X_{\text{te}}) \right| \le \text{Tr}(C) \left( \frac{(n_{\text{tr}} + n_{\text{te}})}{p} \varepsilon_{\text{NTK}} + o_{n_{\text{tr}}, n_{\text{te}}}(1) \right), \tag{18}$$

*where $\varepsilon_{\text{NTK}} \to 0$ as $m \to \infty$ and $o_{n_{\text{tr}}, n_{\text{te}}}(1) \to 0$ as $n_{\text{tr}}, n_{\text{te}} \to \infty$.*

*Proof.* By Assumption A.3, $C = \sigma^2 \mathbf{I}_p$, so

$$\text{TPV}(w; X) = \text{Tr}\big(H_{\text{eff}}(w; X) \sigma^2 \mathbf{I}_p\big) = \sigma^2 \text{Tr}(H_{\text{eff}}(w; X))$$
$$= \sigma^2 \text{Tr}\left( \frac{1}{n_X} J_X(w)^\top J_X(w) \right) = \sigma^2 \text{Tr}\left( \frac{1}{n_X} J_X(w) J_X(w)^\top \right) = \sigma^2 \text{Tr}(G_X(w)). \tag{19}$$

We begin with a triangle inequality decomposition:

$$\left| \text{TPV}(w^\star; X_{\text{tr}}) - \text{TPV}(w^\star; X_{\text{te}}) \right|$$
$$\le \left| \text{TPV}(w^\star; X_{\text{tr}}) - \text{TPV}(w_0; X_{\text{tr}}) \right| + \left| \text{TPV}(w_0; X_{\text{tr}}) - \text{TPV}(w_0; X_{\text{te}}) \right| + \left| \text{TPV}(w_0; X_{\text{te}}) - \text{TPV}(w^\star; X_{\text{te}}) \right|. \tag{20}$$

**Step 1: Controlling the train/test drift from initialization via NTK stability:** Fix a dataset $X$ of size $n_X$. Using (19),

$$\left| \text{TPV}(w^\star; X) - \text{TPV}(w_0; X) \right| = \sigma^2 \left| \text{Tr}\big(G_X(w^\star) - G_X(w_0)\big) \right|.$$

For any matrix $A \in \mathbb{R}^{n_X \times n_X}$, $|\text{Tr}(A)| \le \text{rank}(A) \|A\|_{\text{op}} \le n_X \|A\|_{\text{op}}$, hence

$$\left| \text{TPV}(w^\star; X) - \text{TPV}(w_0; X) \right| \le \sigma^2 n_X \|G_X(w^\star) - G_X(w_0)\|_{\text{op}}.$$

Applying Assumption A.2 yields, for $X \in \{X_{\text{tr}}, X_{\text{te}}\}$,

$$\left| \text{TPV}(w^\star; X) - \text{TPV}(w_0; X) \right| \le \sigma^2 n_X \varepsilon_{\text{NTK}}. \tag{21}$$

**Step 2: Controlling the initialization train–test gap via LLN:** Using (15) and Assumption A.3,

$$\text{TPV}(w_0; X) = \sigma^2 \text{Tr}(H_{\text{eff}}(w_0; X)).$$

Therefore,

$$\left| \text{TPV}(w_0; X_{\text{tr}}) - \text{TPV}(w_0; X_{\text{te}}) \right| = \sigma^2 \left| \text{Tr}(H_{\text{eff}}(w_0; X_{\text{tr}}) - H_{\text{eff}}(w_0; X_{\text{te}})) \right|.$$

For symmetric $B \in \mathbb{R}^{p \times p}$, $|\text{Tr}(B)| \le p\|B\|_{\text{op}}$, so

$$\left| \text{TPV}(w_0; X_{\text{tr}}) - \text{TPV}(w_0; X_{\text{te}}) \right| \le \sigma^2 p \|H_{\text{eff}}(w_0; X_{\text{tr}}) - H_{\text{eff}}(w_0; X_{\text{te}})\|_{\text{op}}.$$

Applying Lemma A.4 gives

$$\left| \text{TPV}(w_0; X_{\text{tr}}) - \text{TPV}(w_0; X_{\text{te}}) \right| \le \sigma^2 p \cdot o_{n_{\text{tr}}, n_{\text{te}}}(1). \tag{22}$$

**Step 3: Combine:** Plugging (21) (for $X_{\text{tr}}$ and $X_{\text{te}}$) and (22) into (20) yields

$$\left|\text{TPV}(w^\star; X_{\text{tr}}) - \text{TPV}(w^\star; X_{\text{te}})\right| \leq \sigma^2 \, n_{\text{tr}} \, \varepsilon_{\text{NTK}} + \sigma^2 \, p \cdot o_{n_{\text{tr}}, n_{\text{te}}}(1) + \sigma^2 \, n_{\text{te}} \, \varepsilon_{\text{NTK}},$$

which is exactly (18). □

## B. TPV for Scalar and Vector Output Models

We clarify the definition of the $H_{\text{eff}}$ and Test Prediction Variance (TPV) for both scalar- and vector-output models.

**Scalar-output case.** Let $f_w(x) \in \mathbb{R}$ be a scalar-output model with parameters $w \in \mathbb{R}^p$, and define the Jacobian $J(x; w) = \nabla_w f_w(x) \in \mathbb{R}^{1 \times p}$. For a dataset $X = \{x_i\}_{i=1}^n$, $H_{\text{eff}}$ is

$$H_{\text{eff}}(w; X) \;=\; \frac{1}{n} \sum_{i=1}^n J(x_i; w)^\top J(x_i; w) \;=\; \frac{1}{n} J_X(w)^\top J_X(w),$$

where $J_X(w) \in \mathbb{R}^{n \times p}$ stacks the per-sample Jacobians. For a zero-mean parameter perturbation $\delta w$ with covariance $C$, the (first-order) TPV is

$$\text{TPV}(X) \;=\; \frac{1}{n} \mathbb{E}_{\delta w} \left[ \|f_{w+\delta w}(X) - f_w(X)\|_2^2 \right] \;\approx\; \text{Tr}(H_{\text{eff}}(w; X) \, C).$$

**Vector-output case.** Let $f_w(x) \in \mathbb{R}^K$ (e.g., logits) and $J(x; w) = \nabla_w f_w(x) \in \mathbb{R}^{K \times p}$. $H_{\text{eff}}$ is defined as

$$H_{\text{eff}}(w; X) \;=\; \frac{1}{n} \sum_{i=1}^n J(x_i; w)^\top J(x_i; w),$$

which is again a $p \times p$ matrix. The definition of TPV now becomes,

$$\text{TPV}(X) \;=\; \frac{1}{n} \mathbb{E}_{\delta w} \left[ \sum_{i=1}^n \|f_{w+\delta w}(x_i) - f_w(x_i)\|_2^2 \right] \;\approx\; \text{Tr}(H_{\text{eff}}(w; X) \, C).$$

In particular, for isotropic perturbations $C = \sigma^2 I$, $\text{TPV}(X) = \sigma^2 \, \text{Tr}(H_{\text{eff}}(w; X))$ in both cases.

## C. TPV for Linear Regression under Label Noise: Benign Overfitting

From $y = X\theta^\star + \varepsilon$ and the definition of $w^\star$,

$$w^\star = X^\top (XX^\top)^{-1}(X\theta^\star + \varepsilon) = \theta^\star + X^\top (XX^\top)^{-1}\varepsilon,$$

using the assumption that $\theta^\star$ lies in the row span of $X$. Hence

$$\delta w := w^\star - \theta^\star = X^\top (XX^\top)^{-1}\varepsilon,$$

so the parameter covariance is

$$C = \text{Cov}(\delta w) = \sigma^2 X^\top (XX^\top)^{-2}X.$$

By whitened input distribution assumption, $H_{\text{eff}} = \mathbb{E}_x[xx^\top] = I_d$, so

$$\text{Tr}(H_{\text{eff}}C) = \text{Tr}(C) = \sigma^2 \text{Tr}\big(X^\top (XX^\top)^{-2}X\big).$$

Using the cyclic property of trace, $\text{Tr}\big(X^\top (XX^\top)^{-2}X\big) = \text{Tr}\big((XX^\top)^{-1}\big)$, yielding

$$\text{Tr}(H_{\text{eff}}C) = \sigma^2 \text{Tr}\big((XX^\top)^{-1}\big),$$

which matches the classical expression for the expected test prediction variance in overparameterized linear regression.

# D. TPV for Non-Linear Regression Under Label Noise

### D.1. Proof

We consider a scalar-output model $f_w : \mathbb{R}^d \to \mathbb{R}$ with parameters $w \in \mathbb{R}^p$, trained on a dataset $\{(x_i, y_i)\}_{i=1}^n$, where $y_i = f_{\theta^\star}(x_i)$ (without noise) for some fixed function $f_{\theta^\star}$. Let $w^\star$ denote a parameter vector that satisfies $f_{w^\star}(x_i) = f_{\theta^\star}(x_i)$ for all training inputs.

We now perturb the training labels by additive noise:

$$y' \;=\; y + \varepsilon,$$

where $y = (y_1, \ldots, y_n)^\top \in \mathbb{R}^n$ is the original label vector and $\varepsilon \in \mathbb{R}^n$ is the label perturbation (noise).

We are interested in the stationary points $w^\star + \delta w$ of the squared loss with respect to the perturbed labels $y'$ under the first order Taylor's expansion around $w^\star$. Under this approximation, the loss for $w^\star + \delta w$ and labels $y' = y + \varepsilon$ is approximated by

$$L(\delta w; y') := \frac{1}{2}\big\|f(w^\star) + J\delta w - (y + \varepsilon)\big\|^2. \tag{23}$$

where,

$$f(w) \;:=\; \begin{bmatrix} f_w(x_1) \\ \vdots \\ f_w(x_n) \end{bmatrix} \in \mathbb{R}^n,$$

and similarly defining the Jacobian of the training outputs w.r.t. parameters at $w^\star$,

$$J \;:=\; \begin{bmatrix} g(x_1)^\top \\ \vdots \\ g(x_n)^\top \end{bmatrix} \in \mathbb{R}^{n \times p}, \tag{24}$$

Assuming that $w^\star$ already fits the original labels well, so that $f(w^\star) = y$, we approximate

$$f(w^\star) + J\delta w - (y + \varepsilon) \;=\; J\delta w - \varepsilon,$$

and hence

$$L(\delta w; y') \;=\; \frac{1}{2}\big\|J\delta w - \varepsilon\big\|^2. \tag{25}$$

Thus, in the linearized regime, the effect of label noise is captured by the least-squares problem

$$\min_{\delta w \in \mathbb{R}^p} \frac{1}{2}\big\|J\delta w - \varepsilon\big\|^2. \tag{26}$$

The gradient of $L(\delta w; y')$ with respect to $\delta w$ is

$$\nabla_{\delta w} L(\delta w; y') \;=\; J^\top(J\delta w - \varepsilon).$$

Setting this to zero yields the *normal equations*

$$J^\top J\delta w \;=\; J^\top \varepsilon. \tag{27}$$

Recall that we are interested in finding the min-norm solution among all the $\delta w$ that satisfy the above equation, i.e. $\arg\min_{\delta w}\{\|\delta w\|^2 s.t. \nabla_{\delta w}L(\delta w; y') = 0\}$. To analytically find this min-norm solution, we start with the compact SVD of $J$:

$$J \;=\; U_r S V_r^\top, \tag{28}$$

where

- $U_r \in \mathbb{R}^{n \times r}$ with orthonormal columns ($U_r^\top U_r = I_r$),

- $V_r \in \mathbb{R}^{p \times r}$ with orthonormal columns ($V_r^\top V_r = I_r$),

- $S \in \mathbb{R}^{r \times r}$ is diagonal with positive singular values $S = \mathrm{diag}(s_1, \ldots, s_r)$, where $s_i > 0$.

Substituting (28) into (27), we have

$$J^\top J \delta w = V_r S U_r^\top U_r S V_r^\top \delta w = V_r S^2 V_r^\top \delta w$$

and

$$J^\top \varepsilon = (U_r S V_r^\top)^\top \varepsilon = V_r S U_r^\top \varepsilon.$$

Thus the normal equations become

$$V_r S^2 V_r^\top \delta w = V_r S U_r^\top \varepsilon. \tag{29}$$

Left-multiplying by $V_r^\top$ and defining $\alpha := V_r^\top \delta w \in \mathbb{R}^r$, we obtain

$$S^2 \alpha = S U_r^\top \varepsilon \quad \Rightarrow \quad S\alpha = U_r^\top \varepsilon \quad \Rightarrow \quad \alpha = S^{-1} U_r^\top \varepsilon.$$

Any $\delta w$ can be decomposed into components along the range of $V_r$ and its orthogonal complement. Extending $V_r$ to an orthonormal basis $[V_r \; V_\perp] \in \mathbb{R}^{p \times p}$, we can write

$$\delta w = V_r \alpha + V_\perp \beta,$$

for some $\beta \in \mathbb{R}^{p-r}$. The normal equations constrain only the component along $V_r$, giving $\alpha = S^{-1} U_r^\top \varepsilon$, while $\beta$ is unconstrained (it lies in the nullspace of $J$).

The *minimum-norm* solution is obtained by setting $\beta = 0$, yielding

$$\delta w_{\min} = V_r S^{-1} U_r^\top \varepsilon. \tag{30}$$

For a new test input $x$, the first-order Taylor expansion gives

$$f_{w^\star + \delta w}(x) \approx f_{w^\star}(x) + g(x)^\top \delta w, \tag{31}$$

where $g(x) = \nabla_w f_{w^\star}(x)$. Note that this is the gradient of the output, not the loss. Hence the prediction change is

$$\delta f(x) := f_{w^\star + \delta w}(x) - f_{w^\star}(x) \approx g(x)^\top \delta w. \tag{32}$$

Plugging in the minimum-norm solution (30), we get

$$\delta f(x) \approx g(x)^\top V_r S^{-1} U_r^\top \varepsilon. \tag{33}$$

We now decompose the test gradient $g(x)$ in the parameter-space basis $[V_r \; V_\perp]$:

$$g(x) = V_r b(x) + V_\perp c(x),$$

where

$$b(x) := V_r^\top g(x) \in \mathbb{R}^r, \qquad c(x) := V_\perp^\top g(x) \in \mathbb{R}^{p-r}.$$

Using orthogonality $V_r^\top V_\perp = 0$, we have

$$g(x)^\top V_r S^{-1} U_r^\top \varepsilon = b(x)^\top S^{-1} U_r^\top \varepsilon.$$

Thus the prediction change for the minimum-norm solution is

$$\delta f(x) = b(x)^\top S^{-1} U_r^\top \varepsilon, \qquad b(x) = V_r^\top g(x). \tag{34}$$

We now treat the label noise as random from any distribution such that it has zero mean and covariance $\mathbb{E}[\varepsilon\varepsilon^\top] = \sigma_\varepsilon^2 I_n$.

Define

$$\eta := U_r^\top \varepsilon \in \mathbb{R}^r.$$

Since $U_r$ has orthonormal columns and $\varepsilon$ is isotropic, we have

$$\mathbb{E}[\eta] = 0, \qquad \mathbb{E}[\eta\eta^\top] = \sigma_\varepsilon^2 I_r.$$

Using (34), we can rewrite

$$\delta f(x) = b(x)^\top S^{-1}\eta.$$

For a fixed test input $x$, the mean and variance over label noise are:

**Mean.**

$$\mathbb{E}_\varepsilon[\delta f(x)] = b(x)^\top S^{-1}\mathbb{E}[\eta] = 0.$$

**Variance.**

$$\begin{aligned}
\mathbb{E}_\varepsilon\big[(\delta f(x))^2\big] &= \mathbb{E}_\varepsilon\big[b(x)^\top S^{-1}\eta\,\eta^\top S^{-1}b(x)\big] \\
&= b(x)^\top S^{-1}\mathbb{E}_\varepsilon[\eta\eta^\top]S^{-1}b(x) \\
&= \sigma_\varepsilon^2\, b(x)^\top S^{-2}b(x),
\end{aligned} \tag{35}$$

where $S^{-2} = S^{-1}S^{-1} = \mathrm{diag}(1/s_1^2, \ldots, 1/s_r^2)$.

To obtain the *expected test prediction variance*, we now average (35) over the test input distribution $x \sim P_X$. Define the second-moment matrix of the projected test gradients:

$$B := \mathbb{E}_{x\sim P_X}\big[b(x)b(x)^\top\big] \;\in\; \mathbb{R}^{r\times r}. \tag{36}$$

Then

$$\begin{aligned}
\mathbb{E}_{x,\varepsilon}\big[(\delta f(x))^2\big] &= \mathbb{E}_x\big[\mathbb{E}_\varepsilon\big[(\delta f(x))^2 \mid x\big]\big] \\
&= \mathbb{E}_x\big[\sigma_\varepsilon^2\, b(x)^\top S^{-2}b(x)\big] \\
&= \sigma_\varepsilon^2\,\mathbb{E}_x\big[\mathrm{Tr}(S^{-2}b(x)b(x)^\top)\big] \\
&= \sigma_\varepsilon^2\,\mathrm{Tr}\Big(S^{-2}\mathbb{E}_x[b(x)b(x)^\top]\Big) \\
&= \sigma_\varepsilon^2\,\mathrm{Tr}\big(S^{-2}B\big).
\end{aligned} \tag{37}$$

Since $S^{-2}$ is diagonal, we can write this explicitly as

$$\mathbb{E}_{x,\varepsilon}\big[(\delta f(x))^2\big] \;=\; \sigma_\varepsilon^2 \sum_{i=1}^r \frac{B_{ii}}{s_i^2}, \tag{38}$$

where $B_{ii} = \mathbb{E}_x[b_i(x)^2]$ is the expected squared projection of the Jacobian under the *test data distribution* onto the $i$-th right singular vector of the training dataset's *empirical Jacobian $J$*.

### D.2. Useful Regime and Pathological Case Analysis

In this section we resolve two subtle but important conceptual points regarding: i) the regime in which TPV stability holds and the label-noise TPV (Theorem 4.2) is useful; ii) pathological cases where label noise perturbations are too large and and TPV stability breaks.

If a model is sufficiently expressive, then it can interpolate arbitrarily small perturbations to the logits on the training set. If interpolation occurs, then we have $f_{w^\star+\delta w}(x) - f_{w^\star}(x) = \varepsilon$ on the training set and the training-set TPV (Eq. 2) becomes exactly $\sigma_\varepsilon^2$ for *every* such model.

If now, the test-set TPV from Theorem 4.2 still depends on the Jacobian spectrum and therefore varies across architectures, it may raise the concern that TPV stability has failed. We argue below that this case happens when the label noise is too

large to the point that either the first order assumption breaks (in which case neither Theorem 3.1 nor Theorem 4.2 apply), or the upper bound of the distance between training and test set TPV in Theorem 3.1 becomes too loose (since it directly depends on the trace of the perturbation covariance) and TPV stability does not hold based on the theorem.

If on the other hand, the label noise is not too large but the training and test distributions are identical, then test-set TPV becomes identical to the training set TPV ($\sigma_\varepsilon^2$). In this case, TPV stability holds trivially, but the label noise TPV object itself loses its discriminative power and is no longer informative.

To shed light on the above cases, we begin by noting that **Theorem 4.2 does *not* assume interpolation of noisy labels.** Theorem 4.2 analyzes the *local* linearized problem around the clean minimizer $w^\star$. Let $J \in \mathbb{R}^{n \times p}$ be the Jacobian of network outputs at the training inputs. Under label noise $\varepsilon$, the linearized loss is

$$L(\delta w) \;=\; \tfrac{1}{2}\|J\delta w - \varepsilon\|^2,$$

and we find the *minimum-norm* stationary point $\delta w_{\min}$ satisfying the normal equations $J^\top(J\delta w_{\min} - \varepsilon) = 0$. The solution is

$$\delta w_{\min} \;=\; V_r S^{-1} U_r^\top \varepsilon,$$

using the compact SVD $J = U_r S V_r^\top$ with rank $r \le n$. Theorem 4.2 itself makes no assumption that $J\delta w_{\min} = \varepsilon$; indeed,

$$J\delta w_{\min} \;=\; U_r U_r^\top \varepsilon,$$

which equals $\varepsilon$ *only* when $r = n$ (full row rank). Thus, interpolation of noisy labels is a special case and is not used anywhere in the proof of Theorem 4.2.

**Training TPV in four local regimes.** We now distinguish the four local regimes relevant for interpreting TPV.

**(1) Ridge/local regime:** Our experimental setup constrains the noisy models to remain close to $w^\star$, through explicit proximity regularization. In the linear approximation, this corresponds to solving the *ridge* problem

$$\min_{\delta w} \tfrac{1}{2}\|J\delta w - \varepsilon\|^2 + \tfrac{\lambda}{2}\|\delta w\|_2^2,$$

with $\lambda > 0$. The minimizer satisfies $(J^\top J + \lambda I)\delta w_\lambda = J^\top \varepsilon$, giving

$$J\delta w_\lambda \;=\; U_r \operatorname{diag}\!\Big(\frac{s_i^2}{s_i^2 + \lambda}\Big) U_r^\top \varepsilon,$$

where $s_i$ are the singular values of $J$. Hence

$$\mathrm{TPV}_{\text{train}} = \frac{1}{n}\mathbb{E}\big[\|J\delta w_\lambda\|_2^2\big] = \sigma_\varepsilon^2 \cdot \frac{1}{n}\sum_{i=1}^{r}\Big(\frac{s_i^2}{s_i^2 + \lambda}\Big)^2 \;<\; \sigma_\varepsilon^2,$$

with explicit dependence on the Jacobian spectrum. Even when $r = n$ (full row rank), the shrinkage factors $s_i^2/(s_i^2 + \lambda)$ are strictly less than 1, so the noisy labels are *not* interpolated and $\mathrm{TPV}_{\text{train}}$ remains strictly below $\sigma_\varepsilon^2$. The test TPV is given by Theorem 4.2 (with $S^{-2}$ replaced by ridge-shrunk directions), and Theorem 3.1 guarantees that train and test TPV remain close in this small-perturbation regime (in theory for extremely wide networks).

**(2) Pure least-squares, full-row-rank interpolation (degenerate case):** We now consider the case $\lambda = 0$, and the linearized least-squares problem admits an exact interpolating solution, and therefore, $\operatorname{rank}(J) = n$. Two related sub-regimes arise.

*(2a) Finite-variance interpolation:* When $r = n$ and $\lambda = 0$, the minimum-norm solution satisfies $J\delta w_{\min} = \varepsilon$, so the noisy labels are interpolated exactly in the linearized model. Consequently,

$$\mathrm{TPV}_{\text{train}} = \sigma_\varepsilon^2.$$

If the train and test distributions match and the Jacobian moments concentrate, then,

$$B \;=\; V^\top H_{\text{eff}}(w^\star; X_{\text{te}})V \;\approx\; V^\top\big(\tfrac{1}{n}VS^2V^\top\big)V \;=\; \tfrac{1}{n}S^2,$$

i.e., $B_{ii} \approx s_i^2/n$, and Theorem 4.2 gives

$$\text{TPV}_{\text{test}} \approx \sigma_\varepsilon^2 \frac{r}{n} = \sigma_\varepsilon^2.$$

Thus training and test TPV coincide exactly, and TPV stability holds trivially. However, in this interpolation regime TPV loses all discriminative power across architectures, since its value is the same constant $\sigma_\varepsilon^2$ for all sufficiently expressive models.

*(2b) Infinitesimal-variance interpolation ($\sigma_\varepsilon^2 \to 0$):* A more extreme version of the interpolation regime occurs when the variance of the injected logit noise is taken to be arbitrarily small. If $\sigma_\varepsilon^2$ is sufficiently small, then for any model with $r = n$ the optimization remains in the linear regime and still satisfies $J\delta w_{\min} = \varepsilon$. In this case the covariance of the parameter perturbation, $C = \mathbb{E}[\delta w_{\min} \delta w_{\min}^\top] = \sigma_\varepsilon^2 V S^{-2} V^\top$, shrinks to zero as $\sigma_\varepsilon^2 \to 0$. Therefore both

$$\text{TPV}_{\text{train}} \to 0, \qquad \text{TPV}_{\text{test}} \to 0,$$

and once again TPV stability holds trivially. As in sub-regime (2a), TPV becomes *uninformative*: although stability is preserved, TPV provides no basis for discriminating among models, as the entire TPV curve collapses to zero in the limit $\sigma_\varepsilon^2 \to 0$.

Across both sub-regimes (finite or infinitesimal noise), TPV stability remains intact but TPV becomes a constant across architectures, and thus ceases to encode meaningful geometric or generalization differences.

**(3) Low-rank or effectively narrow networks, and small perturbations:** If $r < n$, then even with $\lambda = 0$, the interpolation residual $\varepsilon_{\text{res}} = (I - U_r U_r^\top)\varepsilon$ is non-zero, and

$$\text{TPV}_{\text{train}} = \sigma_\varepsilon^2 \frac{r}{n} \ < \ \sigma_\varepsilon^2.$$

In this regime, both training and test TPV depend on $r$ and the Jacobian alignment encoded in $B$, and TPV is robustly discriminative across architectures. TPV stability becomes non-trivial follows from Theorem 3.1.

**(4) Large-noise (Figure 5):** There remains one additional regime, which we empirically probe in Fig. 5: the *large-noise* regime. For sufficiently large noise variance $\sigma_\varepsilon^2$, SGD driven by the noisy logits may move the parameters far away from $w^\star$. In this case, the parameter increment $\delta w$ is no longer small. This can lead to at least one of two situations: (i) the trace of the induced parameter covariance matrix is large, (ii) the actual solution reached by SGD is not well-approximated by the *local* min-(ridge-)norm solution of the linearized problem at $w^\star$ and the first order approximation breaks.

Empirically, in Figure 5, we observe that the training noise is effectively interpolated (the network fits the large noisy perturbations), so the *empirical* training TPV saturates near $\sigma_\varepsilon^2$ and becomes almost independent of width, while the test TPV still varies with width, but closely matches the theoretical TPV. Thus, while TPV stability breaks, theorem 4.2 holds. Since theorem 4.2 requires the first order approximation to hold, we conjecture that TPV stability breaks because the large noise induces parameter perturbations whose covariance matrix trace is large, which in turn weakens the upper bound in the TPV trace stability theorem 3.1 and diminishes the TPV stability guarantee.

**Summary:** The discussion above highlights that TPV stability does *not* fail in either of the interpolation regimes described in (2a)–(2b). When the linearized problem interpolates the noisy labels exactly (either for finite $\sigma_\varepsilon^2$ or in the limit $\sigma_\varepsilon^2 \to 0$), both the training and test TPV collapse to the same constant ($\sigma_\varepsilon^2$) or to zero, respectively. In these limits, Theorem 2.1 is satisfied trivially. What is lost in these regimes is not the validity of TPV stability but the *utility* of TPV as a discriminative signal across architectures—TPV becomes identical for all sufficiently expressive models.

On the other hand, if the parameter perturbations induced by label noise is too large, it can make the upper bound in the TPV stability theorem weak, or worse, break the first order assumption. This diminishes TPV stability guarantee. In this situation, if the first order assumption holds, theorem 4.2 remains applicable (Figure 5), however, TPV stability nonetheless fails and training set cannot be used for estimating robustness.

The practically relevant regime, and the one probed in our experiments (e.g. Figure 4), is the nontrivial, small perturbation setting where the optimization remains local, TPV stability holds, and the noisy labels are *not* interpolated in the linear approximation. In this regime, TPV retains meaningful dependence on the Jacobian geometry as predicted by Theorem 4.2 and training set estimate of TPV remains a good estimator of the test set estimate.

## D.3. Practical Computation of the Label–Noise TPV Quantity

### D.3.1. IDEALIZED LINEARIZED RESPONSE UNDER LABEL NOISE

The label–noise TPV theorem (Section 4.1) analyzes how *infinitesimal* perturbations to the *training labels* propagate through the optimization dynamics to induce fluctuations in *test predictions*. Let

$$J_{\mathrm{tr}} \in \mathbb{R}^{n_{\mathrm{tr}} \times p} \qquad \text{and} \qquad J_{\mathrm{te}} \in \mathbb{R}^{n_{\mathrm{te}} \times p}$$

denote the Jacobians of the training and test logits with respect to the model parameters, evaluated at the reference point $\theta^\star$.

For a perturbation $\varepsilon \in \mathbb{R}^{n_{\mathrm{tr}}}$ to the training labels, the idealized first–order parameter displacement is the *minimum–norm* solution of

$$\delta w^\star = \arg\min_{\delta w \in \mathbb{R}^p} \frac{1}{2} \big\| J_{\mathrm{tr}} \, \delta w - \varepsilon \big\|_2^2, \tag{39}$$

which has the closed–form expression

$$\delta w^\star = J_{\mathrm{tr}}^\top \big( J_{\mathrm{tr}} J_{\mathrm{tr}}^\top \big)^+ \varepsilon.$$

This induces the test–prediction perturbation

$$\delta f_{\mathrm{te}}^\star = J_{\mathrm{te}} \, \delta w^\star = J_{\mathrm{te}} J_{\mathrm{tr}}^\top \big( J_{\mathrm{tr}} J_{\mathrm{tr}}^\top \big)^+ \varepsilon.$$

Hence the label–noise TPV is

$$\mathrm{TPV}_{\mathrm{label}} = \mathbb{E}_\varepsilon \Big[ \big\| \delta f_{\mathrm{te}}^\star \big\|_2^2 \Big],$$

and therefore depends *jointly* on both the training and test Jacobians. The training Jacobian determines how label noise induces parameter motion, while the test Jacobian determines how that motion affects generalization.

### D.3.2. DUAL FORMULATION AND IDEAL NUMERICAL COMPUTATION

Equation (39) admits the well–known dual representation

$$\big( J_{\mathrm{tr}} J_{\mathrm{tr}}^\top + \lambda I \big) \alpha = \varepsilon, \qquad \delta w^\star = J_{\mathrm{tr}}^\top \alpha,$$

where $\lambda \geq 0$ is used for finite–variance perturbations or numerical stability. Thus, computing the exact linearized TPV requires repeatedly solving systems of the form

$$A\,\alpha = \varepsilon, \qquad A := J_{\mathrm{tr}} J_{\mathrm{tr}}^\top + \lambda I. \tag{40}$$

Importantly, iterative solvers such as conjugate gradient (CG) require only matrix–vector products of the form $\alpha \mapsto A\alpha$. These can be implemented via standard Jacobian–vector and vector–Jacobian products in modern autodiff systems without forming the full Jacobian. For smaller networks or modest $n_{\mathrm{tr}}$, CG converges rapidly, enabling exact numerical evaluation of the theoretical TPV.

### D.3.3. EMPIRICAL OBSERVATION: DEEP–NETWORK JACOBIANS ARE EXTREMELY ILL–CONDITIONED

While we were able to compute the SVD of the training set and test set Jacobians for the synthetic data experiments in Fig. 4, full SVD is not possible for modern architectures like CIFAR-10 MobileNetV2. So we attempted to compute the linearized label–noise TPV using the above CG formulation. Despite using:

- subsampled training sets ($n_{\mathrm{tr}} = 2000$–$4000$),

- ridge regularization ($\lambda > 0$),

- diagonal preconditioners via Hutchinson estimator,

CG *failed to converge* in essentially all settings. Residuals stagnated or oscillated rather than decreasing, and solutions did not approach the correct linearized displacement even after hundreds of iterations. Rayleigh–quotient diagnostics revealed spectral scales for $A$ spanning $10^6$–$10^8$, implying condition numbers far beyond the regime in which iterative solvers are computationally viable.

Constructing $A$ explicitly is also infeasible: even with $n_{\mathrm{tr}} = 4000$, computing $A$ requires thousands of Jacobian–vector products *per logit*. Thus, for large deep networks, the exact linearized TPV $\varepsilon^\top A^{-1} \varepsilon$ is a well–defined mathematical object but a *numerically intractable* one.

### D.3.4. PROPOSED LABEL NOISE TPV ALGORITHM

Because direct inversion of $J_{\text{tr}} J_{\text{tr}}^{\top}$ is numerically intractable for deep networks, we approximate the idealized TPV dynamics using a *local perturb–and–retrain procedure* show below. We perform $R$ independent optimization runs where each run performs the following procedure:

1. Initialize $w$ to the reference parameters $w^{\star}$.

2. Perturb the predicted labels on the training data: $y_i' = y_i + \varepsilon$, where $y_i := f_{w^{\star}}(x_i)$. We i.i.d. sample $\varepsilon \sim \mathcal{N}(0, \sigma^2)$, where $\sigma := c\,\sigma_0$ for all experiments throughout this paper unless specified otherwise. We typically use $\sigma_0 = 0.1$, and $c := \text{mean}_{x \in X}(|f_{w^{\star}}(x)|)$, where $X$ is a small subset of the training set, and the operator 'mean' computes the average over both samples in $X$ and logit dimensions (if $f$ has multi-dimensional output). The idea behind using adaptive scale $c$ is to scale noise proportional to the scale at which the model output distribution lives. We find this to be especially helpful when comparing cross-architecture TPV estimates (see §G.3 for a discussion).

3. Train $f_w$ on the training set for a small number of steps using SGD (or AdamW) without weight decay on the objective

$$\min_w \frac{1}{2} \sum_{i=1}^{n_{\text{train}}} \left\| f_w(x_i) - y_i' \right\|^2 \; + \; \frac{\gamma}{2} \|w - w^{\star}\|_2^2,$$

where the regularization is to ensure that the optimization remains in the local neighborhood of $w^{\star}$ where the linearized approximation is valid. We denote the resulting function as $f^{(r)}$, where $r$ denotes the $r^{th}$ run (out of $R$ runs). In most of our experiments, we found that $\gamma = 0$ work, i.e., no proximity regularization is needed. As a side note, we find that using gradient clipping typically hurts TPV estimation reliability, so we do not use it in our experiments.

Estimate TPV (on training or test set) as the empirical variance of the logit prediction across multiple independent perturbation runs:

$$\widehat{\text{TPV}} = \frac{1}{R} \sum_{r=1}^{R} \frac{1}{n} \sum_{i=1}^{n} \left\| f^{(r)}(x_i) - f_{w^{\star}}(x_i) \right\|_2^2, \tag{41}$$

where, $R$ is the number of runs with independent perturbations, $n$ is the number of samples, $f^{(r)}$ is the function learned by minimizing the objective from step 3 above, and $f^{\star}$ is the model with weights $w^{\star}$. The above TPV estimate is training set TPV if the samples used in the above equation are from the training set, and test set TPV if they come from the test set.

While the above objective does not yield the *exact* minimum–norm linearized solution, it provides a robust local approximation to the TPV dynamics in regimes where the exact computation is infeasible.

**Important Practical Considerations:** There are a couple of important practical considerations when using SGD based fine-tuning on noisy target logits for estimating label noise TPV:

1. Models need to be trained in eval mode: we perturb the logits of the clean model's prediction with Gaussian noise and train a copy of the reference model to fit these new targets, which act as infinitesimal change in targets. To achieve this faithfully, training must be done in eval mode, i.e., modules like batch norm and dropout should not be active. The reason is that if these modules are in training mode, even a zero variance perturbation causes the model to have perturbed targets, which conflicts with our goal.

2. Mini-batch shuffling: All sources of randomness other than label noise should be removed as much as possible to isolate the effect of label noise when measuring TPV (e.g. different mini-batch in each epoch). In practice, we do use mini-batch SGD for noisy label fine-tuning for efficiency and to make the training loss go down in some cases. However, we ensure that the sequence of mini-batch remains the same in each epoch or at the very least across different runs. Even though mini-batch gradient is a sum of the expected gradient plus a noise vector, removing random shuffling in this way ensures that each independent run sees the same sequence of mini-batches.

3. MSE training loss must go down during training. This can be easily overlooked, and if the loss does not go down or diverges, it can easily lead to incorrect TPV estimates. We found this to be especially true in the case of our preliminary ImageNet experiments, where is was extremely difficult to fit noisy target logits, and had to choose the experimental protocol carefully to avoid this problem.

# E. TPV for SGD Stationary Noise

**Setup.** Consider a scalar-output model $f(x; w)$ trained on a fixed dataset $\{(x_i, y_i)\}_{i=1}^n$ with squared loss

$$L(w) = \frac{1}{2n} \sum_{i=1}^n \varepsilon_i(w)^2, \qquad \varepsilon_i(w) := f(x_i; w) - y_i.$$

Let $J_i(w) := \nabla_w f(x_i; w) \in \mathbb{R}^{1 \times p}$ denote the output–parameter Jacobian row, and let $J(w) \in \mathbb{R}^{n \times p}$ be the matrix stacking these rows. Then the per-sample gradient is

$$g_i(w) := \nabla_w \ell_i(w) = \varepsilon_i(w) \, J_i(w)^\top.$$

## E.1. Relation Between Gradient Covariance Matrix and Hessian near Minima

Note that similar proportionality between SGD gradient covariance and curvature (Hessian / Fisher) has been shown in prior work (Wu et al., 2022; Mori, 2022; Mandt et al., 2016; Kühn & Rosenow, 2023), typically under population or online assumptions, specific model classes (linear networks, RFMs) or log-likelihood objectives. Here we make this connection explicit for finite-sample nonlinear squared-loss regression on a fixed dataset with no label noise.

Throughout this derivation we fix a parameter $w$ (later evaluated in a small-loss regime near an attractor $w^\star$) and study the covariance of the stochastic gradients induced by random mini-batching, without assuming any stochasticity in the dataset itself.

**Per-sample and mini-batch gradient covariance.** Let $I$ be a random index uniformly distributed in $\{1, \ldots, n\}$. Then

$$g_I(w) = \varepsilon_I(w) \, J_I(w)^\top.$$

Define the diagonal matrix

$$A_2(w) := \operatorname{diag}\big(\varepsilon_1(w)^2, \ldots, \varepsilon_n(w)^2\big),$$

and the full-batch gradient

$$g(w) = \nabla_w L(w) = \frac{1}{n} \sum_{i=1}^n g_i(w) = \frac{1}{n} J(w)^\top \varepsilon(w),$$

where $\varepsilon(w) \in \mathbb{R}^n$ stacks the residuals $\varepsilon_i(w)$.

The per-sample gradient covariance (over the random index $I$) is

$$\Sigma_{\text{sample}}(w) := \operatorname{Cov}(g_I(w)) = \mathbb{E}[g_I(w) g_I(w)^\top] - g(w) g(w)^\top \tag{42}$$

$$= \frac{1}{n} \sum_{i=1}^n \varepsilon_i(w)^2 \, J_i(w)^\top J_i(w) \; - \; \frac{1}{n^2} \Big( \sum_{i=1}^n \varepsilon_i(w) J_i(w)^\top \Big) \Big( \sum_{j=1}^n \varepsilon_j(w) J_j(w)^\top \Big)^\top \tag{43}$$

$$= \frac{1}{n} J(w)^\top A_2(w) J(w) \; - \; \frac{1}{n^2} J(w)^\top \varepsilon(w) \, \varepsilon(w)^\top J(w). \tag{44}$$

Thus we have the exact decomposition

$$\boxed{\Sigma_{\text{sample}}(w) = \frac{1}{n} J(w)^\top A_2(w) J(w) \; - \; \frac{1}{n^2} J(w)^\top \varepsilon(w) \, \varepsilon(w)^\top J(w).} \tag{45}$$

Now consider a mini-batch $B$ of size $b$, sampled uniformly with replacement, and the corresponding mini-batch gradient

$$g_B(w) := \frac{1}{b} \sum_{i \in B} g_i(w).$$

Conditioned on $w$, the covariance of $g_B(w)$ is

$$\Sigma_\xi(w) := \operatorname{Cov}(g_B(w) - g(w) \mid w) = \operatorname{Cov}(g_B(w) \mid w) = \frac{1}{b} \Sigma_{\text{sample}}(w).$$

Using (45), we obtain

$$\boxed{\Sigma_\xi(w) = \frac{1}{bn} J(w)^\top A_2(w) J(w) \; - \; \frac{1}{bn^2} J(w)^\top \varepsilon(w) \, \varepsilon(w)^\top J(w).} \tag{46}$$

**Approximation 1: dropping the $1/n^2$ term.** The second term in (46) carries an explicit factor $1/n^2$ and is the image, under $J(w)^\top(\cdot)J(w)$, of a rank-1 matrix $\varepsilon(w)\varepsilon(w)^\top$. In contrast, the first term scales as $1/n$ and involves a diagonal matrix $A_2(w)$. For large $n$, it is therefore natural to approximate

$$\Sigma_\xi(w) \approx \frac{1}{bn} J(w)^\top A_2(w) J(w). \tag{47}$$

**Approximation2: Trace and residual-variance approximation.** We now focus on the trace of the gradient noise covariance. Using cyclicity of the trace,

$$\mathrm{Tr}\big(\Sigma_\xi(w)\big) \approx \frac{1}{bn} \mathrm{Tr}\big(J(w)^\top A_2(w) J(w)\big) \tag{48}$$

$$= \frac{1}{bn} \mathrm{Tr}\big(A_2(w) J(w) J(w)^\top\big) \tag{49}$$

$$= \frac{1}{bn} \sum_{i=1}^n \varepsilon_i(w)^2 \, (J(w) J(w)^\top)_{ii}. \tag{50}$$

At the late-time SGD equilibrium, we model the residuals as random variables induced by the stationary distribution of $w_t$ and make the following approximation:

(i) the squared residuals $\varepsilon_i(w)^2$ are approximately i.i.d. across samples, with common variance $\mathbb{E}[\varepsilon_i(w)^2] = \sigma_\varepsilon^2$;

(ii) the residual magnitudes are approximately independent of the geometry encoded in the diagonal entries $(J(w)J(w)^\top)_{ii}$.

Under these assumptions,

$$\mathbb{E}\big[\mathrm{Tr}(\Sigma_\xi(w))\big] \approx \frac{1}{bn} \sum_{i=1}^n \mathbb{E}[\varepsilon_i(w)^2] \, (J(w) J(w)^\top)_{ii} \tag{51}$$

$$\approx \frac{\sigma_\varepsilon^2}{bn} \sum_{i=1}^n (J(w) J(w)^\top)_{ii} = \frac{\sigma_\varepsilon^2}{bn} \mathrm{Tr}\big(J(w) J(w)^\top\big). \tag{52}$$

Define the effective Jacobian curvature

$$H_{\mathrm{eff}}(w) := \frac{1}{n} J(w)^\top J(w), \tag{53}$$

so that $\mathrm{Tr}(J(w)J(w)^\top) = n\,\mathrm{Tr}(H_{\mathrm{eff}}(w))$. We obtain

$$\boxed{\mathbb{E}\big[\mathrm{Tr}(\Sigma_\xi(w))\big] \approx \frac{\sigma_\varepsilon^2}{b} \mathrm{Tr}\big(H_{\mathrm{eff}}(w)\big).} \tag{54}$$

Thus, up to a scalar factor determined by the residual variance and the mini-batch size $b$, the trace of the SGD gradient covariance is proportional to the trace of the effective Jacobian curvature $H_{\mathrm{eff}}$.

### E.2. Relation Between $H_{\mathrm{eff}}$ And Hessian Near Minima

For the squared loss, the Hessian of $L(w)$ is

$$\nabla_w^2 L(w) = \frac{1}{n} \sum_{i=1}^n \big(J_i(w)^\top J_i(w) + \varepsilon_i(w)\, \nabla_w^2 f(x_i; w)\big) \tag{55}$$

$$= \underbrace{\frac{1}{n} \sum_{i=1}^n J_i(w)^\top J_i(w)}_{H_{\mathrm{eff}}(w)} + R(w), \tag{56}$$

where

$$R(w) := \frac{1}{n} \sum_{i=1}^{n} \varepsilon_i(w) \, \nabla_w^2 f(x_i; w)$$

collects the second-derivative terms weighted by residuals. In particular,

$$\nabla_w^2 L(w) = H_{\text{eff}}(w) + R(w). \tag{57}$$

Suppose that $w$ lies in a small-loss regime (convergence) where $|\varepsilon_i(w)|$ is uniformly small for all $i$, and that the second derivatives $\nabla_w^2 f(x_i; w)$ are uniformly bounded in operator norm. Then $\|R(w)\|$ is small, and $H_{\text{eff}}(w)$ provides a good approximation to the true Hessian:

$$\|\nabla_w^2 L(w) - H_{\text{eff}}(w)\| = \|R(w)\| \;\le\; \frac{1}{n} \sum_{i=1}^{n} |\varepsilon_i(w)| \, \|\nabla_w^2 f(x_i; w)\| \;\ll\; \|H_{\text{eff}}(w)\|. \tag{58}$$

Consequently, in the late-time regime where training loss is small on all samples, we have

$$\boxed{H_{\text{eff}}(w) \approx \nabla_w^2 L(w)} \tag{59}$$

### E.3. SGD Late Dynamics and Relation Between TPV and Hessian

We consider the late-time dynamics of SGD near an attractor $w^\star$ of the training dynamics. Writing deviations $\delta w_t = w_t - w^\star$, a single SGD step with learning rate $\eta$ and mini-batch gradient $g_B(w_t)$ can be written as

$$\delta w_{t+1} \;=\; \delta w_t \;-\; \eta \, g_B(w_t). \tag{60}$$

Let $g(w_t) = \nabla L(w_t)$ denote the full-batch gradient of the empirical loss $L$, and define the mini-batch noise as

$$\xi_t \;:=\; g_B(w_t) - g(w_t), \qquad \mathbb{E}[\xi_t \mid w_t] = 0. \tag{61}$$

Near $w^\star$ we linearize the deterministic drift as

$$g(w_t) \;\approx\; \nabla_w^2 L(w) \, \delta w_t, \tag{62}$$

where $\nabla_w^2 L(w)$ is the Hessian. We now use the result from appendix E.2 stating $H_{\text{eff}} \approx \nabla_w^2 L(w)$. Thus,

$$g_B(w_t) \;\approx\; H_{\text{eff}} \, \delta w_t + \xi_t, \tag{63}$$

With this notation, the linearized SGD dynamics become

$$\delta w_{t+1} \;=\; (I - \eta H_{\text{eff}}) \, \delta w_t \;-\; \eta \, \xi_t \;=\; A \, \delta w_t - \eta \, \xi_t, \qquad A := I - \eta H_{\text{eff}}. \tag{64}$$

Let $C_t = \mathbb{E}[\delta w_t \delta w_t^\top]$ denote the covariance of the parameters at time $t$, and let $\Sigma_\xi = \text{Cov}(\xi_t)$ denote the covariance of the mini-batch noise at $w^\star$. Using the linear update and the fact that $\xi_t$ is independent of $\delta w_t$ and has zero mean, we obtain the standard covariance recursion

$$C_{t+1} \;=\; A C_t A^\top \;+\; \eta^2 \Sigma_\xi. \tag{65}$$

Assuming convergence to a stationary distribution, we set $C_{t+1} = C_t = C_{\text{sgd}}$ and obtain the discrete Lyapunov equation

$$C_{\text{sgd}} \;=\; A C_{\text{sgd}} A^\top \;+\; \eta^2 \Sigma_\xi. \tag{66}$$

Thus,

$$A C A^\top = (I - \eta H_{\text{eff}}) \, C_{\text{sgd}} \, (I - \eta H_{\text{eff}})^\top \tag{67}$$

$$= C_{\text{sgd}} - \eta(H_{\text{eff}} C_{\text{sgd}} + C_{\text{sgd}} H_{\text{eff}}^\top) + \eta^2 H_{\text{eff}} C_{\text{sgd}} H_{\text{eff}}^\top. \tag{68}$$

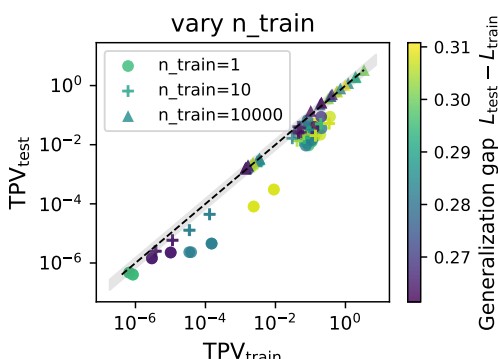

*Figure 12.* **TPV stability on CIFAR-10:** Analogous to Figure 2, this scatter plot shows that **TPV stability breaks for very low values of** $n_{\text{train}}$ and holds increasingly better (close to $y = x$ line and within the 50% error band) for larger values.

Plugging this into Eq. (66) and canceling $C_{\text{sgd}}$ on both sides yields

$$0 \;\approx\; -\eta(H_{\text{eff}}C_{\text{sgd}} + C_{\text{sgd}}H_{\text{eff}}^{\top}) + \eta^2 H_{\text{eff}}C_{\text{sgd}}H_{\text{eff}}^{\top} + \eta^2 \Sigma_\xi. \tag{69}$$

Neglecting $O(\eta^2)$ terms in the drift (the $H_{\text{eff}}C_{\text{sgd}}H_{\text{eff}}^{\top}$ term) under the assumption of a sufficiently small learning rate at convergence, we obtain the continuous-time Lyapunov equation

$$H_{\text{eff}}\, C_{\text{sgd}} \;+\; C_{\text{sgd}}\, H_{\text{eff}}^{\top} \;\approx\; \eta\, \Sigma_\xi. \tag{70}$$

Taking the trace of both sides gives

$$\text{Tr}\!\left(H_{\text{eff}}C_{\text{sgd}}\right) \;+\; \text{Tr}\!\left(C_{\text{sgd}}H_{\text{eff}}^{\top}\right) \;\approx\; \eta\, \text{Tr}(\Sigma_\xi). \tag{71}$$

Using cyclicity of the trace and the symmetry of $C_{\text{sgd}}$, we have

$$\text{Tr}\!\left(C_{\text{sgd}}H_{\text{eff}}^{\top}\right) \;=\; \text{Tr}\!\left((C_{\text{sgd}}H_{\text{eff}}^{\top})^{\top}\right) \;=\; \text{Tr}\!\left(H_{\text{eff}}C_{\text{sgd}}\right), \tag{72}$$

so that

$$\boxed{\text{Tr}\!\left(H_{\text{eff}}C_{\text{sgd}}\right) \;\approx\; \frac{\eta}{2}\, \text{Tr}(\Sigma_\xi)}. \tag{73}$$

Finally, we use the result from Appendix E.1 and E.2, and assuming that the Hessian is stable around the minimum $w^\star$, which lies at the center of the SGD stationary dynamics, we have that TPV under SGD noise is given by,

$$\boxed{\text{Tr}\!\left(H_{\text{eff}}C_{\text{sgd}}\right) \;\approx\; \frac{\eta\sigma_\varepsilon^2}{2b}\, \text{Tr}(\nabla_w^2 L(w^\star))} \tag{74}$$

where $\eta$ and $b$ are the SGD learning rate and batch size, and $\sigma_\varepsilon^2$ denotes the variance of the residual error over the training samples (assumed to be i.i.d.).

## F. TPV for Parameter Quantization Noise

TPV is given by $\text{Tr}(H_{\text{eff}}C)$. We show in Appendix E.2 that $H_{\text{eff}}(w) \approx \nabla_w^2 L(w)$ (Hessian) near minimum $w^\star$. Next, we compute the covariance of $\delta w$. Under $\delta w \sim \text{Unif}(-\delta/2, \delta/2)$, the variance is $(\delta/2)^2/3 = \delta^2/12$ by the standard result $\text{Var}(\text{Unif}(-a, a)) = a^2/3$. Thus under the quantization model, the parameter perturbation covariance is $C_{\text{quant}} \approx \frac{\delta^2}{12} I_p$, where $I_p$ is the identity matrix (thus obeying the TPV trace stability requirement), which proves the claim.

## G. Experiments

### G.1. Details of Experiments in Section 5.1

G.1.1. SYNTHETIC DATA EXPERIMENT:

For the synthetic experiments, we consider three families of data-generating processes: (i) a Gaussian linear teacher ($y = x^\top w_{\text{true}}$), (ii) a ReLU teacher ($y = \text{ReLU}(a^\top x)$), and (iii) a 10-unit multi-ReLU teacher ($y = \sum_{k=1}^{10} \text{ReLU}(a_k^\top x + b_k)$),

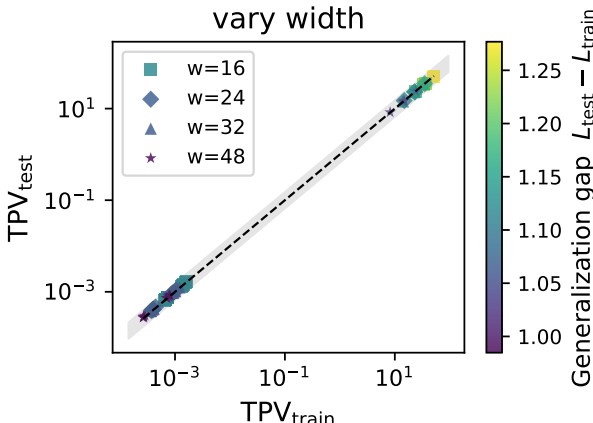
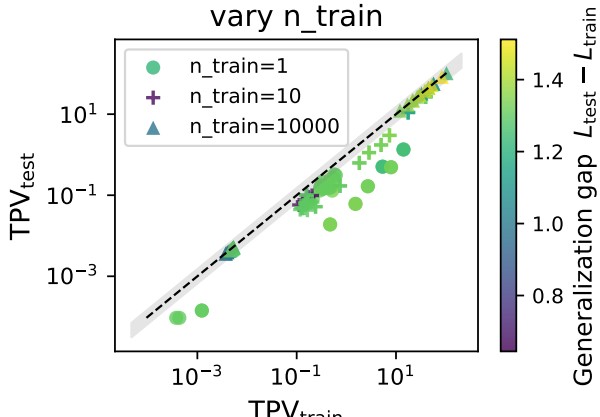

*Figure 13.* **TPV stability on CIFAR-100:** Analogous to Figure 1, this scatter plot shows that TPV stability holds for different width architectures on CIFAR-100.

*Figure 14.* **TPV stability on CIFAR-100:** Analogous to Figure 12, this scatter plot shows that TPV stability breaks for very low values of $n_{\text{train}}$ and holds increasing better (close to $y = x$ line and within the 50% error band) for larger values.

each with isotropic inputs $x \sim \mathcal{N}(0, I_d)$. We sweep over three input dimensions $d \in \{10, 20, 50\}$ and two training-set sizes $n_{\text{tr}} \in \{10, 1000\}$. The test set always contains 5000 samples. For the student network, we use fully-connected ReLU MLPs with widths $w \in \{1, 256\}$ and depths $\{2, 3, 4\}$.

**Clean reference model:** For every configuration (dataset type, $d$, $n_{\text{tr}}$, $w$, depth), we train a clean reference network $f_{w^\star}$ on 1000 samples of noiseless training data using full-batch SGD with learning rate $2 \times 10^{-3}$, cosine-annealing LR schedule, momentum 0.9, no weight decay, and 800 training epochs. The resulting parameters $w^\star$ and predictions $f_{w^\star}(x)$ on both train and test sets are cached and reused across all perturbation experiments.

**Label-noise perturbation:** For the label-noise experiments, we retrain from the fixed initialization $w^\star$ for $R = 20$ independent runs. Each run injects additive i.i.d. Gaussian noise $\varepsilon \sim \mathcal{N}(0, \sigma^2)$ with $\sigma \in \{0.005, 0.01\}$ to the clean labels. Each model is then trained for 200 epochs with full-batch GD, learning rate $2 \times 10^{-3}$, cosine annealing, momentum 0.9, and no weight decay. We record the empirical train/test TPV, train/test MSE, and a scalar first-order Taylor approximation error (described below). Denote the fine-tuned model in the $r^{th}$ run as $f^{(r)}$. Empirical TPV is computed as

$$\widehat{\text{TPV}}_{\text{train}} = \frac{1}{R n_{\text{train}}} \sum_{r=1}^{R} \sum_{i=1}^{n_{\text{train}}} \left\| f^{(r)}(x_i) - f_{w^\star}(x_i) \right\|_2^2,$$

and analogously for the test set. See Appendix D.3.4 for details on the Label Noise TPV estimation algorithm.

**SGD-noise perturbation:** To simulate stationary SGD noise around a minimum, we initialize the model at $w^\star$ and run SGD for 1000 steps with momentum 0.9 and no weight decay, using learning rates $\{10^{-3}, 5 \times 10^{-4}\}$ and batch sizes $\{32, 128\}$. The number of training samples $n_{\text{tr}}$ may be as small as 10, so we disable `drop_last` in the PyTorch `DataLoader` to avoid degenerate cases with empty batches. Snapshots are collected every 20 steps after a burn-in period of 200 steps. Each snapshot is treated as a run and together, the deviation of the fine-tuned model logits from the clean (reference) model logits from these different runs give an estimate of empirical TPV using the same empirical TPV formula as above. We also track the Taylor approximation error.

**First-order validity check:** For every noisy model (label-noise run or SGD snapshot), we compute a relative finite-difference Taylor error to evaluate whether the model remains in a first-order regime around $w^\star$. Let $\Delta = w - w^\star$ and let $h = 10^{-2}$ be a finite-difference step. For a randomly-selected reference set of 128 training inputs $X_{\text{ref}}$, we estimate

$$\text{rel\_err} = \frac{\mathbb{E}_{x \in X_{\text{ref}}} \left[ (f_w(x) - f_{w^\star}(x)) - \frac{f_{w^\star + h\Delta}(x) - f_{w^\star}(x)}{h} \right]^2}{\mathbb{E}_{x \in X_{\text{ref}}} \left[ (f_w(x) - f_{w^\star}(x))^2 \right] + 10^{-12}},$$

where expectations are empirical averages over $X_{\text{ref}}$. We then discard the runs with values above the threshold $10^{-3}$.

**Total configuration count:** We consider two experiment groups: (a) varying $n_{\text{tr}}$ with fixed width, and (b) varying width with fixed $n_{\text{tr}}$. Taken together, the sweeps cover $3 \times 3 \times 2 \times 3 \times 2 = 108$ label-noise configurations and $3 \times 3 \times 2 \times 3 \times 4 = 216$ SGD-noise configurations, for a combined total of 324 distinct settings, each with up to 20 independent label-noise runs or $\approx 40$ SGD-noise snapshots. These yield the TPV scatter plots in Fig. 1 and Fig. 2.

### G.1.2. CIFAR EXPERIMENT

**1. Vary Width Experiment Details:**

We describe the details for the CIFAR-10 experiment in Fig 3 below. Details for CIFAR-100 (Fig. 13) are similar except the output has 100 dimensional logits and we use analogous CIFAR-100 pre-trained architectures.

**Dataset and preprocessing:** We use the standard CIFAR-10 per-channel normalization. From these, we randomly subsample $n_{\text{train}} = 10000$ and $n_{\text{test}} = 10000$.

**Reference models:** We use pre-trained mobilenetv2_x0_5, mobilenetv2_x0_75, mobilenetv2_x1_0, mobilenetv2_x1_4 from Pytorch Hub and denote its logits by $f_{w^\star}(x)$. These models have widths roughly $16, 24, 32, 48$ respectively.

**Label-noise perturbation (logit noise):** For the label-noise experiments, we retrain from the fixed initialization $w^\star$ for $R = 5$ independent runs. Each run injects additive i.i.d. Gaussian noise $\varepsilon \sim \mathcal{N}(0, \sigma^2)$ with $\sigma \in \{0.05, 0.1\}$ to the clean labels. Each model is then trained for 50 epochs using mini-batch SGD MSE regression on the noisy logits with momentum 0.9, learning rate $10^{-4}$, batch size 256, and 0 weight decay. Mini-batch shuffling is turned off and a randomness seed is used so that each run sees the same sequence of mini-batches in order to avoid randomness due to SGD and focus only on randomness due to label noise. Also, we train the models in Pytorch eval mode so that modules like batch norm do not make output logits batch dependent. Denote the fine-tuned model in the $r^{th}$ run as $f^{(r)}$. We record the train/test CE (for generalization gap) and empirical train/test TPV using all these runs.

Empirical TPV is computed as

$$\widehat{\text{TPV}}_{\text{train}} = \frac{1}{Rn_{\text{train}}} \sum_{r=1}^{R} \sum_{i=1}^{n_{\text{train}}} \left\| f^{(r)}(x_i) - f_{w^\star}(x_i) \right\|_2^2,$$

and analogously for the test set. We additionally record the cross-entropy loss on clean labels test set. See Appendix D.3.4 for details on the Label Noise TPV estimation algorithm.

**SGD-noise perturbation:** To simulate stationary SGD noise around a minimum, we initialize the model at $w^\star$ and run SGD on the CIFAR-10 classification task using cross-entropy loss for 10 epochs with momentum 0.9 and no weight decay, using learning rates $\{10^{-4}, 5 \times 10^{-5}\}$ and batch sizes $\{128, 256\}$. Snapshots are collected every epoch. Each snapshot is treated as a run and together, the deviation of the fine-tuned model logits from the clean (reference) model logits from these different runs give an estimate of empirical TPV using the same empirical TPV formula as above.

**2. Vary Number of Samples Experiment Details:**

For the experiment with varying number of training samples (Fig. 12), we use pretrained ResNet–20/32/44/56 models on CIFAR-10. Notice these architectures have different depth, which is not a consideration in the TPV theory, and is merely used as a source of variation in our experiments. The experimental details are similar to the varying width experiment above, except now we group experiments by a randomly selected training dataset subset of size $n_{\text{train}} \in \{1, 10, 10000\}$.

For CIFAR-100, analogous pre-trained models are used corresponding to each of the CIFAR-10 models. All pretrained models are taken from the GitHub repository `chenyaofo/pytorch-cifar-models`.

**Results summary:** Across both CIFAR-10 and CIFAR-100, we find the same pattern as in the synthetic experiments: (i) TPV stability holds regardless of the model's generalization gap; (ii) it holds across all tested widths, including the smallest; (iii) it breaks only when $n_{\text{train}}$ is very small ($n_{\text{train}} = 1$ lies outside the $50\%$ error band; $n_{\text{train}} = 10$ is mostly within it; $n_{\text{train}} = 10000$ is tight).

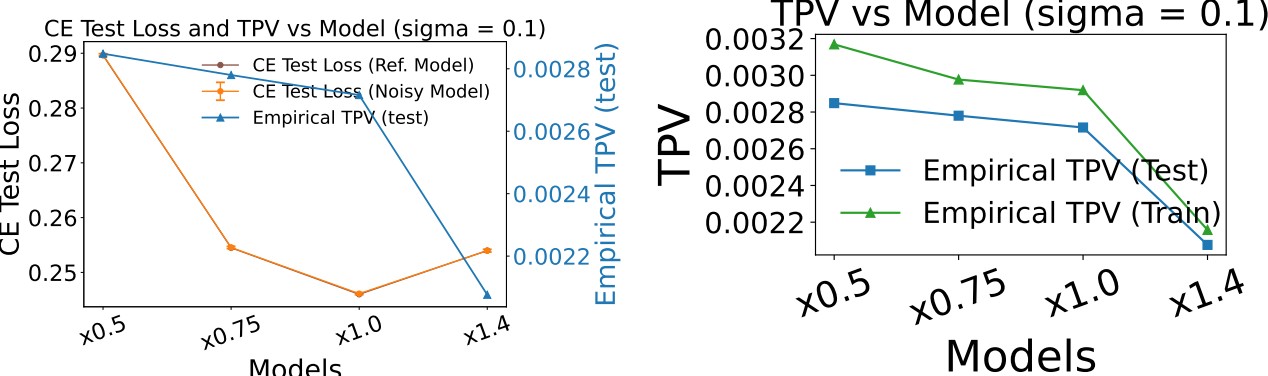

*Figure 15.* Empirical TPV estimates under target logit noise on CIFAR-10 for noise standard deviation $\sigma = 0.01$. Both TPV estimates reduce as width increases and correlate with the test set cross-entropy loss of the reference model.

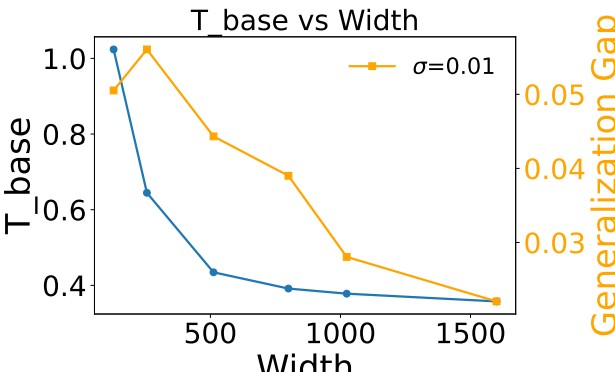

*Figure 16.* Generalization gap and $T_{\text{base}}$ vs. network width. As width increases, both quantities reduce.

## G.2. Details of Experiments in Section 5.2

The experiments in Section 5.2 have three empirical goals: (i) verify that empirical TPV tracks the theoretical base quantity $T_{\text{base}} = \sum_i B_{ii}/s_i^2$ and decreases with width; (ii) confirm TPV stability in this setting; (iii) confirm that lower TPV correlates with lower clean test loss. The following subsections provide full experimental details for the synthetic and CIFAR settings.

### G.2.1. SYNTHETIC DATA EXPERIMENTAL SETUP

We study a controlled synthetic regression problem designed to empirically test Theorem 4.2. Inputs $x \in \mathbb{R}^{20}$ are drawn i.i.d. from $\mathcal{N}(0, I)$, and targets are generated by a fixed teacher $y = x^\top w_{\text{true}}$ with $w_{\text{true}} \sim \mathcal{N}(0, I)$. We sample $n_{\text{train}} = 1000$ training points and $n_{\text{test}} = 5000$ test points. The learner is a three-layer ReLU MLP (input $\to$ width $\to$ width $\to$ 1), and we sweep over widths $\{128, 256, 512, 800, 1024, 1600\}$. For each width, we first train a "clean" reference network on the noiseless labels using full-batch SGD with momentum 0.9, fixed learning rate $5 \times 10^{-3}$, no weight decay, and 800 epochs. This gives a reference parameter $w^\star$ and corresponding reference outputs $f^\star(x)$ on both the training and test sets.

At $w^\star$, we compute the full Jacobian $J \in \mathbb{R}^{n_{\text{train}} \times P}$ via automatic differentiation (one row per input), perform an SVD $J = USV^\top$, and estimate the test-distribution Hessian surrogate $H_{\text{eff}}$ by sampling test inputs and computing $G_{ii} = \mathbb{E}_x[(g(x)^\top v_i)^2]$, where $g(x)$ is the gradient of the network output and $v_i$ are the right singular vectors of $J$. The theoretical base quantity is then $T_{\text{base}} = \sum_i G_{ii}/s_i^2$, so that Theorem 4.2 predicts TPV $\approx \sigma^2 T_{\text{base}}$ for label-noise variance $\sigma^2$.

To estimate empirical TPV, for each pair (width, $\sigma$) with $\sigma \in \{0.01, 0.05, 0.1, 0.2\}$, we run 50 independent Monte Carlo trials. In each trial we add i.i.d. noise $\epsilon \sim \mathcal{N}(0, \sigma^2)$ to the training labels, re-initialize the model at $w^\star$, and retrain using identical optimization settings for 500 epochs, and no proximity penalty. Empirical TPV on train and test sets is computed as the variance across runs of the predictions relative to $f^\star$.

### G.2.2. CIFAR EXPERIMENTAL SETUP

We evaluate empirical TPV under injected Gaussian noise on the *logits* of a clean reference model on CIFAR-10/100. We describe the details for CIFAR-10 below. Details for CIFAR-100 are similar except the output has 100 dimensional logits.

**Dataset and preprocessing:** We use the standard CIFAR-10 per-channel normalization. From these, we randomly subsample $n_{\text{train}} = 4000$ and $n_{\text{test}} = 4000$. We find that it is important to use a sufficiently large samples size for consistent estimate.

**Reference models:** For each architecture in `cifar10_mobilenetv2_x0_5`, `cifar10_mobilenetv2_x0_75`, `cifar10_mobilenetv2_x1_0`, and `cifar10_mobilenetv2_x1_4`, we load a pretrained clean model from Pytorch Hub and denote its logits by $f_{w^\star}(x)$. For every architecture, we compute the baseline cross-entropy losses on the clean labels for both train/test sets.

**Label-noise perturbation (logit noise):** For each model, we use Gaussian noise level $\sigma = 0.1$, and we run $R = 20$ Monte Carlo replicates. For each replicate, we sample noise

$$\varepsilon_i \sim \mathcal{N}(0, \, \sigma^2 I_{10}),$$

and construct the noisy regression target using the randomly selected $n_{\text{train}}$ samples in the training set as

$$y_i^{\text{noisy}} = f_{w^\star}(x_i) + \varepsilon_i \qquad (1 \le i \le n_{\text{train}}).$$

Each replicate begins by re-initializing the model to the clean reference weights $w^\star$. We then fine-tune this model for 10 epochs using mini-batch SGD MSE regression on the noisy logits with momentum 0.9, learning rate $10^{-4}$, batch size 256, and 0 weight decay. Mini-batch shuffling is turned off and a random seed is used so that each run sees the same sequence of mini-batches in order to avoid randomness due to SGD and focus only on randomness due to label noise. Also, we train the models in Pytorch eval mode so that modules like batch norm do not make output logits batch dependent. Denote the fine-tuned model in the $r^{th}$ run as $f^{(r)}$.

After training each noisy model, we record its logits $f^{(r)}(x)$ on both train and test subsets. Empirical TPV is computed as

$$\widehat{\text{TPV}}_{\text{train}} = \frac{1}{R n_{\text{train}}} \sum_{r=1}^{R} \sum_{i=1}^{n_{\text{train}}} \left\| f^{(r)}(x_i) - f_{w^\star}(x_i) \right\|_2^2,$$

and analogously for the test set. We additionally record the cross-entropy loss on clean labels test set. See Appendix D.3.4 for details on the Label Noise TPV estimation algorithm.

### G.3. Details of Experiments in Section 5.3 (Label Noise TPV and Generalization)

Section 5.3 showed the TPV–test-loss relationship under a single regularizer (label smoothing) on an MLP architecture swept across parameter counts. Here we concretely discuss this empirical relationship, document the experimental setup in detail and report the analogous results for dropout and weight decay.

### G.3.1. RELATIONSHIP BETWEEN LABEL NOISE TPV AND TEST LOSS

Recall the local bias-variance decomposition Equation 3,

$$\mathcal{E}_{\text{test}} = \text{TPV} + \mathbb{E}_x \big[ \underbrace{\big( f_{w^\star}(x) - f^\star(x) \big)^2}_{\text{bias}^2} \big]. \tag{75}$$

This equation decomposes the expected test error into a variance component (TPV) and a bias$^2$ component. The quantity we correlate TPV against in our experiments is the clean test loss at $w^\star$, which approximates bias$^2(w^\star)$. The relationship between TPV and clean test loss is therefore an *empirical* question about how the two components in the R.H.S. of the above equation co-vary across configurations.

We find that the correlation between the two terms exhibit opposite relation depending on the *training loss regime* of the model being considered. Specifically, consider two regimes– low training loss regime and high training loss regime. These regimes can be operationalized, for instance, by using a pre-fixed loss threshold. High training loss regime corresponds

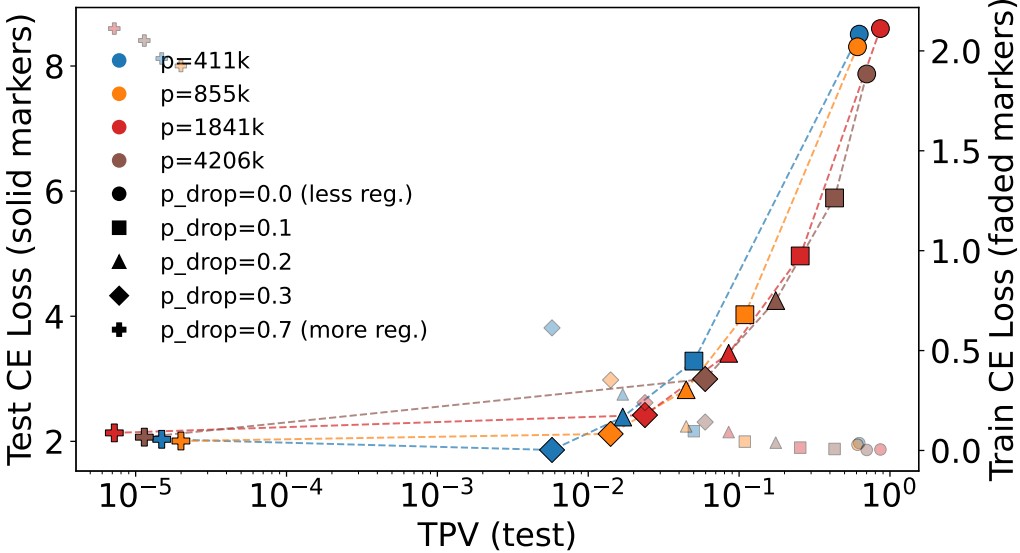

*Figure 17.* **Label Noise TPV vs. test loss on CIFAR-10 for dropout.** Same plotting convention as Fig. 7.

to underfitting in the presence of effective model-capacity bottlenecks (e.g. small model size, high regularization levels, poor/insufficient optimization, etc.). Low training loss regime on the other hand is more typical in modern deep learning with overparameterized networks and can exhibit a wide range of generalization gap.

Our experiments reveal a consistent pattern: in the low training loss regime, smaller TPV (via regularization, width, or other knobs) is accompanied by a reduction in test loss ($bias^2$); so TPV and test loss are positively correlated. In the high training loss regime, regularization that lowers TPV simultaneously raises test loss ($bias^2$), inverting the correlation due to underfitting.

### G.3.2. DETAILS FOR THE CIFAR-10 MLP EXPERIMENT

Now we describe the details behind the experiment in Section 5.3.

**Architecture and data:** We use a 2-hidden-layer ReLU MLP with hidden width $h \in \{128, 256, 512, 1024\}$, giving parameter counts $p \in \{\approx 201k, 411k, 855k, 1.8M, 4.2M\}$. The input is a flattened $3 \times 32 \times 32$ CIFAR-10 image with standard mean/std normalization. We use a fixed random subset of $10,000$ training images and $4000$ test images.

**Regularization grids:**

- Weight decay: $\{0, 10^{-5}, 10^{-4}, 10^{-3}, 10^{-2}\}$.

- Dropout (applied after every hidden layer): $\{0, 0.1, 0.2, 0.3, 0.7\}$.

- Label smoothing: $\{0, 0.05, 0.1, 0.2, 0.7, 0.9\}$.

Each architecture is trained once at each regularization level.

**Reference-model training:** The reference model $w^\star$ is trained for 300 epochs of SGD with learning rate 0.05, Nesterov momentum 0.9, step-annealed learning rate multiplying learning rate by 0.1 every 1/3rd of the total training epochs, and batch size 256. Weight decay is passed through SGD's weight-decay parameter; dropout is applied in the architecture; label smoothing is applied in the cross-entropy loss. After training, test and train CE are evaluated with a plain cross-entropy criterion (no smoothing) for comparability.

**TPV estimation:** We use the Label Noise TPV estimation algorithm described in §D.3.4. Given $w^\star$, we estimate TPV under additive Gaussian noise on the teacher logits. For each of $R = 20$ independent runs, we add i.i.d. Gaussian noise with

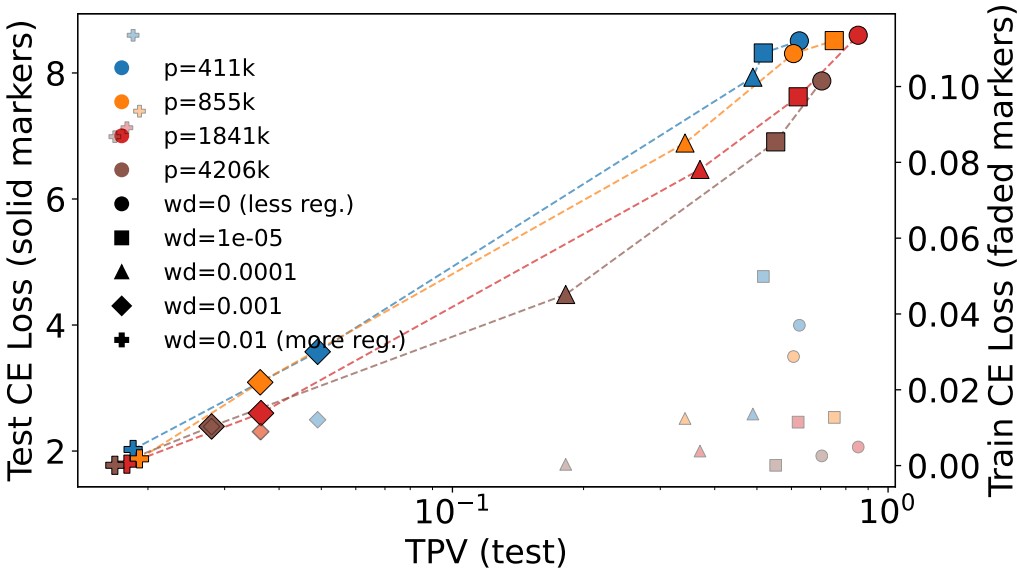

*Figure 18.* **Label Noise TPV vs. test loss on CIFAR-10 for weight decay.** Same plotting convention as Fig. 7.

$\sigma = c\,\sigma_0$ ($\sigma_0 = 0.1$ and $c$ set as described in §D.3.4) to the logits of $w^\star$ on the training inputs, reload $w^\star$, and fine-tune the model with MSE regression against the noisy target logits for 20 epochs using SGD with lr $10^{-4}$, momentum 0.9, batch size 256, and no weight decay. The noisy fine-tuning is conducted in `eval` mode (dropout disabled) and with a fixed data-loader seed across runs to isolate the effect of label noise from other sources of randomness. Proximity regularization is not used ($\lambda_{\mathrm{prox}} = 0$).

**Results:** Figures 17 and 18 respectively show the TPV-vs-test-CE plots for dropout and weight decay, analogous to label smoothing (Fig. 7) in the main text. Dropout and label smoothing regularizers exhibit U-shaped relationship: test loss decreases with TPV monotonically with increasing regularization strength in the low training loss regime, and then test loss increases as TPV reduces further in the high training loss regime (underfitting). Within the low training loss regime, ordering models by TPV recovers their ordering by test CE within each architecture. The weight decay plot on the other hand contains all points in the low training loss regime and therefore shows a consistent positive correlation between test loss and TPV.

### G.3.3. TRAINING-SET TPV ALONG THE TRAINING TRAJECTORY (RESNET-18 CIFAR-100 EXPERIMENT)

This appendix documents the experimental setup behind Figure 8 in Section 5.3.

**Setup:** We train a CIFAR-adapted ResNet-18 on CIFAR-100 with 30% symmetric label noise: a fixed RNG seed is used to select 30% of training samples whose labels are reassigned uniformly at random to one of the remaining 99 classes. The architecture is a standard CIFAR ResNet-18 with a $3\times3$ stem (no max-pool), four stages of two residual blocks each at widths $\{64, 128, 256, 512\}$, BatchNorm, and a final linear classifier; total parameter count is $\approx 11.2$M. Training uses SGD with Nesterov momentum 0.9, weight decay $5\times10^{-4}$, batch size 128, initial learning rate 0.1, and a cosine annealing schedule over 200 epochs. Cross-entropy loss with label smoothing $\varepsilon_{\mathrm{LS}} = 0.1$ is used as the training objective. The training stream applies standard CIFAR data augmentation (random crop with 4-pixel padding, random horizontal flip) followed by channel-wise normalization with CIFAR-100 statistics.

**Accuracy evaluation:** Training accuracy is measured against the noisy labels the model is being optimized against, on the augmented training stream in `model.eval()` mode (BatchNorm running statistics fixed). Validation accuracy is measured on the standard CIFAR-100 test split (10,000 images, normalization only, no augmentation).

**TPV estimation:** Training-set TPV is estimated every 5 epochs (and at epoch 0). The TPV subset consists of 10,000 training images randomly sampled once from the un-augmented training set; only channel-wise normalization is applied to

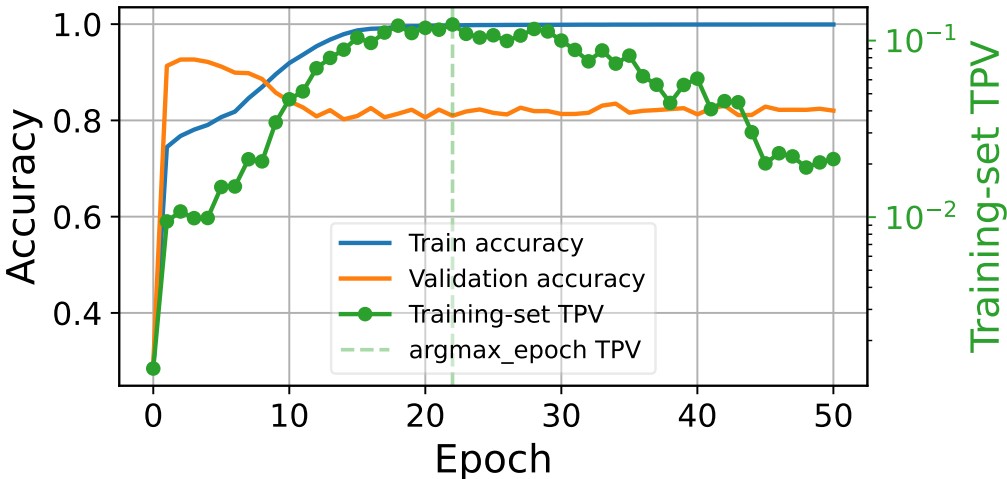

*Figure 19.* **TPV trajectory (BERT-small on AG News dataset with** $20\%$ **label noise).** Same plotting convection as Fig. 8.

these images, so that no augmentation randomness enters the TPV estimate. For each of $R = 5$ independent runs, we add i.i.d. Gaussian noise with $\sigma = c\,\sigma_0$ ($\sigma_0 = 0.1$ and $c$ set as described in §D.3.4) to the logits of $w^\star$ on the training inputs, reload $w^\star$, and fine-tune the model with MSE regression against the noisy target logits for 3 epochs using SGD with lr $10^{-4}$, momentum 0.9, batch size 256, and no weight decay. The noisy fine-tuning is conducted in `eval` mode (dropout disabled) and with a fixed data-loader seed across runs to isolate the effect of label noise from other sources of randomness. Proximity regularization is not used ($\lambda_{\text{prox}} = 0$).

**Argmax-TPV landmark:** The vertical dashed line in Figure 8 is placed at $\arg\max_e \widehat{\text{TPV}}_{\text{train}}(e)$ over all TPV checkpoints $e$. This is computed purely from the trajectory of training-set TPV estimates—no validation labels or test loss are used to identify this epoch.

**Observations:** Two regimes are visible in Figure 8. First, the high-training-loss regime (epoch $\lesssim 125$) shows training accuracy slowly climbing toward $100\%$ on the (noisy) supervision while TPV gradually rises; this corresponds to the underfitting branch of the U-shape in Figure 7, where the model has not yet interpolated. Second, the low-training-loss regime (epoch $\gtrsim 125$) is reached once training accuracy saturates; TPV peaks and then decreases monotonically while validation accuracy rises in step. The argmax-TPV landmark approximately identifies the transition between the first and second regimes, recovering—along the time axis of a single training run—the same TPV-versus-test-error relationship that Figure 7 exhibits across a sweep of regularization strengths at convergence.

### G.3.4. TRAINING-SET TPV ALONG THE TRAINING TRAJECTORY (ADDITIONAL EXPERIMENTS ON NLU TASKS USING BERT)

**Setup:** We fine-tune BERT-Small (`prajjwal1/bert-small`; 4 transformer layers, hidden size 512, $\approx$29M parameters) on two text-classification benchmarks with $20\%$ symmetric label noise: a fixed RNG seed is used to select $20\%$ of training examples whose labels are reassigned uniformly at random to one of the remaining classes. The tasks differ only in their label spaces and dataset sizes: AG News is a 4-class topic classification task with $\approx$120,000 training examples and 7,600 test examples, while TREC is a 6-class coarse question-type classification task with $\approx$5,452 training examples and 500 test examples. In both cases a linear classifier is placed on the `[CLS]` representation, and inputs are truncated/padded to 128 tokens. Training uses AdamW with the standard BERT parameter-group convention (weight decay 0.01 for non-bias, non-LayerNorm parameters; 0 otherwise), learning rate $10^{-4}$, batch size 32, and a linear warmup over $10\%$ of total steps followed by linear decay to zero over 50 epochs. Cross-entropy loss is used as the training objective.

**Accuracy evaluation:** Training accuracy is measured against the noisy labels the model is optimized against. Validation accuracy is measured on the respective test splits (7,600 examples for AG News; 500 for TREC), with no augmentation.

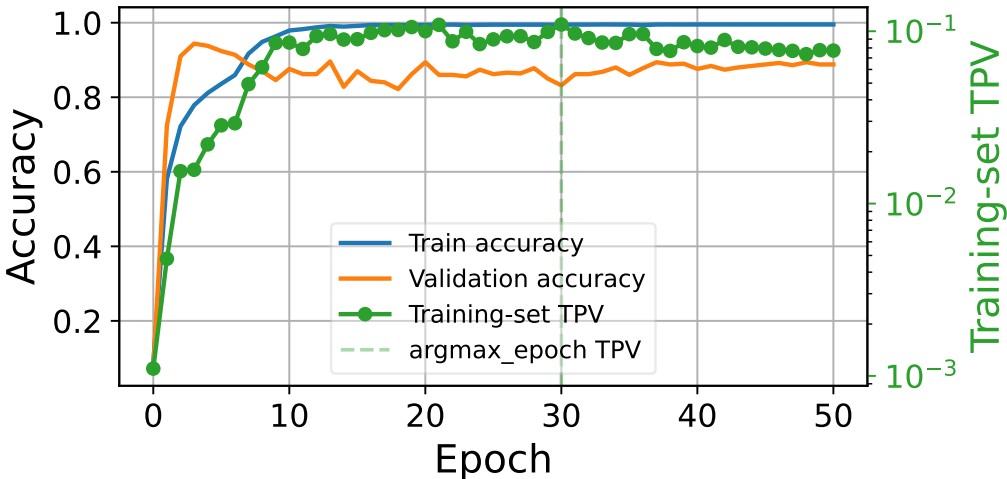

*Figure 20.* **TPV trajectory (BERT-small on TREC dataset with** $20\%$ **label noise).** Same plotting convection as Fig. 8.

**TPV estimation:** Training-set TPV is estimated at every epoch (and at epoch $0$). The TPV subset consists of 3,000 training examples randomly sampled once from the training set and pre-tokenized; the same fixed subset is reused at every checkpoint so that no sampling randomness enters the TPV trajectory. For each of $R = 3$ independent runs, i.i.d. Gaussian noise with $\sigma = c\,\sigma_0$ ($\sigma_0 = 0.1$, $c = \text{mean}(|f^\star|)$ over the TPV subset, as described in §D.3.4) is added to the reference logits of $w^\star$; the model is then reloaded from $w^\star$ and fine-tuned with MSE regression against the noisy target logits for 3 epochs using AdamW (lr $= 10^{-5}$, no weight decay, batch size $64$). The noisy fine-tuning is conducted in `eval()` mode so that dropout is disabled. A fixed data-loader seed is used across runs to isolate logit noise as the sole source of run-to-run variation. Proximity regularization is not used ($\lambda_{\text{prox}} = 0$).

**Argmax-TPV landmark:** A vertical dashed line is placed at $\arg\max_e \widehat{\text{TPV}}_{\text{train}}(e)$ over all per-epoch TPV estimates. This is computed purely from the training-set TPV trajectory—no validation labels or test loss are consulted.

**Observations:** The results are shown in Fig. 19 for the AG News dataset and in Fig. 20 for the TREC dataset. Both datasets exhibit the two-regime pattern visible in the CIFAR-100 experiment, now along the fine-tuning time axis of a single BERT-Small run. In the early regime (epoch $\lesssim 22$ for AG News; epoch $\lesssim 30$ for TREC), training accuracy on the noisy labels is still rising and the model has not yet interpolated the training set; TPV grows steadily throughout this phase. In the late regime, training accuracy saturates at $100\%$ on the noisy supervision while validation accuracy plateaus and TPV peaks then declines. The argmax-TPV landmark identifies the transition between these two regimes in both cases, using only the training-set TPV trajectory and no validation labels.

# H. Pruning (Application)

## H.1. Jacobian-Based Rebalancing (JBR) for Pruning

We adopt the viewpoint that pruning should preserve the model's *predicted class* on the correctly classified training samples. For a classifier with logits $f(x; w) \in \mathbb{R}^K$, the full logit vector is irrelevant for this purpose; prediction changes only if the probability of the currently predicted class drops relative to the others. Thus, instead of treating $f$ itself as the task in the TPV framework, we work with the scalar functional

$$u(x; w) := -\log(p_{c(x)}(x; w)),$$

where $c(x) := \arg\max_k p_k(x; w^\star)$ and $p = \text{softmax}(f)$ and $w^\star$ is the reference (unpruned) network. This quantity uses the probability assigned to the class the model originally predicted, and pruning should leave this value stable. Specifically, we only want to include training samples for which the model predicts correctly. As a proxy, we select samples for which $p_{c(x)}(x; w) > \tau$, where $\tau$ is a probability threshold (we use $\tau = 0.9$). We caution that the TPV Trace Stability theorem

only applies to logits ($f(.)$) and does not directly apply to functionals like $u(.)$ considered above. We leave that analysis for future work. The focus here is to formulate pruning score as a TPV object.

**Pruning as a Source of Parameter Noise**   Once training has converged and we are interested purely in pruning, the dominant source of parameter perturbation is the pruning operation itself. In many practical settings we prune entire groups at once (e.g., channels). Define $g \subseteq \{1, \ldots, p\}$ as a group with parameter vector $w_g \in \mathbb{R}^{p_g}$, and for a given group $g$ we either prune all its parameters or keep them all. We model pruning as a structured parameter perturbation that acts coherently on each parameter group. For sensitivity analysis within the TPV framework, we introduce a zero-mean, group-aligned perturbation:

$$\delta w_g = \sigma \, \xi_g \, w_g, \qquad g \in \mathcal{G}, \tag{76}$$

where $\sigma > 0$ is a small scalar controlling the perturbation scale and $\xi_g$ is a Rademacher random variable taking values $\pm 1$ with equal probability. Thus $\mathbb{E}[\delta w_g] = 0$ and the group-wise pruning covariance is

$$C_{\text{prune},g} := \mathbb{E}[\delta w_g \delta w_g^\top] = \sigma^2 \, w_g w_g^\top. \tag{77}$$

Treating masks as being independent across groups, the full pruning covariance $C_{\text{prune}}$ is block-diagonal with blocks $\{C_{\text{prune},g}\}_{g \in \mathcal{G}}$. Using the cyclic property of the trace, the contribution of group $g$ to the TPV object is

$$\text{Tr}\big(H_{\text{eff},g} \, C_{\text{prune},g}\big) = \sigma^2 . \mathbb{E}[w_g^\top J_{u,g}^T J_{u,g} w_g]. \tag{78}$$

where $J_{u,g}(x; w) := \partial u(x; w)/\partial w_g \in \mathbb{R}^{1 \times p_g}$. Thus, to keep the overall TPV low for a pruned network, we want to set $\sigma^2 = 0$ for groups with large $w_g^\top J_{u,g}^T J_{u,g} w_g$ and only prune groups (equivalently $\sigma^2 > 0$) with small $w_g^\top J_{u,g}^T J_{u,g} w_g$. Thus we define the JBR importance of a parameter group as

$$\boxed{\text{score}_{\text{JBR}}(w_g) := \mathbb{E}_x[w_g^\top J_g^T \delta_u^\top \delta_u J_g w_g].} \tag{79}$$

where $J_g := \partial f(x; w)/\partial w_g \in \mathbb{R}^{K \times p_g}$, and $\delta_u := \partial u(x; w)/\partial f(x; w) \in \mathbb{R}^K$.

**Connection and Contrast with JC**   The proposed JBR is very similar to Jacobian Criterion (JC) proposed by (Chen et al., 2025) and can be seen as a label-free version of JC. Both JBR and JC assign a score to each parameter group $g$ of the form

$$\text{score}(w_g) = \mathbb{E}_x\big[w_g^\top J_g(x)^\top m(x) m(x)^\top J_g(x) w_g\big]$$
$$= \mathbb{E}_x\big[(m(x)^\top v_g(x))^2\big],$$

where $v_g(x) = J_g(x) w_g$ is the logit-space direction induced by group $g$, and the only difference between the two methods lies in the choice of the logit–space vector $m(x)$:

$$m_{\text{JC}}(x) := \delta_L(x) = p(x) - y(x),$$
$$m_{\text{JBR}}(x) := \delta_u(x) = p(x) - e_{c(x)}.$$

To understand the relationship between JC and JBR clearly, consider the clean setting in which all the labels in the training data are correct. Now, if the trained (unpruned) model predicts all the training samples correctly, then $y(x) = e_{c(x)}$, and JBR and JC importance scores become identical. The scores differ when either: i) the labels used in JC have noise; or, ii) the model predicted class labels used in JBR are incorrect.

## H.2. Related Work on Pruning

Pruning aims at reducing model size and improving inference speed (LeCun et al., 1989; Hassibi et al., 1993). These methods can be broadly categorized into unstructured and structured pruning.

Unstructured pruning removes each parameter individually (Han et al., 2015; Wang et al., 2020; Frankle & Carbin, 2018; Paul et al., 2022). While this approach aligns with the goal of reducing model size, it often does not significantly boost inference speed. Structured pruning (Molchanov et al., 2019; Liu et al., 2021), on the other hand, removes entire neurons, filters, or attention heads, resulting in models that are easier to accelerate on standard hardware.

Structured pruning can be further divided into data-dependent and data-independent pruning. Data-independent strategies typically rely on pretrained weight statistics; removing groups with small parameter norm (Li et al., 2017), BatchNorm scale

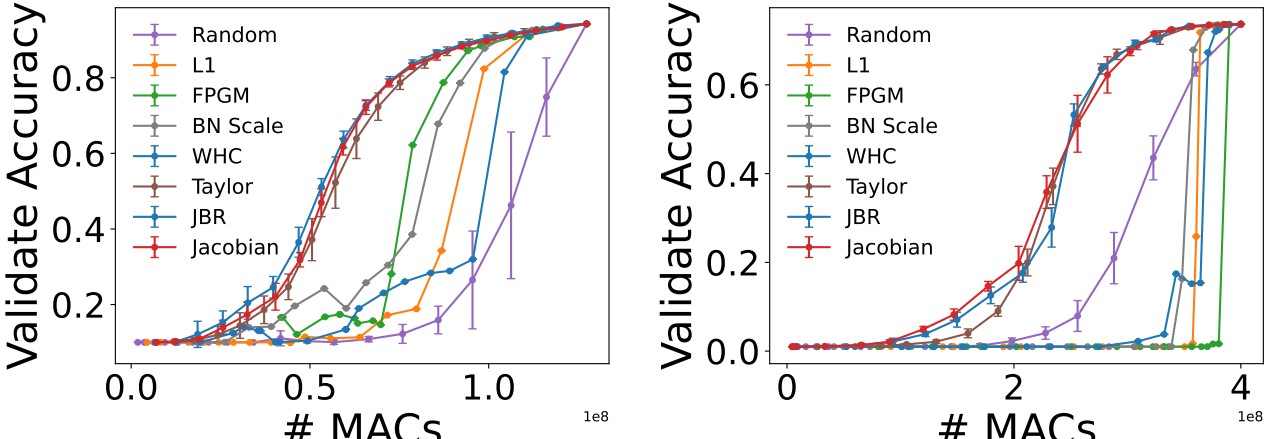

*Figure 21.* Pruning results of various criteria on Cifar-10 with ResNet-56 (left) and Cifar-100 with VGG-19 (right). JBR matches or outperforms existing methods.

(Liu et al., 2017), or geometric and structural properties of the pretrained weights—such as norms, redundancy, clustering, or subspace contribution—to estimate filter importance without relying on labels or loss gradients (He et al., 2019; Singh et al., 2020; Chen et al., 2023). Data-dependent strategies on the other hand prune parameter groups based on the sensitivity of loss w.r.t. neurons or parameters. The sensitivity can be estimated in different ways– second order approximation (Liu et al., 2021), Fisher approximation (Theis et al., 2018), and first order approximation (You et al., 2019; Molchanov et al., 2016; 2019; Chen et al., 2025).

The proposed JBR pruning strategy is particularly close to Jacobian Criterion (JC) (Chen et al., 2025). The key difference is that JC measures loss sensitivity w.r.t. parameter groups using ground-truth labels under the first order approximation, while JBR measures loss sensitivity using the model predicted class labels for confident samples. Under the specific scenario where ground-truth labels are fully correct and model predictions are fully accurate on the training samples, the two become equivalent. The main motivation behind JBR is that it models pruning as a special case of parameter perturbation noise under the general TPV framework.

### H.3. Pruning Experiments

We evaluate whether the TPV-motivated pruning criterion (JBR) improves accuracy–compression tradeoffs relative to standard groupwise criteria. Following the OBC pruning protocol, we perform global channel pruning with iterative removal and recomputation of importance scores. We compare JBR against Jacobian, L1-norm, BatchNorm-scale, FPGM, WHC, Taylor, and Random. We prune two ImageNet models (ResNet-50, MobileNet-V2) at $50\%$ global sparsity and two CIFAR models (ResNet-56 on CIFAR-10, VGG-19-BN on CIFAR-100) at $90\%$ sparsity. Each model is pruned iteratively for 18 steps, averaging results over 5 independent runs. Results are shown in Fig. 21 and Fig. 22 in terms of MACs. Across all architectures, the JBR criterion consistently matches or exceeds the performance of all baselines. See Appendix H.4 for details.

### H.4. Pruning Experiment Details

This appendix provides the experimental details for the pruning results reported in Section H.3. All pruning experiments follow the global structured pruning protocol introduced in *Optimal Brain Compression (OBC)* (Chen et al., 2025). Our implementation is based on the official GitHub code of (Chen et al., 2025) and extends the OBC framework by adding the TPV-motivated JBR importance criteria.

#### H.4.1. PRUNING FRAMEWORK

We adopt the global channel-pruning pipeline of Chen et al. (2025). At each pruning iteration, an importance score is computed for every pruning group (typically convolutional or linear output channels, together with the corresponding input-channel dependencies). The least-important groups are removed globally, and the dependency graph ensures architectural

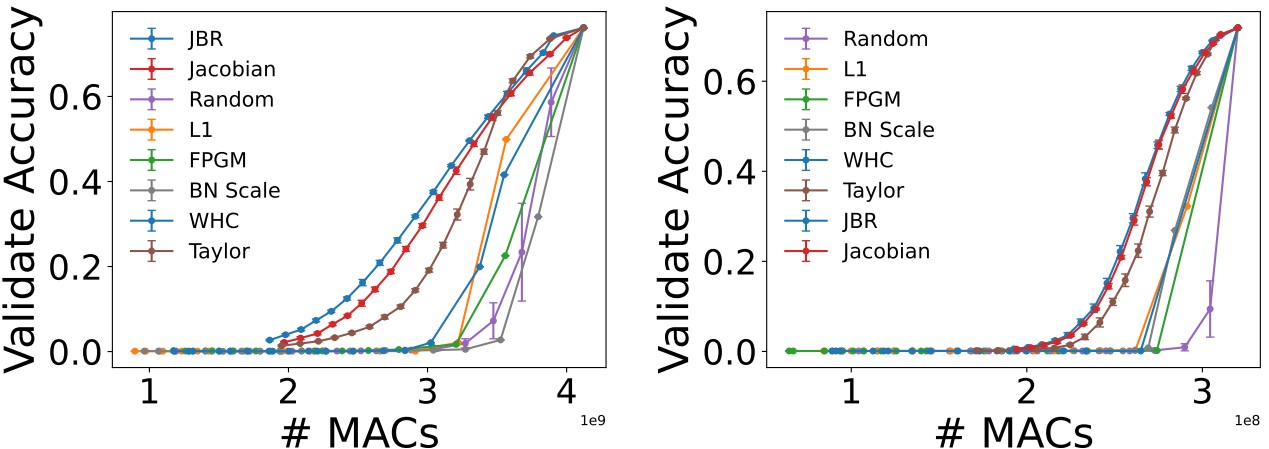

*Figure 22.* Pruning results of various criteria on ImageNet dataset using ResNet-50 (left) and MobileNet-v2 (right) without fine-tuning. JBR matches or outperforms existing methods.

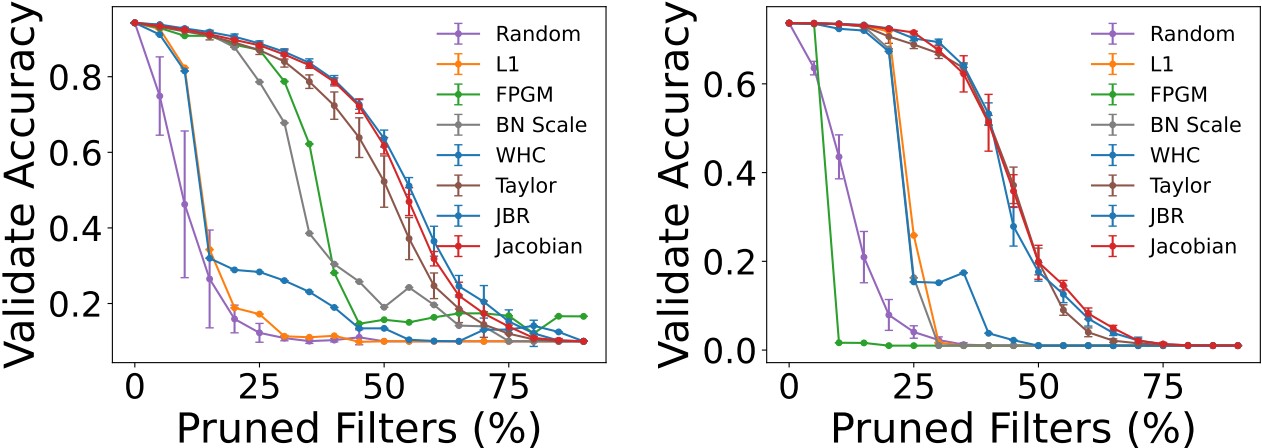

*Figure 23.* Pruning results of various criteria on Cifar-10 with ResNet-56 (left) and Cifar-100 with VGG-19 (right). JBR matches or outperforms existing methods.

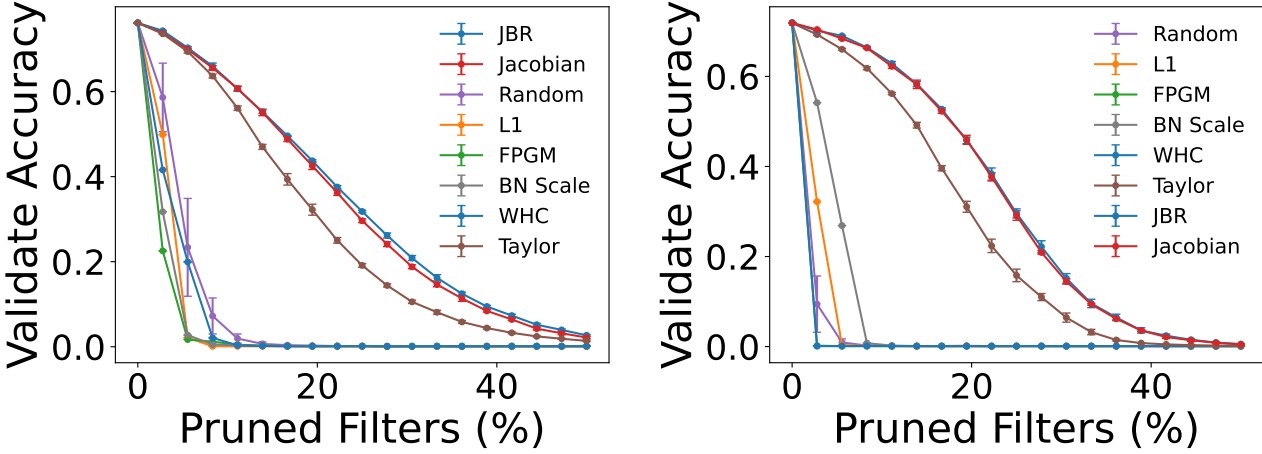

*Figure 24.* Pruning results of various criteria on ImageNet dataset using ResNet-50 (left) and MobileNet-v2 (right) without fine-tuning. JBR matches or outperforms existing methods.

consistency. No fine-tuning is performed between pruning iterations.

All pruning criteria evaluated in this work share the same dependency framework, pruning granularity, and iteration schedule. Thus, differences in accuracy arise solely from differences in importance scoring.

Fig. 23 and Fig. 24 show the same experiments as the ones in the main text, except the x-axis is now the percentage of pruned parameters.

### H.4.2. MODELS AND DATASETS

We evaluate four standard classification settings:

- **ImageNet-1k**:
    - ResNet-50, pruned to $50\%$ global channel sparsity.
    - MobileNet-V2, pruned to $50\%$ global channel sparsity.

- **CIFAR-10**:
    - ResNet-56, pruned to $90\%$ global sparsity.

- **CIFAR-100**:
    - VGG19-BN, pruned to $90\%$ global sparsity.

### H.4.3. PRUNING CRITERIA COMPARED

We compare the TPV-based **JBR** criterion with several established structured pruning criteria:

- **Jacobian** (Chen et al., 2025),

- **L1** weight-norm (Li et al., 2017),

- **Random** pruning,

- **BatchNorm scale** (Liu et al., 2017),

- **FPGM** (He et al., 2019) ,

- **WHC** (Chen et al., 2023),

- **Taylor** first-order saliency (Molchanov et al., 2019).

### H.4.4. ITERATIVE PRUNING SCHEDULE

All pruning experiments use the following schedule:

- **Global pruning**: channel groups ranked and removed across the whole network.

- **Number of iterations**: 18.

- **Sparsity targets**:
    - 0.5 for ImageNet models,
    - 0.9 for CIFAR models.

- **Score recomputation**: importance scores recalculated at every iteration.

- **Data usage**:
    - 50 minibatches with batch size 256 for ImageNet scoring,
    - 50 minibatches with batch size 128 CIFAR datasets' scoring.

All reported scores are averaged over 5 independent pruning runs.

# I. Applications

We present the details behind the several model/recipe selection experiment in §6.2. We note that TPV based model selection is much more robust in the within-architecture settings compared to cross-architecture settings, and in general holds in the low training loss regime. Also importantly, training set TPV based model selection relies on TPV stability (training set TPV $\approx$ test-set TPV).

## I.1. In-distribution training recipe selection: Joint-Regularizer Configurations on CIFAR-10

This section documents the experimental setup behind Fig. 9 in Section 6.2.

**Setup:** We use a CIFAR-10 subset of 10,000 training and 4,000 test images (drawn with a fixed RNG seed), normalized with dataset statistics. The architecture family is a CIFAR-adapted ResNet-18 with controllable base channel width $C$; the four width variants have $C \in \{8, 16, 24, 32\}$, yielding parameter counts of approximately 176k, 701k, 1,575k, and 2,797k respectively. Each residual stage uses two $3 \times 3$ convolutional blocks with batch normalization; spatial resolution is halved at stages 2–4 via strided convolutions, and a CIFAR-style stem (no max-pooling) preserves the $32 \times 32$ input resolution at stage 1.

Reference models are trained for 300 epochs with SGD (Nesterov momentum 0.9, initial learning rate $5 \times 10^{-2}$, step-decay by factor 0.1 every 100 epochs, batch size 256). All three regularizers—weight decay, dropout (Dropout2d inside each residual block), and label smoothing—are applied simultaneously according to the sampled configuration. Test and train CE losses are always evaluated without label smoothing for comparability across configurations.

TPV is estimated via noisy-logit fine-tuning with $R = 20$ independent noise draws, additive Gaussian noise with standard deviation $\sigma = c\,\sigma_0$, where $\sigma_0 = 0.1$ and $c$ is the mean absolute logit value of the reference model (adaptive noise scaling), and 20 fine-tuning epochs at learning rate $10^{-4}$. See Appendix D.3.4 for details on the Label Noise TPV estimation algorithm.

**Candidate grids:** Rather than sweeping a single regularizer, we sample five *joint* configurations, each specifying a triple (weight decay, dropout, label smoothing) drawn independently from fixed candidate grids:

- Weight decay: $\{0,\ 10^{-5},\ 5 \cdot 10^{-4}\}$.

- Dropout: $\{0,\ 0.1,\ 0.2,\ 0.3,\ 0.5\}$.

- Label smoothing: $\{0,\ 0.05,\ 0.1,\ 0.2,\ 0.3\}$.

Zero is included in every grid so that "no regularization on this axis" is a possible outcome of sampling.

We draw $N = 5$ joint configurations using a fixed RNG seed (seed 0 in the reported run). For each configuration $c \in \{0, \ldots, 4\}$, one value is independently sampled from each of the three grids. The same five configurations are used for every width variant, so each (width, configuration) pair defines one reference model. The sampled configurations used in Fig. 9 are:

- C0: $w_{\mathrm{WD}} = 5 \cdot 10^{-4}, p_{\mathrm{drop}} = 0.3, \epsilon_{\mathrm{LS}} = 0.1$  (moderate multi-axis regularization)

- C1: $w_{\mathrm{WD}} = 0, \qquad p_{\mathrm{drop}} = 0.1, \epsilon_{\mathrm{LS}} = 0$  (very light regularization)

- C2: $w_{\mathrm{WD}} = 0, \qquad p_{\mathrm{drop}} = 0, \quad \epsilon_{\mathrm{LS}} = 0$  (no regularization)

- C3: $w_{\mathrm{WD}} = 5 \cdot 10^{-4}, p_{\mathrm{drop}} = 0.3, \epsilon_{\mathrm{LS}} = 0.3$  (strong multi-axis regularization)

- C4: $w_{\mathrm{WD}} = 10^{-5}, \qquad p_{\mathrm{drop}} = 0.3, \epsilon_{\mathrm{LS}} = 0.3$  (weak WD, strong dropout and label smoothing)

No single regularizer orders these five configurations consistently: by weight decay the ranking is C1=C2<C4<C0=C3, whereas by label smoothing it is C1=C2<C0<C3=C4.

**Observation:** Across all four width variants, configuration C0 achieves the lowest test CE loss, and TPV correctly identifies it as the preferred configuration regardless of model size. Configurations C1 and C2 (little or no regularization) achieve low training loss but substantially higher test loss, and they receive the largest TPV values—consistent with overfitting increasing sensitivity to parameter perturbations.

## I.2. In-distribution Cross-architecture model selection: ImageNet pretrained models

This section documents the experimental setup behind Fig. 10 in Section 6.2.

**Setup:** We evaluate TPV as a predictor of top-1 validation accuracy across seven diverse pretrained ImageNet architectures: ResNet-18, ResNet-50, Wide ResNet-50-2, ShuffleNet V2 ($\times 1.0$), EfficientNet-B0, MNASNet 1.0, and ConvNeXt-Tiny. All models are loaded with standard `torchvision` pretrained weights (trained on the full ImageNet-1k training set) and evaluated without any further fine-tuning.

Data is drawn from ImageNet using a fixed RNG seed (seed 0): 10,000 training images and 10,000 validation images are sampled without replacement from the respective splits. Images are preprocessed with standard normalization only (no data augmentation), using batch size 512 for teacher logit computation and batch size 64 for noisy fine-tuning and evaluation.

**TPV estimation:** TPV is estimated via noisy-logit fine-tuning with $R = 5$ independent noise draws, additive Gaussian noise with standard deviation $\sigma = c\,\sigma_0$, where $\sigma_0 = 0.1$ and $c$ is the mean absolute logit value of the reference model (adaptive noise scaling), and using AdamW for 5 fine-tuning epochs at learning rate $10^{-8}$, no proximity regularization and no weight decay. The DataLoader shuffle order is held fixed across runs (using fixed seed) so that only the logit noise varies between runs. See Appendix D.3.4 for details on the Label Noise TPV estimation algorithm.

Fig. 10 reports train and validation set TPV against the baseline top-1 validation accuracy of the clean reference model.

## I.3. Transfer Learning with label noise: Training Recipe for Oxford Pets Dataset

This section documents the experimental setup behind Fig. 25 in Section 6.2.

**Dataset and task:** We use the Oxford-IIIT Pets dataset (Parkhi et al., 2012), which contains 3,680 `training` images and 3,669 `test` images across 37 fine-grained pet categories. All images are resized to $224{\times}224$ with standard ImageNet normalization ($\mu{=}[0.485, 0.456, 0.406]$, $\sigma{=}[0.229, 0.224, 0.225]$). We use the `test` split as a held-out validation set throughout. To simulate a noisy downstream fine-tuning scenario, 10% of the `training set` labels are flipped uniformly at random to a different class (fixed seed); the validation set always uses clean labels.

**Backbones and head replacement:** We evaluate four ImageNet-pretrained torchvision backbones: ResNet-18, ResNet-50, EfficientNet-B0, and ConvNeXt-Tiny. The original classification head of each backbone is replaced with a fresh randomly-initialized Linear($d$, 37) layer.

**Two-phase fine-tuning protocol:** Each (backbone, regularization configuration) cell is fine-tuned via a two-phase protocol using AdamW throughout. *Phase 1* (head-only warm-up): the backbone is frozen and only the new classification head is trained for 1 epoch at learning rate $10^{-3}$, preventing large random-head gradients from disrupting the pretrained backbone features at the start of training. *Phase 2* (joint fine-tuning): all parameters are unfrozen and trained jointly for 10 epochs with a learning rate of $10^{-3}$, with the learning rate divided by 10 at epoch 5. The cross-entropy loss uses the label-smoothing value $\varepsilon_{\text{LS}}$ specified by the regularization configuration. Each of the final finetuned model is referred to as a reference model.

**Regularization configurations:** Following the CIFAR-10 joint multi-regularization experiment (§I.1), we sample $N{=}5$ joint configurations ($w_{\text{WD}}$, $p_{\text{drop}}$, $\varepsilon_{\text{LS}}$) independently from the candidate grids

$$w_{\text{WD}} \in \{0,\ 10^{-6},\ 5{\times}10^{-6},\ 10^{-5},\ 5{\times}10^{-5},\ 10^{-4}\},$$
$$p_{\text{drop}} \in \{0,\ 0.1,\ 0.2\},$$
$$\varepsilon_{\text{LS}} \in \{0,\ 0.05,\ 0.1\},$$

using a fixed random seed, and apply the same five configurations to every backbone. The sampled configurations are: C0: ($w_{\text{WD}}{=}10^{-4}$, $p_{\text{drop}}{=}0.1$, $\varepsilon_{\text{LS}}{=}0.05$); C1: ($10^{-6}$, 0, 0); C2: (0, 0, 0); C3: ($5{\times}10^{-5}$, 0.1, 0.1); C4: ($10^{-5}$, 0.1, 0.1).

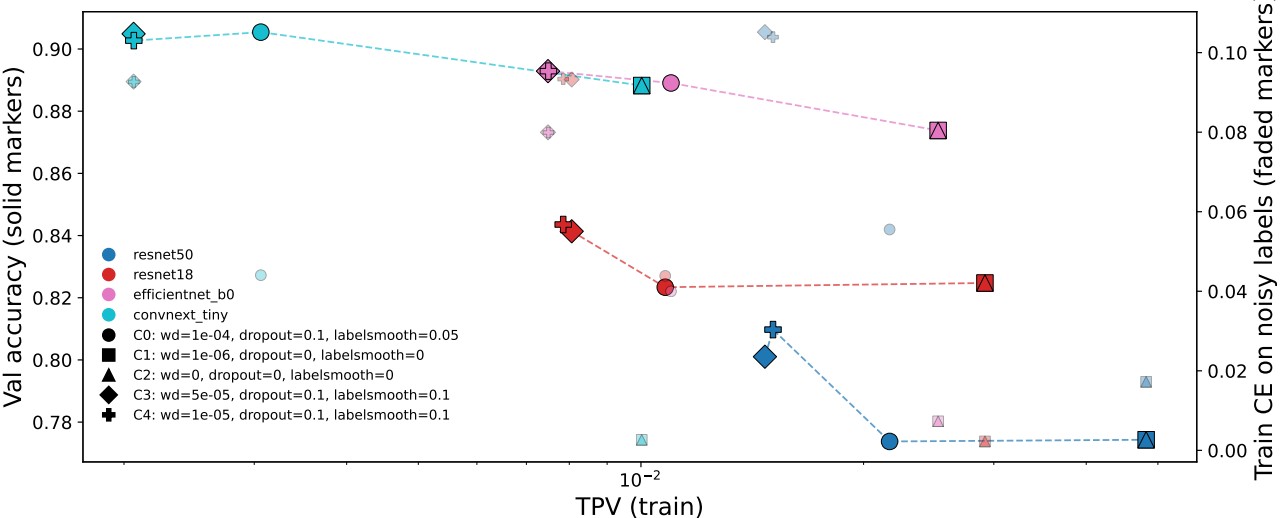

*Figure 25.* **TPV vs. validation accuracy on Oxford Pets under jointly-varied regularization.** Each color is one backbone; each marker shape is one of five joint regularization configurations (weight decay, dropout, label smoothing) sampled once and reused across all backbones. Solid markers (left axis): validation accuracy on clean labels. Faded markers (right axis): training cross-entropy on 10%-noisy label training set at $w^\star$. Dashed lines connect same-backbone points sorted by TPV. Within each backbone, lower TPV tracks higher validation accuracy, demonstrating that training-set TPV serves as a reliable label-free model-selection signal in the transfer-learning regime.

**TPV estimation:** At each $w^\star$ we estimate training-set TPV via the noisy-logit fine-tuning procedure described in Appendix D.3.4. Concretely, we add i.i.d. Gaussian noise with standard deviation $\sigma = c\,\sigma_0$, where $\sigma_0 = 0.1$ and $c$ is the mean absolute logit value of the reference model (adaptive noise scaling), then fine-tune a copy of the model on these noisy regression targets using AdamW at learning rate $10^{-6}$ for 5 epochs with no regularization and no proximity penalty. The empirical TPV is computed as the mean squared difference between the perturbed and reference logits over the training set, averaged over $R=5$ independent noise draws. All fine-tuning for TPV estimation is performed in `eval` mode (batch normalization and dropout inactive) to isolate the effect of label noise. See Appendix D.3.4 for details on the Label Noise TPV estimation algorithm.

**Results:** Figure 25 shows training-set TPV versus validation accuracy for all (backbone, configuration) cells. The faded markers on the right axis show training cross-entropy on the noisy labels at $w^\star$; all cells achieve low train CE ($\leq 0.10$), suggesting that the fine-tuning reaches the low training loss regime.

Within each backbone, configurations with lower TPV consistently achieve higher validation accuracy, with the dashed trend lines sloping downward across all four architectures. This mirrors the within-architecture pattern observed for CIFAR-10 MLPs in Figure 9 and confirms that training-set TPV aggregates the joint effect of weight decay, dropout, and label smoothing into a single scalar ranking without access to any test labels. Across architectures, the ordering is less preserved. Even though the overall TPV-validation accuracy trend remains negatively correlated, there is variation in validation accuracy across different architectures at similar TPV values. This variation needs further investigation as it hurts cross-architecture model selection reliability using TPV and we leave this for future work.

We point out that our use of Adam in the two-phase finetuning process above to achieve the reference models ensures a small training loss at the end, which is important because TPV-test loss correlation is positive in the small training loss regime (see section 5.3). In our preliminary experiments, we found that using SGD instead, which is known to achieve better generalization compared to Adam on vision tasks (especially in the presence to label noise), ended up at a much higher training loss in our fixed budget finetuning process, due to less memorization of the corrupted labels introduced in the training set. In this high training loss regime, we found that TPV-test loss correlation was poor as expected in this regime (see section 5.3).

## I.4. Sensitivity to Label Noise: Training Recipe Selection

This section documents the experimental setup behind Fig. 11 in Section 6.2.

**Setup** We train a four-layer ReLU MLP (input dimension 20, width 256, output 1) on a synthetic Gaussian linear teacher ($n_{\text{train}} = 3000$, $n_{\text{test}} = 5000$) using full-batch AdamW (lr$= 3 \times 10^{-3}$, 100 epochs) under seven weight-decay values $\lambda \in \{10^{-5}, 10^{-4}, 5 \times 10^{-4}, 10^{-3}, 3 \times 10^{-3}, 10^{-2}, 10^{-1}\}$. The resulting models $w^\star(\lambda)$ serve as reference points.

**Sharpness (SGD Noise TPV) estimation** For each $w^\star(\lambda)$ we estimate $\text{Tr}(H_{\text{eff}}(w^\star))$ using the doubly-stochastic Hutchinson Jacobian estimator with 100 Rademacher vectors.

We use $\text{Tr}(H_{\text{eff}})$ rather than the loss-Hessian trace $\text{Tr}(\nabla^2 L(w^\star))$ deliberately. The two coincide only at squared-loss minima with small residuals (Appendix E.2); off that regime they differ by the residual term $R(w^\star) = \frac{1}{n} \sum_i \varepsilon_i(w^\star) \nabla_w^2 f(x_i; w^\star)$, which is a function of the optimizer trajectory rather than the local Jacobian geometry. Since the trained models in this experiment are not near zero-residual minima, using the loss-Hessian trace would conflate Jacobian curvature with trajectory-dependent residual effects. $\text{Tr}(H_{\text{eff}})$ is the cleaner, more robust quantity for testing whether "Jacobian-spectral" sharpness predicts label-noise sensitivity.

**Label Noise TPV estimation** We use additive Gaussian noise with standard deviation $\sigma = c \sigma_0$, where $\sigma_0 = 0.1$ and $c$ is the mean absolute logit value of the reference model (adaptive noise scaling), $R = 50$ replicates. For each replicate we add i.i.d. Gaussian noise to the logits of $w^\star$ on the training inputs, reload $w^\star$, and fine-tune the model with MSE regression against the noisy target logits for 20 epochs using full-batch SGD on MSE loss (lr$= 3 \times 10^{-3}$, momentum$= 0.9$, $\lambda_{\text{wd}} = 0$, eval mode). We set proximity regularization weight $\gamma = 0$. See Appendix D.3.4 for details on the label-noise TPV estimation algorithm.

**Results** Figure 11 (main text) shows the key scatter plots. Across the weight-decay sweep, training-set label-noise TPV tracks test-set label-noise TPV closely (Spearman correlation near 1), so it correctly orders configurations by their actual label-noise sensitivity using only training-set quantities. Sharpness, in contrast, exhibits a strong *negative* Spearman correlation with test-set label-noise TPV in this sweep: the configuration with the lowest sharpness is in fact the most sensitive to label noise.

**Why sharpness can invert in sign** This sign inversion is not an artifact; it follows directly from Theorem 4.2. Both quantities are functions of the singular values $\{s_i\}$ of the output-parameter Jacobian $J$ at $w^\star$, but they emphasize opposite ends of the spectrum:

$$\text{TPV}_{\text{SGD}} \approx \frac{\eta \sigma_\varepsilon^2}{2b} \text{Tr}(H_{\text{eff}}) = \frac{\eta \sigma_\varepsilon^2}{2b} \sum_i s_i^2, \qquad \text{(dominated by the largest } s_i\text{)}$$

$$\text{TPV}_{\text{label}} \approx \sigma_\varepsilon^2 \sum_i \frac{B_{ii}}{s_i^2}. \qquad \text{(dominated by the smallest nonzero } s_i\text{)}$$

Regularization choices that primarily reshape the small-$s_i$ end of the spectrum therefore induce changes in label-noise TPV that sharpness cannot detect, and can even drive the two quantities in opposite directions across a sweep. We do not claim this sign inversion is universal — it depends on which part of the Jacobian spectrum the regularizer most affects — but its very possibility demonstrates that sharpness is an *unreliable* predictor of label-noise sensitivity. Training-set label-noise TPV, by contrast, measures the same Jacobian-spectral object as test-set label-noise TPV and is therefore robust by construction (Theorem 3.1). The practical implication is that a practitioner can use training-set label-noise TPV to select among candidate training recipes without access to clean test labels, with confidence that the ranking it produces will track test-time robustness to label noise.

