# OpenReview forum: "TPV: Parameter Perturbations Through the Lens of Test Prediction Variance"
_ICML.cc/2026/Conference — ICML 2026 regular_

### Official Review · Reviewer_QCHe · 2026-03-11

**Soundness:** 3
**Presentation:** 3
**Significance:** 1
**Originality:** 2
**Overall Recommendation:** 5
**Confidence:** 4

**Summary:**

This paper introduces Test Prediction Variance (TPV), which is the variance of solutions $f(x; w)$ derived from a model near convergence (with parameters $w^* $), due to perturbations such as SGD noise or label noise during further training, or quantization noise. Under an assumption that $f(x; w^* )$ can be linearized in a neighbourhood of $w^*$, derivations of TPV are given  to bound the difference between TPV computed on training vs test data, and give closed form expressions for TPV under label, SGD, and quantization noise.

**Compliance With Llm Reviewing Policy:**

Affirmed.

**Final Justification:**

My concerns have been resolved by the author replies which include an additional experiment. Please refer to the discussion for all of the details (there are many). The main issues turn out to be mostly in the presentation, as it was unclear what the purpose and application of TVP were. The added experiment helps tie all of the points together and also provides a novel result. Some citations were also missing that would clarify the text.

**Key Questions For Authors:**

There are other kinds of perturbations, such as adversarial perturbations, data distribution shifts, and which are of great interest both theoretically and practically. Can TPV be adapted to non-stationary and non-isotropic perturbations such as from these sources?

Minor question: does generalization in "irrespective of the model's generalization performance" (finding 2, line 90) and "TPV stability is decoupled from generalization" (line 376) refer to absolute performance (i.e. model test error)? It is a bit ambiguous and could mean TPV is decoupled from (an increase in) generalization risk/error", which would be contradictory.

**Limitations:**

The assumptions and regimes under which TPV results hold are discussed in detail. There are no societal impacts.

**Strengths And Weaknesses:**

Overall I think TPV is a really interesting tool and perspective on the loss landscape around trained solutions, but the interpretations and demonstrations in the paper were not chosen in a way that shows off TPV in the best light. The originality and significance might be improved simply by interpreting and applying TPV in some different settings.

**Soundness**: generally good. One issue is the experiments manipulate the Jacobian indirectly via width. It would be clearer to restate the derivations in terms of width to make the experiments easier to interpret.

**Presentation:** generally good. The paper's organization is a bit odd in my opinion: related work could be earlier because it is sorely needed for readers to understand the relative utility of TPV, and pruning (appendix H) is an interesting application that gets very little attention in the main text. Some minor points are:
- variables reused in slightly confusing ways, e.g. in the appendix the curly $\epsilon$ has a few different meanings
- Appendix H: MAC is not defined, and blue lines are reused in figure 14.

**Significance and originality:** this is where the paper in its current form falls short for me.
- Theorem 3.2 (TPV under label noise) only applies in a potentially narrow setting, as discussed under **Pathological Cases** and Appendix D.2. I'm not convinced that it is rare for $w^*$ to interpolate to the noisy labels, nor that it is rare for the rows of $J$ to be rank (assuming the dataset does not contain duplicate examples). Since Theorem 3.2 reduces to $\sigma_\epsilon^2$ in these pathological cases, it does not seem that useful overall. Perhaps some experiments showing when the conditions hold for Theorem 3.2 would be helpful here.
- Theorem 3.3 (TPV under SGD noise) should make it more explicit that we are looking at the variance due to a single SGD step at or near convergence, as opposed to say SGD noise applied throughout training. The connection to wide minima in section 3.2 does not seem that original: see Wu et. al 2022 (cited in the paper) and Wu, L., & Ma, C. "How SGD selects the global minima in over-parameterized learning: A dynamical stability perspective." (NeurIPS 2018).
- There is a lot of experimental effort spent on the correspondence between test loss and TPV, but what is the advantage of using TPV as a proxy for test loss when one can just have a held out test set?
- In the relevant plots (e.g. figures 4 and 6), the theoretical and empirical TPV estimates do not seem to align - are they supposed to predict each other in magnitude or in trend only, and by what margin of error? It's hard to interpret these results.
- In the context of the various theorems, how does one interpret the spectral properties of the Jacobian? That is, how does model width, depth, etc. affect TPV? Currently, the paper's results feel disjoint due to a lack of interpretation of the various quantities presented.
- There is a lot of emphasis that TPV is a local property of a trained model, but why does this make TPV significant? I can see a lot of potential applications (model compression, fine-tuning from pretrained models, meta learning, continual learning, privacy-preserving training, adversarial perturbation), but apart from compression (appendix H), none of these applications are really explored.

---

> ### Author Rebuttal · Authors · 2026-03-29
>
> We appreciate the detailed review. Several concerns reflect presentation gaps or underweighting of contributions already present in the paper. We address each point carefully.
>
> **On significance and originality**
>
> Several applications are already present but not sufficiently visible. (1) **JBR pruning** (Appendix H) matches or exceeds state-of-the-art across four benchmarks — we commit to moving it to the main text. (2) **Unification**: the paper provides the first unified analysis of SGD noise, label noise, and quantization under the single Tr($H_{\text{eff}}C$) template, where previously each required separate analyses. (3) **TPV trace stability** (Theorem 2.1) is the first result showing local prediction variance can be estimated from training inputs alone — distinct from NTK results which characterise training dynamics. A new Oxford Pets experiment (see hsh3 Weakness 1) confirms both TPV stability and the TPV-accuracy correlation hold under domain shift.
>
> **On Theorem 3.2 (label noise TPV) being narrow**
>
> Both pathological cases are treated in Appendix D.2. The experimental setup constrains noisy models near $w^*$ (proximity regularisation, Appendix D.3.4), preventing interpolation in the practically relevant regime. When interpolation does occur, TPV correctly collapses to $\sigma^2_\varepsilon$ for all architectures — the framework signals it cannot discriminate, like a thermometer at 0°C. We will add a clearer statement of the useful regime to the main text.
>
> **On Theorem 3.3 (SGD noise) and prior work**
>
> The novelty is not the flat-minima connection in isolation — that is well-studied (Wu et al. 2022, Wu & Ma 2018, both cited) — but that SGD noise, label noise, and quantization all appear through the same Tr($H_{\text{eff}}C$) template with only $C$ differing. We will also clarify in the theorem statement that it analyses the **stationary distribution of SGD near convergence**, not variance accumulated throughout training.
>
> **On the experimental emphasis on TPV-test loss correspondence**
>
> The deployment settings motivating TPV — fine-tuning for a new domain, quantizing for edge deployment — are precisely where held-out test sets are not cheaply available. Our Oxford Pets result (hsh3 Weakness 1) demonstrates this: training-set TPV predicts validation accuracy under domain shift (Spearman = -0.77), without test labels. Beyond scalar correlation, TPV identifies *which* perturbation mechanisms a model is sensitive to — information a test set cannot provide.
>
> **On the margin of error between theoretical and empirical TPV**
>
> Agreement in trend and ranking across widths — not absolute magnitude — is the meaningful criterion; Figures 4 and 12 clearly show this holds. The deviation in magnitude is expected since empirical TPV uses SGD fine-tuning as a proxy for the exact min-norm solution. We will state this criterion explicitly in Section 5.2.
>
> **On the paper feeling disjoint and spectral interpretation**
>
> In revision: (a) a roadmap paragraph in Section 2 with a summary table of perturbation sources and their $C$ matrices; (b) JBR moved to main text; (c) related work restructured earlier. Width affects TPV through conditioning of $H_{\text{eff}}$ — wider networks have better-conditioned Jacobians, suppressing $B_{ii}/s_i^2$ terms in Theorem 3.2, as shown in Figure 12.
>
> **On the significance of TPV being a local property and unexplored applications**
>
> Locality matters for three reasons: (i) post-training perturbations (SGD noise, quantization, pruning, label noise during fine-tuning) all act locally around a fixed $w^*$ — global variance is the wrong object; (ii) TPV stability enables training-set-only diagnostics not possible with global variance; (iii) TPV is perturbation-source-specific, identifying which mechanisms a model is sensitive to. **On unexplored applications**: we see the reviewer's own list as a strong indicator of the paper's breadth, not a weakness — each application requires its own experiments and baselines. This paper establishes the theoretical foundations, proves the stability result, validates across CIFAR-10/100 and ImageNet, and contributes JBR. We will restructure the conclusion to separate contributions from future directions.
>
> **On adversarial perturbations and distribution shift**
>
> Adversarial and distribution shift perturbations can be analyzed within the TPV framework one step upstream: they act on inputs, changing the loss, inducing $\delta w$ through fine-tuning dynamics — exactly as label noise does (Section 3.1). Deriving $C$ induced by input-space perturbations is a natural extension we will add to the future work section.
>
> **On the ambiguity in "decoupled from generalization"**
>
> "Decoupled from generalization" means train/val TPV agreement holds regardless of the generalization gap $L_{\text{test}} - L_{\text{train}}$ — not that TPV is uncorrelated with test error (Section 5.2 shows it is). We will add a clarifying sentence.

---

> > ### Author Rebuttal · Reviewer_QCHe · 2026-04-04
> >
> > Thank you for the detailed reply. I have further comments to your points below.
> >
> > **Unification:** while I appreciate the elegance of unifying different perturbations under one analysis, I am not entirely sure that the results are presented in a unified manner. Perhaps this is simply an issue of presentation, as it isn't even obvious that $Tr(C)$ is common to theorems 3.2 and 3.3 due to the notation, and it's also hard to understand from the main text how $Tr(C)$ changes depending on model properties such as width - one needs to look at the proofs in the appendix.
> >
> > **Theorem 3.2:** I appreciate the additional discussion of pathological cases in Appendix D.2, but they do not really address my concern, which was about how common the pathological cases are for practical purposes. More simply, I would like to know:
> > 1. how large is the "sufficiently large" perturbation needed to leave the linearization regime, and how large is this perturbation relative to what one might encounter in practical settings (e.g. say, 5% label noise)
> > 2. for some reasonable value(s) of $\sigma^2_\epsilon$, how likely is $w*$ to interpolate?
> >
> > **Theorem 3.3:** my issue here is that the text in section 3.2 does not cite prior works that support the wide minima hypothesis, so TPV's support for the hypothesis looks more original than it is. Other than supporting the wide minima hypothesis, what other insights can be offered?
> >
> > **TPV and test loss:** I agree there is great potential here for TPV to be a useful metric for practical diagnostics, but I think that is a) out of scope in this work, and b) not well supported by empirical results presently. Re. *domain shift*, I don't think a correlation of 0.77 is alone enough to say that TPV should be used a proxy for validation accuracy when doing model selection, especially when the experiment is done in a single, fairly small setting. Re. *identifying what perturbation mechanisms a model is sensitive to*, I'm not sure this is of great practical utility:
> > 1. SGD noise, quantization noise, and label noise are typically not things practitioners are trading off with each other. It would be more useful if, for instance, TPV could be used to decide on whether to do more pruning vs quantization to compress a model.
> > 2. As shown in plots, the bounds in equations 10-12 are not that tight, so I am skeptical TPV could really help practitioners decide on a tradeoff.
> > 3. When deciding on a tradeoff, one could also just test them empirically, i.e. try different combinations and directly compare their validation accuracy (or TPV in place of validation acc).
> >
> > **Theoretical vs empirical TPV:** I think there are 2 ways to go about this: if the theoretical TPV is a tight bound on empirical TPV then one can use the theory to make decisions without empirical experiments. However, if the theory is a loose bound (as is currently in the paper), then we want the theory to predict interesting relationships between TPV and variables such as model width. My issue is that the trends presented are just not that surprising. I do not want to underplay the support TPV gives to existing hypotheses (wide minima, model width trending towards NTK), which is valuable(!), but more could be said towards originality and significance.
> >
> > **Significance:** could you clarify what kind of "global variance" measures you are referring to?
> >
> > **Adversarial and domain shift:** how are these equivalent to label noise when the training data has also been changed?

---

> > > ### Author Response · Authors · 2026-04-05
> > >
> > > We thank the reviewer for the detailed follow-up. We address each point below, including a new experiment run.
> > >
> > > **Unification:** This is a presentation issue we will fix. We will add a roadmap paragraph in Section 2 that explicitly writes out $\text{Tr}(H_\text{eff} C)$ as the common object and shows how each theorem instantiates it.
> > >
> > > **Theorem 3.2 — conditions for the useful regime:** The reviewer's original concern was that $\text{rank}(J) = n$ is typical in overparameterized networks, causing the linearized model to interpolate noisy labels and collapsing Theorem 3.2 to $\sigma^2_\varepsilon$.
> > >
> > > We want to first clarify the practical motivation. In realistic fine-tuning scenarios, a practitioner has a fixed $w^*$ and a downstream dataset with *unknown* label noise. The question is: which model is more robust to fine-tuning on this noisy data (typically requires access to a clean test set)? TPV measures exactly this robustness, and Theorem 2.1 guarantees training-set TPV is a reliable estimator of test-set TPV (no test set required) — the practitioner does not choose $\sigma$, it is determined by the data.
> > >
> > > $\lambda$ (proximity regularization) is the primary practical knob. It keeps $\delta w$ local around $w^*$, ensuring the first-order approximation holds and TPV remains discriminative. In practice, for our CIFAR-100 MobileNetV2 experiments with $\lambda=0$ and $\sigma=0.1$, Figure 7 shows train and test TPV tracking each other and correlating with CE loss — confirming the useful regime is reached without proximity regularization. Across most experiments in the paper, $\lambda=0$ was sufficient. But in general, a practitioner can use the first-order validity check (Appendix G.1.1) to know if the first-order assumption is intact.
> > >
> > > Finally, Theorem 3.2 extends the benign overfitting result of Bartlett et al. (2020) from linear to non-linear networks — itself a new result — and uses NTK connections to explain why overparameterized networks are more robust to label noise in practice.
> > >
> > > **Theorem 3.3 — prior work and additional insights:** We will add citations to Wu et al. (2022), Wu & Ma (2018), and Keskar et al. (2017) directly in Section 3.2 (currently listed in Appendix E.1). Our new experiment (below) provides an insight beyond wide minima: the optimal regularization regime for label-noise robustness does not coincide with the flattest minimum — a finding sharpness-based reasoning would not predict.
> > >
> > > **TPV practical utility and new experiment:** We address the reviewer's concern with a new controlled experiment showing that sharpness does NOT reliably predict label-noise sensitivity while label-noise TPV (theorem 3.2) does (please see Reply Rebuttal Comment to reviewer Ufri for further details). We train the same architecture (MLP, width=128, depth=4) under six weight decay values 0 to 3 × $10^{−3}$ on a synthetic dataset. For each $w^*$ we compute: (a) sharpness = Tr(∇2L(w∗)) via Hutchinson estimator — training data only; (b) training-set TPV — inject Gaussian label noise (σ=0.01), fine-tune from w∗, measure prediction variance on the *training set* — no test labels; (c) test-set TPV — same on the *test set* — ground truth.
> > >
> > > Result: Spearman(sharpness, TPVtest)=−0.257 (wrong direction); Spearman(TPVtrain, TPVtest)=0.771. Label-noise training-set TPV correctly identifies wd $=10^{-4}$ as the most label-noise-robust recipe. A practitioner can run this on the training set alone to select the best model without test labels.
> > >
> > > On the **"practitioners don't trade off between SGD noise, quantization, and label noise"** point: we agree, and want to clarify that we never claimed they do. The value of the unified framework is that a practitioner with a specific deployment concern (e.g. label noise) uses the corresponding TPV variant as their diagnostic (kindly see Reply Rebuttal Comment to Ufri for details). The unification means one framework handles all cases rather than requiring separate tools.
> > >
> > > **Global variance:** "Global variance" averages prediction variance across all solutions the learning algorithm can produce- across seeds, initializations, and data samples — as in Neal et al. (2018) and Yang et al. (2020). TPV is strictly local around one fixed $w^*$.
> > >
> > > **Adversarial perturbations and domain shift:** We do not claim equivalence to label noise. The connection is mechanistic: input perturbations triggering fine-tuning from $w^*$ induce $\delta w$ through the loss gradient, as label noise does. Under the first-order approximation, the loss under input perturbation $\eta$ is
> > >
> > > $$\mathcal{L}(\delta w;\, \eta) = \tfrac{1}{2}\| f(w^* + \delta w,\, x + \eta) - y \|^2 \approx \tfrac{1}{2}\| J_w\, \delta w + J_x\, \eta \|^2,$$
> > >
> > > giving $\delta w$ with covariance $C_\text{input} \propto J_w^{+} J_x\, \text{Cov}(\eta)\, J_x^\top (J_w^{+})^\top$, feeding into $\text{Tr}(H_\text{eff} C)$. We leave the full analysis of input distribution shift under the TPV framework as future work.

---

### Official Review · Reviewer_Ufri · 2026-03-13

**Soundness:** 2
**Presentation:** 3
**Significance:** 3
**Originality:** 3
**Overall Recommendation:** 4
**Confidence:** 2

**Summary:**

This paper proposes a local metric, Test Prediction Variance (TPV), to evaluate the degree to which the predictions of a trained deep neural network change when subjected to parameter perturbations. Unlike traditional bias-variance analysis, which requires retraining the model to calculate global variance, TPV focuses on the local sensitivity near a fixed solution. Based on a first-order Taylor expansion, the paper approximates TPV as a concise trace form $\mathrm{Tr}(H_{\text{eff}}C)$, thus decoupling the learned geometry (represented by the second moment of the Jacobian matrix $H_{\text{eff}}$) from the specific perturbation mechanism (represented by the perturbation covariance matrix $C$). Through this unified framework, the paper analyzes various real-world sources of parameter perturbations, including SGD steady-state noise, label noise, quantization noise, and pruning. Theoretically, the authors prove that under overparameterization, infinitely wide networks and certain assumptions, the TPV estimated using the training set converges to the TPV on the test set. Experiments show that this "TPV stability" remains valid under a wider range of conditions and is decoupled from the generalization gap. Furthermore, the empirically estimated TPV is highly correlated with the test loss and can serve as a generalization evaluation metric without test labels, leading to a new and effective network pruning criterion (JBR).

**Compliance With Llm Reviewing Policy:**

Affirmed.

**Key Questions For Authors:**

1. In the CIFAR/ImageNet experiments, label noise is modeled as continuous Gaussian perturbations on logits and optimized with MSE. In standard classification settings with cross-entropy loss and discrete label-flipping noise, does the TPV framework, especially Theorem 3.2 still apply? If so, what modifications would be required?

2. Can TPV be further leveraged to design stronger algorithms for solving practical problems, rather than primarily serving as an interpretive or diagnostic quantity? For example, could it be used to guide training, model selection, robustness improvement, or compression in a more direct way?

**Limitations:**

yes

**Strengths And Weaknesses:**

Strengths

1. The trace-form characterization, $\mathrm{Tr}(H_{\text{eff}}C)$, cleanly disentangles the model’s learned geometric structure from the external perturbation mechanism. This provides a unified framework for interpreting several seemingly different phenomena, including flat minima, quantization noise, and label noise.

2. Beyond the theoretical result establishing the asymptotic equivalence between training-set TPV and test-set TPV (under NTK-style assumptions), the paper also provides broad empirical validation on both synthetic and real image datasets. Notably, the observed TPV stability appears to hold even beyond the regime covered by the theory.

Weaknesses

1. Label-noise setup may not fully reflect standard classification noise. In the experiments, “label noise” is simulated by injecting continuous Gaussian noise into logits and then fine-tuning with an MSE objective. This differs from the more common discrete label-flipping setting in classification, as well as from the training dynamics induced by cross-entropy loss. As a result, it is unclear how well the experimental setup captures realistic label-noise behavior in standard classification tasks.

2. Heavy reliance on a first-order local approximation. The framework depends critically on a local first-order Taylor expansion around the trained solution. When the perturbation magnitude becomes larger (e.g., under stronger label noise), the model may leave the local linear regime, at which point TPV stability can break down.

3. Limited algorithmic payoff at this stage. While TPV is an interesting analytical lens, the paper does not yet develop a clearly strong algorithmic method based on TPV that substantially improves performance on practical tasks.

---

> ### Author Rebuttal · Authors · 2026-03-29
>
> We appreciate the review and the important questions raised. We address all the concerns of the reviewer below.
>
> **Weakness 1: Label noise setup may not reflect standard classification noise**
>
> The continuous Gaussian logit-noise model with MSE is deliberately chosen to enable the analytical derivation in Theorem 3.2 — the min-norm solution structure and spectral decomposition (Eq. 10) require a continuous, zero-mean perturbation with a well-defined covariance $C$. For discrete label-flipping under cross-entropy loss, the Tr($H_{\text{eff}}C$) template requires the induced $\delta w$ to remain within the local linear regime, and for aggressive flipping this cannot be guaranteed. We therefore cannot claim the template applies in general to this setting and will reflect this in the future work section. We also note the logit-noise setup is directly motivated by fine-tuning on continuous noisy annotations or soft labels — arguably the most practically relevant setting for label noise at convergence.
>
> **Weakness 2: Heavy reliance on first-order approximation**
>
> We first question whether this constitutes a fundamental limitation. TPV measures robustness to small, realistic perturbations — SGD noise near convergence, quantization, label noise during fine-tuning. These are all typically small in nature; a deployed model is not subjected to arbitrarily large weight perturbations. The first-order approximation is therefore not a restriction but a natural fit — it is valid precisely in the regime where TPV is meaningful. Two further points: (i) the paper already discusses where the approximation breaks (Appendix D.2) and provides a quantitative validity check that discards runs where it fails (Appendix G.1); (ii) The utility of TPV as a principled framework holds even when the first-order approximations do not hold (e.g. structured pruning perturbations in Appendix H).
>
> **Weakness 3: Limited algorithmic payoff**
>
> We push back on this characterisation. Appendix H presents JBR, a pruning criterion derived directly from TPV geometry that matches or exceeds state-of-the-art across four benchmarks (ResNet-56/CIFAR-10, VGG-19/CIFAR-100, ResNet-50/ImageNet, MobileNet-V2/ImageNet) without fine-tuning. This contribution is currently buried in the appendix — we commit to moving it to the main text. Beyond pruning, our new Oxford Pets experiment (see hsh3 Weakness 1) demonstrates that TPV computed on a downstream training set predicts validation accuracy across architectures without test labels (Spearman = -0.77), confirming immediate practical value for model selection under domain shift.
>
> **Q1: Does Theorem 3.2 apply with cross-entropy and discrete label-flipping?**
>
> As discussed under W1, extending to discrete label-flipping requires verifying that the induced $\delta w$ stays small — which cannot be guaranteed in general. Regarding CE loss specifically: the bias-variance decomposition of CE test error has the same structural form as MSE, and TPV (Eq. 2) remains a well-defined local variance functional under CE loss. The key open step is deriving the closed-form $C$ induced by label noise under CE gradient dynamics, which we leave for future work.
>
> **Q2: Can TPV be leveraged for stronger algorithms beyond diagnosis?**
>
> JBR (Appendix H) is one concrete algorithm already in the paper. Beyond pruning: (i) TPV-guided training could regularise toward lower-TPV solutions; (ii) TPV could guide model selection during fine-tuning without test labels, as demonstrated by the Oxford Pets result. We will expand these directions in the conclusion.

---

> > ### Author Rebuttal · Reviewer_Ufri · 2026-04-04
> >
> > Thank you for the clarifications. I appreciate the elegant TPV framework and its trace-form unification of optimization geometry with perturbation mechanisms.
> >
> > However, my concern regarding the limited algorithmic payoff is only partially resolved. While I acknowledge the JBR pruning criterion presented in the appendix (and your commitment to moving it to the main text), the methodological innovation based on the TPV framework is somewhat limited. As explicitly noted in Appendix H.1, the JBR method is very similar to the existing Jacobian Criterion (JC), essentially acting as a label-free version of it where the primary difference is the choice of the logit-space vector.
> >
> > I will maintain my current score.

---

> > > ### Author Response · Authors · 2026-04-05
> > >
> > > We thank the reviewer for the continued engagement. We want to highlight two experimental results that directly address the concern about limited algorithmic/diagnostic payoff.
> > >
> > > **New experiment — TPV as a training recipe selector:** Consider a practitioner who has trained the same architecture under different hyperparameter choices and wants to select the recipe most robust to label noise during subsequent fine-tuning — without running any test evaluation (e.g. clean test set not available). We show that training set label noise TPV (theorem 3.2) as a metric successfully selects the most robust model among the candidates. We use sharpness Tr(∇2L w∗)) as a baseline, since by Theorem 3.3 it is proportional to SGD noise TPV— the correct metric for *SGD noise* sensitivity. We show that the latter fails at predicting *label noise* sensitivity. This is precisely the point: the TPV framework provides the right metric for each noise source — using sharpness to predict label-noise robustness is using the wrong TPV variant for the task.
> > >
> > > Specifically, we train the same architecture (MLP, width=128, depth=4) under six weight decay values 0 to 3 × $10^{−3}$ on a synthetic dataset. For each $w^*$ we compute: (a) sharpness = Tr(∇2L(w∗)) via Hutchinson estimator — training data only; (b) training-set TPV — inject Gaussian label noise (σ=0.01), fine-tune from w∗, measure prediction variance on the *training set* — no test labels; (c) test-set TPV — same on the *test set* — ground truth.
> > >
> > > Result: Spearman(sharpness, TPVtest)=−0.257 (sharpness actively misleads, pointing in the wrong direction); Spearman(TPVtrain, TPVtest)=0.771. Training-set TPV correctly identifies wd $=10^{-4}$ as the most label-noise-robust recipe, a finding sharpness misses. A practitioner can run this on the training set alone to select the best model without test labels.
> > >
> > > **Oxford Pets domain shift experiment (from our original rebuttal, unaddressed in follow-up):** Consider a practitioner who has several ImageNet-pretrained architectures and wants to deploy one on a new downstream domain (Oxford Pets). They have a small labeled training set for the new domain but no test labels. The question is: which architecture will generalize better on this domain? We show that training-set TPV — computed by injecting label noise, fine-tuning from w∗, and measuring prediction variance on the *downstream training set only* — answers this question without any test labels. Specifically, we take six ImageNet-pretrained architectures (ResNet18, ResNet50, WideResNet50-2, EfficientNet-B0, MNASNet1\_0, ConvNeXt-Tiny), adapt them to Oxford Pets, and estimate training-set TPV. Training-set TPV predicts validation accuracy across all six architectures with Spearman =−0.77 (lower TPV correlates with higher accuracy). This is a direct, practical use case: label-free model selection under domain shift.
> > >
> > > **On JBR and JC:** We acknowledge that JBR can be seen as a label-free variant of JC, as stated explicitly in Appendix H.1. The contribution is that the TPV framework provides a principled first-principles derivation that recovers the label-free version of JC as a special case — the fact that it converges to an existing strong baseline validates the framework rather than limiting it.

---

### Official Review · Reviewer_hsh3 · 2026-03-13

**Soundness:** 3
**Presentation:** 3
**Significance:** 2
**Originality:** 3
**Overall Recommendation:** 4
**Confidence:** 3

**Summary:**

In this paper, the authors study the variance of a trained model's predictions under small parameter perturbations quantified by test prediction variance (TPV).  The perturbations studied in the paper include label noise, SGD noise, and common post-hoc interventions like quantization and pruning. The authors provide a clean label-free form for TPV that depends on the trace of second moment of the Jacobian multiplied by the covariance of the noise. The derivation follows from the assumptions on NTK and isotropic covariance of the noise. The authors provide strong evidence for TPV stability---i.e. TPV estimated from the training set is a reliable estimator of the actual test prediction.

**Compliance With Llm Reviewing Policy:**

Affirmed.

**Final Justification:**

I stand with my original assessment, the authors addressed most of my questions and when small changes are incorporated for better clarity and the significance, the paper would be a valuable addition to the literature.

**Key Questions For Authors:**

1. How realistic is assumption A.3 (isotropic covariance)? For example for pruning this will likely not hold.
2. How did the authors arrive at 1/12 scaling (in Equation 12) for quantization?
3. I am not sure if I totally understand the w=1 in the synthetic case and n_train=1 in Cifar experiments. Wouldn't we expect the setting with 1 example to be extremely noisy? Could the authors please elaborate on the significance of these two edge cases?
4. I am a bit confused on how to interpret the figures.
	1. For example in Figure 7, right TPV test and train follow a very similar trend and they look close enough but at the same time the the absolute scale is hard to interpret. Do the authors have an insight about the scale of TPV (aside from scaling with $\sigma^2$)?
	2. In Figure 7, what is the CE test loss of the reference model?
	3. Similarly what makes $\sigma=0.1$ a large noise in Figure 6, but not so large in Figure 7?
5. I believe it would be interesting if the authors could expand this notion for linear mode connectivity and model merging literature too. To me it sounds like TPV is a local analysis of what LMC studies globally. For example [1] study such small perturbations during pre-training and fine-tuning, and show divergence (from the basin of the spawned copy) with respect to noise and time. And I believe model merging ($\alpha \theta_A + (1-\alpha) \theta_B$) in a basin could be formulated within the perturbations framework of this work.
6. Do the authors have a potential explanation of why resnet18 model deviates from the general trend in Figure 5?


[1] Altıntaş, et al. The Butterfly Effect: Neural Network Training Trajectories Are Highly Sensitive to Initial Conditions. 2025

**Limitations:**

yes

**Strengths And Weaknesses:**

**Strengths**
- Studying stochastic gradient noise near convergence is well-motivated.
- The derivation of the TPV is clear in most perturbations considered in the paper. There is generally good experimental support for the claims (synthetic setting, linear regression, Cifar/Imagenet scale experiments).


**Weaknesses**
- I think the theory presented in the paper is strong and sound. However I don't think there is enough evidence in the paper for TPV being a "new post-training diagnostic tool [...] for characterizing model robustness and generalization behavior [...], making TPV a practical diagnostic for deployed models".
- I think one missing piece in the paper is a fine-tuning setting where the target domain doesn't completely match the pre-training data. For example, I feel like if the experiments on ImageNet were to be repeated for some downstream tasks (could be cars, Oxford Pets, EuroSAT) and like domain shift benchmarks, e.g. ImageNet-C, the authors could have a stronger claim for the point above.
- Presentation: The paper is dense and the reader is often left to read the proofs in the corresponding Appendix sections without an intuitive explanation
	- Appendix should have a organization summary paragraph.
	- In pruning figures JBR and WHC share the same color, while one blue performs poorly the other performs well.
	- Abbreviations should be named at the first place they appear. For example LLN is explained much later. Pruning figures use MACs at the x-axis, I am not sure what MACs are, is it multiply-accumulate?
	- Another minor point: the font size mismatch across figures

---

> ### Author Rebuttal · Authors · 2026-03-29
>
> We thank the reviewer for the thoughtful and constructive feedback. Due to space constraints we focus on the main concerns below, with additional clarifications available upon request.
>
> **Weakness 1: Insufficient evidence for the practical diagnostic claim**
>
> We have run an additional experiment on Oxford-IIIT Pets directly in response to this concern (results at https://ibb.co/5W7BhMHY; to be included in the revision). Oxford Pets represents a genuine domain shift from ImageNet. We take ImageNet-pretrained models (ResNet18, ResNet50, WideResNet50-2, EfficientNet-B0, MNASNet1\_0, ConvNeXt-Tiny), replace and briefly fine-tune their classification heads on Oxford Pets, and estimate empirical TPV under label noise using only the Oxford Pets training set — without any access to validation labels. The result mirrors Figure 5: TPV computed on the downstream training set correlates negatively with validation accuracy across all six architectures (Spearman = -0.77), and train/val TPV agree closely (mean relative difference 4%, max 6.5%), confirming TPV stability holds under domain shift. A practitioner deploying a pretrained model on a new domain can thus use the downstream training set alone to estimate which architecture will generalise better, without test labels.
>
> **Weakness 2: Presentation**
>
> We will add intuitive explanations before each theorem and an organisation summary paragraph at the start of the appendix, and fix all minor presentation issues (figure colors, abbreviations, font sizes).
>
> **Q1: How realistic is Assumption A.3 (isotropic covariance) for pruning?**
>
> TPV stability (Eq. 2) and TPV trace stability (Eq. 6, Theorem 2.1) are different objects — footnote 1 (Pg. 7) makes this explicit. Assumption A.3 is only required for Theorem 2.1. The empirical TPV stability results throughout Section 5 use Eq. 2 directly and do not require isotropic covariance. For pruning specifically, JBR uses the structured, explicitly non-isotropic covariance $C_{\text{prune},g} = \sigma^2 w_g w_g^\top$ (Eq. 76) — the isotropic assumption is never invoked. We will add a clarifying remark to make this distinction more prominent.
>
> **Q2: How did the authors arrive at the 1/12 scaling for quantization?**
>
> Under $\delta w_j \sim \text{Unif}(-\delta/2, \delta/2)$, the variance is $(\delta/2)^2/3 = \delta^2/12$ by the standard result $\text{Var}(\text{Unif}[-a,a]) = a^2/3$. We will add this derivation to the paper.
>
> **Q3: Significance of w=1 and n_train=1**
>
> These are intentional stress tests mapping the boundary conditions of TPV stability, not representative operating points. w=1 is extreme — finding stability holds there is surprising given the theory requires $m\to\infty$, indicating stability is more intrinsic than the theory formally predicts. n_train=1 is shown precisely because this is where stability breaks, consistent with the $o_{n_{\text{tr}},n_{\text{te}}}(1)$ term in Theorem 2.1. Together they trace where the property holds and where it fails.
>
> **Q4: Interpretation of figures**
>
> **Q4a:** The absolute scale of TPV depends on $\sigma^2$, output dimension $K$, and Jacobian geometry. The intended interpretation is the relative ordering across architectures and the agreement in trend between train and test TPV. We will add a clarifying note to the caption.
>
> **Q4b:** The reference model CE test loss closely tracks the noisy model CE test loss, making it visually hidden behind the noisy model curve. The two are more distinct in Fig. 11 (CIFAR-10 version).
>
> **Q4c:** The difference between Figures 6 and 7 is about the magnitude of the induced parameter perturbation covariance $C$. The Theorem 2.1 bound on $|TPV_{train} - TPV_{test}|$ scales with Tr($C$). In Figure 6, $\sigma=0.1$ on a simple synthetic task induces large Tr($C$) — the noisy model moves far from $w^*$ — making the bound loose and breaking stability. In Figure 7, the same $\sigma=0.1$ applied to MobileNetV2 on CIFAR-100 induces small $\delta w$ relative to the parameter space due to higher task dimensionality, larger model capacity, and short fine-tuning — Tr($C$) stays controlled and stability holds. What matters is not $\sigma$ alone but the induced parameter perturbation magnitude.
>
> **Q5: Connection to LMC and model merging**
>
> LMC studies the global landscape between two solutions; TPV is strictly local around a single solution. Model merging $\alpha\theta_A + (1-\alpha)\theta_B$ fits the TPV framework for small $\alpha$: it represents a perturbation $\delta w = \alpha(\theta_B - \theta_A)$ around $\theta_A$, and TPV predicts the resulting prediction variance. We will add this connection to the related work section.

---

> > ### Author Rebuttal · Reviewer_hsh3 · 2026-04-01
> >
> > I thank the authors for the clarifications and Oxford Pets experiments though the link doesn't display the image.

---

### Decision · Program_Chairs · 2026-04-30

**Decision:**

Accept (regular)

**Comment:**

The paper introduces Test Prediction Variance (TPV), a local, label-free quantity measuring how much a trained model’s predictions change under small parameter perturbations around a fixed solution. The key idea is that many perturbation types, such as SGD noise, label noise during fine-tuning, quantization noise, and pruning-like perturbations, can be analysed in one framework through a trace expression that separates the model’s learned geometry from the perturbation covariance. The main theoretical claim is TPV stability: in the overparameterized limit, TPV estimated on the training set converges to TPV on the test set, so test-time sensitivity can be inferred from training inputs alone. Empirically, the paper reports that this stability holds surprisingly broadly, even beyond the formal theory, and that TPV correlates well with test loss.

All reviewers agreed to accept this paper based on grounds including (1) the paper was interesting, technically solid, (2) the paper is potentially useful, (3) the clean trace-form view, (4) the unification of several perturbation mechanisms under one lens, and (5) the breadth of experiments spanning synthetic settings, linear regression, CIFAR, and ImageNet-scale models.